

# Accurate and precise quantification of atmospheric nitrate in streams draining land of various uses by using triple oxygen isotopes as tracers

**Urumu Tsunogai[1], Takanori Miyauchi[1], Takuya Ohyama[1], Daisuke D. Komatsu[1,*], Fumiko Nakagawa[1], Yusuke Obata[2], Keiichi Sato[3], and Tsuyoshi Ohizumi[3]**

[1]{Graduate School of Environmental Studies, Nagoya University, Furo-cho, Chikusa-ku, Nagoya 464-8601, Japan}

[2]{Faculty of Bioresources, Mie University, 1577 Kurimamachiya-cho, Tsu 514-8507, Japan}

[3]{Asia Center for Air Pollution Research, 1182 Sowa Nishi-ku, Niigata 950-2144, Japan}

[*]{now at School of Marine Science and Technology, Tokai University, 3-20-1 Orito, Shimizu, Shizuoka 424-8610, Japan}

Correspondence to: U. Tsunogai (urumu@nagoya-u.jp)

## Abstract

$^{17}O$ anomalies were used to quantify the influence of changes in land use and population density between each catchment area on the fate of atmospheric nitrate by determining the areal distribution and seasonal variation in stable isotopic compositions including the $^{17}O$ anomalies ($\Delta^{17}O$) of nitrate for more than 30 streams within the same watershed. Those in precipitation (wet deposition; $n$ = 213) sampled at Sado-seki monitoring station were determined for three years as well. The deposited nitrate showed similar large $^{17}O$ anomalies with those already reported for mid-latitudes: $\Delta^{17}O$ values ranged from +18.6‰ to +32.4‰ with a three-year average of +26.3‰. However, nitrate in each inflow stream showed small annual average $\Delta^{17}O$ values ranging from +0.5‰ to +3.1‰, which corresponds to the mixing ratios of unprocessed atmospheric nitrate to total nitrate from 1.8 ± 0.3% to 11.8 ± 1.8%, with 5.1 ± 0.5% as the average of all inflow streams. Although the annual average $\Delta^{17}O$ values





tended to be smaller in accordance with the increase in annual average nitrate concentration
from 12.7 to 106.2 $\mu mol\ L^{-1}$, the absolute concentrations of unprocessed atmospheric nitrate
in the streams were almost stable at $2.3 \pm 1.1\ \mu mol\ L^{-1}$ irrespective of the changes in
population density and land use in each catchment area. We conclude that changes in
population density and land use between each catchment area had little impact on the
concentration of atmospheric nitrate. Thus, the total nitrate concentration originated primarily
from additional contribution of remineralized nitrate from both natural sources, having values
of $+4.4 \pm 1.8$‰ and $-2.3 \pm 0.9$‰ for $\delta^{15}N$ and $\delta^{18}O$, respectively, and anthropogenic sources
having values of $+9.2 \pm 1.3$‰ and $-2.2 \pm 1.1$‰ for $\delta^{15}N$ and $\delta^{18}O$, respectively. In addition,
both the uniform absolute concentration of atmospheric nitrate and the low and uniform $\delta^{18}O$
values of the remineralized portion of nitrate in the streams imply that in-stream removal of
nitrate through assimilation or denitrification had small impact on the concentrations and
stable isotopic compositions of nitrate in the streams, except for a few streams in summer
having catchments of urban/suburban land uses. Additional measurements of the $\Delta^{17}O$ values
of nitrate together with $\delta^{15}N$ and $\delta^{18}O$ enabled us to exclude the contribution of unprocessed
atmospheric nitrate from the determined $\delta^{15}N$ and $\delta^{18}O$ values of total nitrate and to use the
corrected $\delta^{15}N$ and $\delta^{18}O$ values to evaluate the source and behaviour of the remineralized
portion of nitrate in each stream.

## 1   Introduction

Nitrate ($NO_3^-$) in stream water can be an important clue for understanding the biogeochemical
cycles within its catchment area (Durka et al. 1994; Likens et al. 1970; Swank et al. 2001). In
addition, the nitrate concentration in stream water is important to primary production and thus
eutrophication downstream, including lakes, estuaries, and oceans (Mcisaac et al. 2001; Paerl
2009). Nitrate concentrations in stream water, however, are determined through a complicated
interplay of several processes within its catchment area including (1) the addition of
atmospheric nitrate ($NO_3^-{}_{atm}$) through deposition, (2) the production of remineralized nitrate
($NO_3^-{}_{re}$) through microbial nitrification, and (3) the removal of nitrate through assimilation
and denitrification by plants and microbes. In addition to natural processes, anthropogenic
processes could have a significant impact on the nitrate dynamics within each catchment area,
particularly for those eluted from urban or agricultural catchment zones. Therefore,




interpretation of the processes regulating nitrate concentration in stream water is not always
straightforward.
The $^{15}N/^{14}N$ and $^{18}O/^{16}O$ ratios of nitrate have been widely applied in the determination of the
sources and behaviours of nitrate in stream water worldwide (Barnes et al. 2008; Barnes and
Raymond 2010; Burns et al. 2009; Burns and Kendall 2002; Campbell et al. 2002; Campbell
et al. 2006; Costa et al. 2011; Curtis et al. 2011; Durka et al. 1994; Hales et al. 2007;
Johannsen et al. 2008; Kaushal et al. 2011; Lohse et al. 2013; Mayer et al. 2002; Nestler et al.
2011; Ohte 2013; Ohte et al. 2004; Ohte et al. 2010; Pellerin et al. 2012; Silva et al. 2002;
Thibodeau et al. 2013; Tobari et al. 2010; Wankel et al. 2006; Williard et al. 2001; Yue et al.
2013; Zeng and Wu 2015). By combining the two isotopic ratios, the relative mixing ratios
among various nitrate sources such as atmospheric (unprocessed), fertilizer, manure, and
sewage plants can be quantified through a simple isotope mass balance approach. Partial
removal of nitrate through either assimilation or denitrification, however, results in residual
nitrate being enriched with $^{15}N$ and $^{18}O$ (Böttcher et al. 1990; Granger et al. 2010), which
complicates their interpretation beyond that of the simple isotope mass balance approach. In
addition, trace contributions of unprocessed $NO_3^-{}_{atm}$ could have a significant impact on the
$^{18}O/^{16}O$ ratios of the total nitrate in stream water (Durka et al. 1994; Kendall 1998; Mayer et
al. 2001; Michalski et al. 2004; Tsunogai et al. 2010). Therefore, $^{18}O/^{16}O$ ratios are used as
tracers based on assumptions such as (1) the $^{18}O/^{16}O$ ratios of nitrate in stream water simply
reflect the mixing ratio of unprocessed $NO_3^-{}_{atm}$ within total nitrate (Barnes et al. 2008; Burns
et al. 2009; Campbell et al. 2006; Durka et al. 1994; Ohte et al. 2004; Ohte et al. 2010; Zeng
and Wu 2015), and (2) the mixing ratios of unprocessed $NO_3^-{}_{atm}$ within total nitrate are
minimum or uniform for whole or specific stream water samples (Johannsen et al. 2008;
Mayer et al. 2002; Wankel et al. 2006). To verify the reliability of these assumptions and to
utilize the $^{18}O/^{16}O$ ratios for the quantification of the accurate and precise mixing ratios
among various nitrate sources based on the isotope mass balance approach, the mixing ratio
of $NO_3^-{}_{atm}$ within the total nitrate in stream water must be better understood; otherwise, the
processes regulating nitrate concentration in stream water will be ambiguous even when
adding the data of $^{18}O/^{16}O$ ratios of nitrate.
To overcome the limitation in using the $^{15}N/^{14}N$ and $^{18}O/^{16}O$ ratios, the $^{17}O/^{16}O$ ratios of
nitrate have been used as an additional tracer of $NO_3^-{}_{atm}$ in stream water in recent studies
(Dejwakh et al. 2012; Michalski et al. 2004; Riha et al. 2014; Rose et al. 2015; Tsunogai et





2010; Tsunogai et al. 2014). The $^{17}O/^{16}O$ ratios were used because, whereas $NO_3^-_{re}$, the
oxygen atoms of which are derived from either terrestrial $O_2$ or $H_2O$ through nitrification,
shows mass-dependent relative variations between $^{17}O/^{16}O$ and $^{18}O/^{16}O$ ratios, unprocessed
$NO_3^-_{atm}$ displays an anomalous enrichment in $^{17}O$ from the mass-dependent relative variations,
reflecting oxygen atom transfers from ozone during the conversion of $NO_x$ to $NO_3^-_{atm}$
(Michalski et al. 2003; Morin et al. 2008). By using the $\Delta^{17}O$ signature defined by the
following equation (Kaiser et al. 2007; Miller 2002), we can distinguish $NO_3^-_{atm}$ ($\Delta^{17}O > 0$)
from $NO_3^-_{re}$ ($\Delta^{17}O = 0$):

$$\Delta^{17}O = \frac{1+\delta^{17}O}{\left(1+\delta^{18}O\right)^{\beta}} - 1, \tag{1}$$

where the constant β is 0.5247 (Kaiser et al. 2007; Miller 2002), $\delta^{18}O = R_{sample}/R_{standard} - 1$,
and $R$ is the $^{18}O/^{16}O$ ratio of the sample (or the $^{17}O/^{16}O$ ratio in the case of $\delta^{17}O$ or the $^{15}N/^{14}N$
ratio in the case of $\delta^{15}N$) and each standard reference material. In addition, $\Delta^{17}O$ is stable
during the mass-dependent isotope fractionation processes within surface ecosystems.
Therefore, although the atmospheric $\delta^{15}N$ or $\delta^{18}O$ signature can be overprinted by
biogeochemical processes subsequent to deposition, $\Delta^{17}O$ can be used as a robust tracer of
unprocessed $NO_3^-_{atm}$ to reflect the mixing ratio of unprocessed $NO_3^-_{atm}$ within total $NO_3^-$
regardless of biogeochemical removal processes subsequent to deposition by using the
following equation:

$$\frac{C_{atm}}{C_{total}} = \frac{\Delta^{17}O}{\Delta^{17}O_{atm}}, \tag{2}$$

where $C_{atm}$ and $C_{total}$ denote the concentration of $NO_3^-_{atm}$ and $NO_3^-$ in each water sample,
respectively, and $\Delta^{17}O_{atm}$ and $\Delta^{17}O$ denote the $\Delta^{17}O$ values of $NO_3^-_{atm}$ and nitrate (total) in
each water sample, respectively.
Moreover, additional measurements of the $\Delta^{17}O$ values of nitrate together with $\delta^{15}N$ and $\delta^{18}O$
enable us to exclude the contribution of $NO_3^-_{atm}$ in the determined $\delta^{15}N$ and $\delta^{18}O$ values and
to estimate the corrected $\delta^{15}N$ and $\delta^{18}O$ values ($\delta^{15}N_{re}$ and $\delta^{18}O_{re}$, respectively) for accurate
evaluation on the source and behaviour of $NO_3^-_{re}$ (Dejwakh et al. 2012; Liu et al. 2013; Riha
et al. 2014; Tsunogai et al. 2011; Tsunogai et al. 2010; Tsunogai et al. 2014).
Previous studies have successfully applied the $\Delta^{17}O$ tracer of nitrate to those eluted from
arid/semi-arid watersheds (Dejwakh et al. 2012; Michalski et al. 2004; Riha et al. 2014),



forested watersheds (Rose et al. 2015; Tsunogai et al. 2010; Tsunogai et al. 2014), and a large
river basin (Liu et al. 2013) to determine the accurate and precise mixing ratios of
unprocessed $NO_3^-{}_{atm}$ in total nitrate in addition to the fate of the $NO_3^-{}_{atm}$ that had been
deposited into each watershed. However, relative changes in the source and fate of $NO_3^-{}_{atm}$ in
accordance with the changes in land use of catchments have not been studied thus far by using
the $\Delta^{17}O$ tracer of nitrate.
In this study, we measure the concentrations and the stable isotopic compositions of nitrate
including $\Delta^{17}O$ values for more than 30 streams flowing into a lake in Japan with catchments
of widely varying land uses within the same watershed, which includes urban, suburban,
agricultural (mostly rice paddies), and forested catchments. By using the $\Delta^{17}O$ tracer, we
quantify both areal and temporal variations in the concentrations of both $NO_3^-{}_{atm}$ and $NO_3^-{}_{re}$
in streams across the land use settings to gain insight into the processes controlling the source,
transport, and fate of $NO_3^-{}_{atm}$ and $NO_3^-{}_{re}$ (Fig. 1). Although $NO_3^-{}_{re}$ increases during
nitrification within each catchment area, for instance, $NO_3^-{}_{atm}$ is stable, so that we can
evaluate the progress of nitrification within each catchment area by using the changes in the
concentrations of both $NO_3^-{}_{atm}$ and $NO_3^-{}_{re}$. Besides to the streams, we determine those values
in precipitation (wet deposition) for comparison to obtain accurate and precise mixing ratios
of both $NO_3^-{}_{atm}$ and $NO_3^-{}_{re}$ within nitrate (total) in each stream. Moreover, we exclude the
contribution of $NO_3^-{}_{atm}$ in the determined $\delta^{15}N$ and $\delta^{18}O$ values to estimate the corrected $\delta^{15}N$
and $\delta^{18}O$ values for accurate evaluation of the source and behaviour of $NO_3^-{}_{re}$. Further, we
determine those values in an outflow stream of the same lake to evaluate the influences of
flow stagnation into the lake on nitrate by using the differences between inflows and outflows
(Fig. 1). The results presented herein will increase our understanding of the fate of $NO_3^-{}_{atm}$
deposited onto land, particularly those deposited onto urban/suburban and forested
catchments (Fig. 1).
**2    Experimental Section**
**2.1    Site description**
**2.1.1 Lake Biwa watershed basin**



Lake Biwa, located in the central part of the Japanese Islands, is the largest freshwater lake in
Japan (Fig. 2). It has a surface area of 670.4 km$^2$ and a total catchment area of 3174 km$^2$ with
annual precipitation of around 2000 mm. More than 120 streams flow into the lake, whereas
the Seta River (No. 33 in Fig. 2(b)) at the southern end, also known as the Yodo River, is the
only natural outflow. The average residence time of water in the lake is 5.5 years.
Similar to many lakes throughout the world, Lake Biwa has experienced eutrophication in the
past. Urbanization near the lake beginning in the 1960s, particularly on the southern and
eastern shore, likely caused an increase in nutrient loading. Blooms of *Uroglena americana*
and cyanobacteria have occurred since 1977 and 1983, respectively (Hsieh et al. 2011). To
clarify the pathways and sources of nitrate that were fed into the lake, the stable isotopic
compositions ($\delta^{15}$N and $\delta^{18}$O) of dissolved nitrate were determined in the major streams
flowing into the lake (Ohte et al. 2010). Based on the $\delta^{15}$N values of nitrate showing positive
correlation with population densities of each catchment area, it was concluded that sewage
effluent was the dominant source contributing to the increase in the $\delta^{15}$N values of nitrate.

**2.1.2 Sado-seki monitoring station**

Sado-seki National Acid Rain Monitoring Station (38°14′59″N, 138°24′00″E) was established
on Sado Island (Fig. 2(a)), at 110 m above sea level, as a monitoring observatory of the Acid
Deposition Monitoring Network in East Asia (EANET) representing the central Japan area
(EANET 2014). The monitoring has shown that the observatory received 24.5 mmol NO$_3^-$
m$^{-2}$yr$^{-1}$ and 17.1 mmol NH$_4^+$ m$^{-2}$yr$^{-1}$ on average from FY2009 to 2011 (EANET 2014),
which corresponds to a total fixed N deposition rate of 5.8 kg of N ha$^{-1}$yr$^{-1}$.

**2.2  Sampling**

Stream water samples were collected near the mouths of each stream during base flow periods
four times in 2013 on March 15, June 17, August 5, and October 21 from 33 inflow streams
and 1 outflow river (Seta River) of Lake Biwa (Table 1; Fig. 2(b)) except for stream Nos. 3
and 28 in June, which became dry arroyos at that time. The catchments of the studied inflow
streams occupied 70% of the entire Lake Biwa basin area. The streams were selected to cover
those in which the concentrations and stable isotope compositions of nitrate, $\delta^{15}$N and $\delta^{18}$O,
had already been determined in 2004–2006 (Ohte et al. 2010). The categories of locations
classified by Ohte et al. (2010) were also used in this study to classify the location of the





streams (Table 1). Either a bucket or dipper was used to collect samples as far from the bank
as possible. Each sample was transferred into a dark polyethylene bottle that was pre-rinsed at
least twice with the sample itself and stored under refrigeration. Then, the samples were
filtered through a pre-combusted Whatman GF/F filter with a 0.7 μm pore size within a few
hours after collection, and the filtrate was stored in a different dark polyethylene bottle under
refrigeration at 4°C until analysis.
To calculate the annual influx/efflux of nitrate via each stream to/from Lake Biwa as well as
its seasonal variation, we used the sampling number n, where n = 1, 2, 3, and 4, to represent
the sampling in March, June, August, and October, respectively. In addition, we used one
more hypothetical sampling number (n = 5) set just one year later than the n = 1 date to
quantify the annual influx/efflux. Accordingly, we assumed that secular change was minimum
in the streams and that the streams returned back to the initial state one year later.
Furthermore, we rated the interval between n = 1 and 2 as spring, n = 2 and 3 as summer, n =
3 and 4 as autumn, and n = 4 and 5 as winter for the streams in this study.
Samples of wet deposition were taken at the Sado-seki National Acid Rain Monitoring Station
by using standard methods for evaluating acid deposition in Japan from April 2009 to March
2012. An automatic wet deposition sampler (US-420, Ogasawara) was used in the collection.
All of the deposition samples were introduced and stored in 1 L polyethylene bottles under
refrigeration until daily recovery. After measuring both the conductivity and pH, the
recovered samples were filtered through a 0.2 μm pore-size membrane filter (Dismic-25CS,
ADVANTEC) and stored in a refrigerator until analysis. The annual wet deposition rate of
nitrate was 19.3 mmol $m^{-2}y^{-1}$ for FY2009, from April 2009 to March 2010; 28.0 mmol $m^{-2}y^{-1}$
for FY2010, from April 2010 to March 2011; and 27.0 mmol $m^{-2}y^{-1}$ for FY2011, from April
2011 to March 2012 (EANET 2014).
**2.3   Analysis**
The concentrations of nitrate ($NO_3^-$) and nitrite ($NO_2^-$) in each filtrate sample were measured
by ion chromatography (Prominence HIC-SP, Shimadzu, Japan) within a few days after each
sampling. The $\delta^{18}O$ values of $H_2O$ in the samples were analysed by using the cavity ring-
down spectroscopy method by employing the Picarro L2120-I instrument equipped with an
A0211 vaporizer and auto sampler; the error in this method was ±0.1‰. Both Vienna



Standard Mean Ocean Water (VSMOW) and Standard Light Antarctic Precipitation (SLAP)
were used to calibrate the values to the international scale.
To determine the stable isotopic compositions, nitrate in each filtrate sample was chemically
converted to $N_2O$ by using a method originally developed to determine the $^{15}N/^{14}N$ and
$^{18}O/^{16}O$ ratios of seawater and freshwater nitrate (Mcilvin and Altabet 2005) which was later
modified (Konno et al. 2010; Nakagawa et al. 2013; Tsunogai et al. 2011; Tsunogai et al.
2008; Tsunogai et al. 2010; Yamazaki et al. 2011). Then, the stable isotopic compositions of
$N_2O$ were determined by using a continuous-flow isotope ratio mass spectrometry (CF-IRMS)
system in Nagoya University (Hirota et al. 2010; Komatsu et al. 2008). The analytical
procedures are the same as those detailed in previous research (Nakagawa et al. 2013;
Tsunogai et al. 2014).
To determine whether samples were deteriorated or contaminated during storage and whether
the conversion rate from nitrate to $N_2O$ was sufficient, concentrations of nitrate in the samples
were determined each time we analysed isotopic compositions using MS based on the $N_2O^+$
or $O_2^+$ outputs. We adopted the $\delta^{15}N$, $\delta^{18}O$, or $\Delta^{17}O$ values only when concentrations
measured by MS correlated with those measured by ion chromatography just after the
sampling within 10% differences.
We repeated the analyses on the $\delta^{15}N$, $\delta^{18}O$, and $\Delta^{17}O$ values of nitrate at least three times for
each sample to attain high precision. Most of the samples had nitrate concentrations of more
than 5.0 µmol $L^{-1}$, which corresponded to nitrate quantities greater than 50 nmol in a 10 mL
sample. This amount was sufficient for determining the $\delta^{15}N$, $\delta^{18}O$, and $\Delta^{17}O$ values with high
precision. The sample volume was increased to 30 mL for those having nitrate concentrations
less than 5.0 µmol $L^{-1}$; the number of analyses was also increased in such cases. Thus, all
isotopic data presented in this study have an error better than ±0.2‰ for $\delta^{15}N$, ±0.3‰ for $\delta^{18}O$,
and ±0.1‰ for $\Delta^{17}O$.
Because the more precise power law shown in Eq. (1) was used to calculate $\Delta^{17}O$, the
estimated $\Delta^{17}O$ values were somewhat different from those estimated based on traditional
linear approximation (Michalski et al. 2002). Please note that our $\Delta^{17}O$ values of $NO_3^-{}_{atm}$
would be 0.9±0.1 ‰ higher if we use the linear approximation. However, these differences
were insignificant for most of the stream water samples evaluated in this study.
Nitrite ($NO_2^-$) in the samples interferes with the final $N_2O$ produced from nitrate ($NO_3^-$)
because the chemical method also converts $NO_2^-$ to $N_2O$ (Mcilvin and Altabet 2005).





Therefore, it is sometimes necessary to correct for the contribution of $NO_2^-$-derived $N_2O$ to
accurately determine the stable isotopic compositions of the sample nitrate. However, all
samples analysed for stable isotopic compositions in this study showed $NO_2^-/NO_3^-$ ratios of
less than 5%; thus, the results were used with no corrections.
**2.4  Calculating average concentration and isotopic compositions in each**

6       **stream**

To quantitatively clarify the chemical and isotopic characteristics of each stream, we
determined both the flow-weighted annual average concentration ($\overline{C}_{total}$) and flow-weighted
annual average $\delta^{15}N$, $\delta^{18}O$, and $\Delta^{17}O$ values ($\overline{\delta}$) of nitrate for each stream by using Eqs. (3),
(4), and (5):
$$q = \sum_{n=1}^{4}\left(f_n \cdot \Delta t_n\right), \tag{3}$$
$$\overline{C}_{total} = \frac{\sum_{n=1}^{4}\left(C_n \cdot f_n \cdot \Delta t_n\right)}{q}, \tag{4}$$
$$\overline{\delta} = \frac{\sum_{n=1}^{4}\left(\delta_n \cdot C_n \cdot f_n \cdot \Delta t_n\right)}{\sum_{n=1}^{4}\left(C_n \cdot f_n \cdot \Delta t_n\right)}, \tag{5}$$
where $C_n$ and $\delta_n$ denote the concentration ($C_{total}$ in Eq. (2)) and isotopic values ($\delta^{15}N$, $\delta^{18}O$, or
$\Delta^{17}O$) of nitrate in each stream during each observation $n$, respectively; $f_n$ denotes the flow
rate of each stream during each observation $n$; and $\Delta t_n$ denotes the time interval between the
observation $n$ and the next observation $n+1$. When possible, we used the flow rate of each
stream which was determined monthly by the Shiga Prefecture (Shiga_Prefecture 2015) for $f_n$.
For small streams with no data for the flow rate, we used a small and stable flow rate of 0.1
$m^3$/s for $f_n$.
**2.5  Calculating $\delta^{15}N$ and $\delta^{18}O$ of remineralized nitrate**
To exclude the contribution of $NO_3^-{}_{atm}$ from the $\delta^{15}N$ and $\delta^{18}O$ values of nitrate and to clarify
the source and behaviour of $NO_3^-{}_{re}$ by using both $\delta^{15}N$ and $\delta^{18}O$ as tracers, we estimated the
end-member $\delta^{15}N$ and $\delta^{18}O$ values of the remineralized nitrate portion, $\delta^{15}N_{re}$ and $\delta^{18}O_{re}$, by





excluding the contribution of $NO_3^-{}_{atm}$ in each nitrate (Dejwakh et al. 2012; Liu et al. 2013;
Riha et al. 2014; Tsunogai et al. 2011; Tsunogai et al. 2010; Tsunogai et al. 2014) by using
Eqs. (6) and (7):
$$\delta^{15}N_{re} = \frac{C_{total} \cdot \delta^{15}N - C_{atm} \cdot \delta^{15}N_{atm}}{C_{total} - C_{atm}},$$    (6)
$$\delta^{18}O_{re} = \frac{C_{total} \cdot \delta^{18}O - C_{atm} \cdot \delta^{18}O_{atm}}{C_{total} - C_{atm}},$$    (7)
where $C_{atm}$ and $C_{total}$ denote the concentration of $NO_3^-{}_{atm}$ and nitrate in each water sample,
respectively, and $\delta^{15}N_{atm}$, $\delta^{18}O_{atm}$, and $\Delta^{17}O_{atm}$ denote the $\delta^{15}N$, $\delta^{18}O$, and $\Delta^{17}O$ values of
$NO_3^-{}_{atm}$ in each sample, respectively. The actual values of $\delta^{15}N_{atm}$, $\delta^{18}O_{atm}$, and $\Delta^{17}O_{atm}$ used
in this study will be determined in section 3.1.

## 11    3    Results and Discussion

### 12    3.1    Atmospheric Nitrate

The triple oxygen isotopic compositions ($\Delta^{17}O$) of atmospheric nitrate ($NO_3^-{}_{atm}$) are plotted in
Fig. 3(c) as a function of the sampling day (local time, UT +9:00), together with the $\delta^{15}N$ and
$\delta^{18}O$ of $NO_3^-{}_{atm}$ in Figs. 3(a) and 3(b). The atmospheric nitrate at Sado-seki monitoring
station showed large $^{17}O$ anomalies with $\Delta^{17}O$ values from +18.6‰ to +32.4‰.
Moreover, a clear normal correlation with $\delta^{18}O$ was shown (Supplement Fig. S1). A similar
trend has been reported in atmospheric nitrate aerosols collected for a one-year period in La
Jolla, California (32.7°N, 117.2°W) (Michalski et al. 2003), and in other areas worldwide
(Kaiser et al. 2007; Morin et al. 2009). Michalski et al. (2003) interpreted that the linear
correlation corresponds to the mixing line between the tropospheric ozone and tropospheric
$H_2O$, and thus tropospheric OH radicals, with $\Delta^{17}O = 0$‰ and $\delta^{18}O = -5$‰. The $NO_3^-{}_{atm}$ data
obtained at Sado-seki monitoring station, however, showed a somewhat different trend in the
$\Delta^{17}O$–$\delta^{18}O$ plot between summer, from May to October, and winter, from November to April
(Supplement Fig. S1). Although the linearly fitted line to the summer data showed a slope of
$2.21 \pm 0.22$ and an intercept of +19.7 ± 5.1‰ in the $\Delta^{17}O$–$\delta^{18}O$ plot, that to the winter data
showed a statistically significant larger slope of $2.89 \pm 0.38$ and the smaller intercept of +3.0
$\pm 9.2$‰; all errors were in the 2 σ range. Although the winter data included an intercept of



−5‰ as the end member $\delta^{18}O$ value of the tropospheric OH radical within the possible error
range, as reported by Michalski et al. (2003), the intercept of summer data deviated strongly
from the value. Because the monitoring station is located in the Asian monsoon area, the
major air mass that arrived at the station was seasonally different; Pacific air originated from
the south-eastern direction in summer whereas continental air originated from north-western
direction in winter. The present results imply seasonal and regional changes in the $\delta^{18}O/\Delta^{17}O$
ratios of tropospheric ozone and in the OH radical.
On the basis of both the temporal variation in the depositional flux of $NO_3^-{}_{atm}$ and the $\Delta^{17}O$
value, we estimated the average $\Delta^{17}O$ value of $NO_3^-{}_{atm}$ ($\Delta^{17}O_{avg}$) deposited at Sado-seki
monitoring station as +25.5‰ for FY2009, +27.2‰ for FY2010, +25.7‰ for FY2011, and
+26.3‰ for the three years by using

$$\Delta^{17}O_{avg} = \frac{\sum_k \left( C_k \cdot V_k \cdot \Delta^{17}O_k \right)}{\sum_k \left( C_k \cdot V_k \right)},\tag{8}$$

where $C_k$ denotes the concentration of nitrate in each wet deposition sample, and $V_k$ denotes
the total water volume of each wet deposition sample. Substituting $\Delta^{17}O$ with $\delta^{15}N$ ($\delta^{18}O$) in
Eq. (8), we estimated $\delta^{15}N_{avg}$ ($\delta^{18}O_{avg}$) as −4.4 ‰ (+78.5‰) for FY2009, −3.8 ‰ (+81.8‰)
for FY2010, −4.4 ‰ (+78.6‰) for FY2011, and −4.2‰ (+79.8‰) for the three years.
To apply the $\Delta^{17}O_{avg}$ values obtained at Sado-seki monitoring station as for $\Delta^{17}O$ of $NO_3^-{}_{atm}$
deposited into the studied watershed (i.e. $\Delta^{17}O_{atm}$ in Eq. (2)), however, additional corrections
could be needed because the $\Delta^{17}O$ value of $NO_3^-{}_{atm}$ is a function of the $NO_x$ oxidation
channels in the atmosphere that shift depending on the intensity of sunlight, temperature, and
oxidant levels (e.g. Alexander et al. 2009; Kunasek et al. 2008; Michalski et al. 2003; Morin
et al. 2012; Morin et al. 2008; Savarino et al. 2013). The latitudinal difference between Sado-
seki monitoring station (38°15′N, 138°24′E; Fig. 2) and the watershed studied (35°15′N,
136°5′E; Fig. 2) could change the intensity of sunlight and thus the $NO_x$ oxidation channel.
Moreover, Tsunogai et al. (2010) reported that nitrate in polluted air masses derived directly
from megacities in winter showed slightly larger $\Delta^{17}O$ values than those in the same seasons
owing likely to the relative increase in the reaction via $NO_3$ radicals within the entire $NO_3^-{}_{atm}$
production channel to produce $NO_3^-{}_{atm}$ in the polluted air mass. The annual average $\Delta^{17}O$
values determined in this study was the lowest value in FY2009 when the deposition rate was
the smallest, at 19.3 mmol m$^{-2}$y$^{-1}$ (EANET 2014), whereas the annual average $\Delta^{17}O$ value



was the highest value in FY2010 when the deposition rate was the largest, at 28.0 mmol
$m^{-2}y^{-1}$ for nitrate, within the three years of observation. These results also imply that we must
correct for the difference in arrival frequency of polluted air mass as well.
Nevertheless, both the annual average and the seasonal variation range of $\Delta^{17}O$ correlated
strongly with those determined at Rishiri monitoring station (45°07′11″N, 141°12′33″E) in
FY2008, at +26.2‰ (Tsunogai et al. 2010), where the wet deposition rate of $NO_3^-{}_{atm}$ was an
average 40% smaller that at Sado-seki monitoring station from 2000 to 2013 (EANET 2014).
Moreover, the values also coincided with those reported for mid-latitudes, such as at La Jolla,
at 33°N (Michalski et al. 2003) and Princeton, at 40°N (Kaiser et al. 2007). We concluded that
by allowing an appropriate range of errors presented later, the obtained $\Delta^{17}O_{avg}$ value of
$NO_3^-{}_{atm}$ can represent those deposited at middle latitudes worldwide, including the Lake Biwa
watershed basin.
In addition, the actual $\Delta^{17}O_{atm}$ values of $NO_3^-{}_{atm}$ in each stream water sample can differ from
the $\Delta^{17}O_{avg}$ owing to the seasonal variation in the $\Delta^{17}O$ values of $NO_3^-{}_{atm}$. In correcting for the
seasonal variation, however, it is not adequate to use the $\Delta^{17}O$ values determined for the
seasons of sampling, as $\Delta^{17}O_{atm}$ in Eq. (2), because the duration of the recession period is
longer than a few months for most of the forest catchments in Japan with a humid temperate
climate (Ohte et al. 2010; Takimoto et al. 1994). That is, nitrate in base flow stream water had
been stored in subsurface runoff and groundwater, of which seasonal $\Delta^{17}O$ changes have not
been found thus far (Nakagawa et al. 2013; Tsunogai et al. 2010).
In summary, we used the obtained $\Delta^{17}O_{avg}$ value of $NO_3^-{}_{atm}$ as $\Delta^{17}O_{atm}$ in Eq. (2) to estimate
$C_{atm}$ in the streams of the Lake Biwa watershed basin by allowing the error range of 3.0‰,
considering the whole factor change of $\Delta^{17}O_{atm}$ from $\Delta^{17}O_{avg}$. About 65% of the all $\Delta^{17}O$ data
of $NO_3^-{}_{atm}$ obtained at Sado-seki monitoring station were included in this range of +26.3 ±
3.0‰.
In the case of $\delta^{15}N$ and $\delta^{18}O$, the values of $NO_3^-{}_{atm}$ in each stream water sample (i.e. $\delta^{15}N_{atm}$
and $\delta^{18}O_{atm}$ in Eqs. (6) and (7)) differed further from $\delta^{15}N_{avg}$ and $\delta^{18}O_{avg}$ owing to isotopic
fractionation during partial removal subsequent to deposition. As a result, while using the
$\delta^{15}N_{avg}$ and $\delta^{18}O_{avg}$ values as $\delta^{15}N_{atm}$ and $\delta^{18}O_{atm}$, we assumed much larger error range on the
values; i.e. ± 10‰ for both $\delta^{15}N$ and $\delta^{18}O$. We will further discuss the appropriateness of
these error ranges section 3.3. Because of the small $C_{atm}/C_{total}$ ratios of stream water at





generally less than 7% (section 3.2), the error propagated to $\delta^{15}N_{re}$ and $\delta^{18}O_{re}$ was generally
small, less than 1‰ and 2‰, respectively, for most of the data presented in this study.
**3.2   Stream nitrate overview**
The concentrations ($C_{total}$) and $\delta^{15}N$, $\delta^{18}O$, and $\Delta^{17}O$ values of nitrate in the stream water
samples determined for each observation (n = 1 2, 3, and 4) are presented in Fig. 4. The
annual average concentration ($\overline{C}_{total}$) and annual average $\delta^{15}N$, $\delta^{18}O$, and $\Delta^{17}O$ values ($\overline{\delta^{15}N}$,
$\overline{\delta^{18}O}$, and $\overline{\Delta^{17}O}$, respectively) in each stream estimated by using Eqs. (3), (4), and (5) are
shown in the figure as black bars. In this figure, each stream was plotted on the x-axis in the
order of location beginning from stream No. 31, which lies southwest of all of the streams
(Fig. 2), and proceeding in a clockwise direction. The spatially continuous variation in the
values of $\overline{\delta^{15}N}$, $\overline{\delta^{18}O}$, and $\overline{\Delta^{17}O}$ imply that the values may represent land use changes in each
catchment area.
Although the $\Delta^{17}O$ values presented significant areal and temporal variation from +0.0‰ to
+6.8‰, the range of the $\overline{\Delta^{17}O}$ values from +0.5‰ to +3.1‰ was typical for nitrate in natural
stream water (Liu et al. 2013; Michalski et al. 2004; Rose et al. 2015; Tsunogai et al. 2010;
Tsunogai et al. 2014). These results correspond to the mixing ratios of unprocessed $NO_3{}^-_{atm}$ to
total nitrate from 1.8 ± 0.3% to 11.8 ± 1.3%, obtained by using Eq. (2).
By using the concentration ($C_{total}$) and $\delta^{15}N$, $\delta^{18}O$, and $\Delta^{17}O$ values of nitrate, the $NO_3{}^-_{atm}$
concentration ($C_{atm}$) and the $\delta^{15}N$ and $\delta^{18}O$ values of the remineralized portion of nitrate
($\delta^{15}N_{re}$ and $\delta^{18}O_{re}$) in the samples were calculated by using Eqs. (2), (6), and (7) and are
plotted in Fig. 5. In addition, the annual average concentration of $NO_3{}^-_{atm}$ ($\overline{C}_{atm}$) and annual
average values of $\delta^{15}N_{re}$ and $\delta^{18}O_{re}$ ($\overline{\delta^{15}N_{re}}$ and $\overline{\delta^{18}O_{re}}$, respectively) in stream nitrate were
calculated and are presented in Fig. 5 as black bars. Because of the large $\delta^{18}O$ differences of
about 80‰ between nitrate in streams and $NO_3{}^-_{atm}$, the $\delta^{18}O_{re}$ values were a few ‰ lower
than each original $\delta^{18}O$ value in total nitrate. On the contrary, owing to the small $\delta^{15}N$
differences of less than 15‰ between the total nitrate in streams and $NO_3{}^-_{atm}$, as well as the
small $C_{atm}/C_{total}$ ratios in the streams, most of the $\delta^{15}N_{re}$ values showed small deviations of
less than 1‰ from each original $\delta^{15}N$ value in the total nitrate in most of the streams.
To verify possible secular changes, the estimated $\overline{C}_{total,}$ $\overline{\delta^{15}N}$, and $\overline{\delta^{18}O}$ for each stream were
compared with those determined by Ohte et al. (2010) in which annual average concentration





and annual average $\delta^{15}N$ and $\delta^{18}O$ values of nitrate (total) were determined for the same
streams in 2004 to 2006 (Supplement Fig. S2). Although both concentrations and $\delta^{15}N$ and
$\delta^{18}O$ values in the streams showed significant areal and temporal variations during 2013, as
presented in Fig. 4, the annual average values almost correlated with the values determined in
2004 to 2006. We concluded that secular changes were minimal for nitrate in the streams, at
least for the most recent 10-year period of observations.

### 7    3.3    Relationship between $\Delta^{17}O$ and $\delta^{18}O$

One of the features in the areal variation shown in Fig. 4 is the positive correlation between
$\Delta^{17}O$ and $\delta^{18}O$. As clearly presented in the relationship between $\overline{\Delta^{17}O}$ and $\overline{\delta^{18}O}$ (Fig. 6), these
values showed linear correlation with the $r^2$ value of 0.88. Because $NO_3{}^-_{atm}$ is characterized
by highly elevated values of both $\Delta^{17}O$ and $\delta^{18}O$ (Fig. 3), changes in the mixing ratio of
unprocessed $NO_3{}^-_{atm}$ within the total nitrate pool must be strongly responsible for the positive
correlation between $\overline{\Delta^{17}O}$ and $\overline{\delta^{18}O}$ for nitrate in the streams.
The slope value of the least–squares-fitted line between $\overline{\Delta^{17}O}$ and $\overline{\delta^{18}O}$ (Fig. 6) also supports
this hypothesis. By extrapolating the least–square-fitted line to the region of $NO_3{}^-_{atm}$ having a
$\Delta^{17}O$ value of +26.3‰, we obtained $\delta^{18}O = +86 \pm 7$‰, which also corresponds with the
average $\delta^{18}O$ value of $NO_3{}^-_{atm}$ of +79.8‰ obtained in section 3.1. We concluded that $\overline{\delta^{18}O}$
values primarily reflect the mixing ratio of $NO_3{}^-_{atm}$ within nitrate as well.
By extrapolating the linear correlation between $\overline{\Delta^{17}O}$ and $\overline{\delta^{18}O}$ to $\overline{\Delta^{17}O} = 0$‰, we obtained the
$\delta^{18}O$ value of $-2.9 \pm 1.2$‰ as the average $\delta^{18}O$ value of the remineralized portion of nitrate
($NO_3{}^-_{re}$) in the streams. Although the $\delta^{18}O$ value was substantially $^{18}O$-depleted compared
with that obtained through in vitro incubation experiments in past studies (Burns and Kendall
2002; Mayer et al. 2001; Spoelstra et al. 2007), it correlated strongly with the $\delta^{18}O$ value of
$NO_3{}^-_{re}$ determined recently by using the linear relationship between $\Delta^{17}O$ and $\delta^{18}O$. This
correlation occurred in the groundwater of cool-temperate forested watersheds at $-4.2 \pm 2.4$‰,
where the $\delta^{18}O(H_2O)$ was around $-13$‰ (Tsunogai et al. 2010) and in stream water in a cool-
temperate forested watershed at $-3.6 \pm 0.7$‰, where the $\delta^{18}O(H_2O)$ was around $-11$‰
(Tsunogai et al. 2014). Moreover, the $\delta^{18}O$ value of $NO_3{}^-_{re}$ obtained in this study, $-2.9 \pm$
1.2‰, is close to the possible lowermost $\delta^{18}O$ value of $NO_3{}^-_{re}$ produced through nitrification
under $H_2O$ of $-7.8 \pm 1.0$‰ (Buchwald et al. 2012). Furthermore, the $\delta^{18}O$ value of $NO_3{}^-_{re}$





correlates strongly with that obtained through in vitro incubation experiments in recent studies
that simulated temperate forest soils (Fang et al. 2012). We concluded that the $\delta^{18}O$ value of
$NO_3^-{}_{re}$ produced through nitrification in the temperate watershed having $\delta^{18}O(H_2O)$ values of
$-7.8 \pm 1.0‰$ was $-2.9 \pm 1.2‰$ and that we should use such a low $\delta^{18}O$ value as for that
produced through nitrification in the watershed. Understanding the relationship between $\Delta^{17}O$
and $\delta^{18}O$ of nitrate shown in Fig. 6 is highly useful for determining the $\delta^{18}O$ value of $NO_3^-{}_{re}$
in each watershed (Tsunogai et al. 2010).
Although the $\Delta^{17}O$ values of nitrate were stable during the biogeochemical processing such as
partial removal through assimilation or denitrification, the $\delta^{18}O$ values of nitrate could vary
through the isotopic fractionation processes within each catchment area. Nevertheless, the
$\overline{\delta^{18}O}$ values of nitrate in the streams were plotted on the mixing line between the $NO_3^-{}_{atm}$ that
had been deposited into the watershed and $NO_3^-{}_{re}$ having $\delta^{18}O$ and $\Delta^{17}O$ values close to those
produced through nitrification in the catchments. We concluded that the range of isotopic
fractionations owing to partial removal through assimilation or denitrification subsequent to
deposition of $NO_3^-{}_{atm}$ or production of $NO_3^-{}_{re}$ within each catchment area was generally small
for the major portion of nitrate eluted from the watershed. This result also supports our
assumption in section 3.1 such that the actual $\delta^{15}N$ and $\delta^{18}O$ values of $NO_3^-{}_{atm}$ in each stream
water sample ($\delta^{15}N_{atm}$ and $\delta^{18}O_{atm}$ in Eqs. (6) and (7)) correlate with the $\delta^{15}N_{avg}$ and $\delta^{18}O_{avg}$
estimated at Sado-seki monitoring station within an error of $\pm10‰$.
**3.4   $\delta^{15}N$ values of remineralized nitrate in streams**
To trace the source of the $^{18}O$-depleted $NO_3^-{}_{re}$ eluted from the watershed into the lake, the
annual average $\delta^{15}N$ values of the remineralized portion of nitrate ($\overline{\delta^{15}N_{re}}$) in each inflow
stream were estimated and are plotted as a function of population density in Fig. 7(c), together
with $\overline{\delta^{18}O_{re}}$ and $\overline{\delta^{18}O}$ in Fig. 7(d). Although the annual average $\overline{\delta^{18}O_{re}}$ values were low and
almost uniform from $-4.0‰$ to $-0.1‰$, as implied in the linear correlation between $\overline{\Delta^{17}O}$ and
$\overline{\delta^{18}O}$ in Fig. 6, $\overline{\delta^{15}N_{re}}$ showed larger variation from $+1.7‰$ to $+10.9‰$. Moreover, $\overline{\delta^{15}N_{re}}$
showed positive correlation with the population density (Fig. 7). A similar trend was reported
for the $\delta^{15}N$ values of total nitrate in past studies in this watershed (Ohte et al. 2010) and
others (Mayer et al. 2002). We further verified that the remineralized portion of nitrate
($NO_3^-{}_{re}$) was responsible for the positive correlation between the $\delta^{15}N$ values of total nitrate
and population density.





Both the concentrations and the isotopic compositions shown in Fig. 7 clearly demonstrate
that most portions of the nitrate eluted from the catchments with lower population densities of
less than 100 km$^{-2}$ were produced through nitrification in naturally occurring soil organic
matter (Kendall et al. 1995; Ohte et al. 2010), showing $\delta^{15}N$ values of +4.4 ± 1.8‰ and $\delta^{18}O$
values of about −2.3 ± 0.9‰. In this section, we discuss the source of the $^{15}N$-enriched $NO_3^-{}_{re}$
eluted from the catchments with higher population densities of more than 1000 km$^{-2}$, showing
$\delta^{15}N$ values of +9.2 ± 1.3‰ or more and $\delta^{18}O$ values of about −2.2 ± 1.1‰.
Denitrification in riverbed sediments adjacent to riparian zones or groundwater bodies
(Mcmahon and Böhlke 1996) can increase the $\delta^{15}N$ value of stream nitrate. However, if such
post-production alternation were responsible for the $^{15}N$ enrichment of $NO_3^-{}_{re}$ and thus the
total nitrate, the values of $\delta^{18}O_{re}$ in addition to those of $\delta^{15}N_{re}$ should increase (Granger et al.
2008). Moreover, the absolute concentration of $NO_3^-{}_{atm}$ ($C_{atm}$) should decrease in accordance
with the progress of denitrification. The low and uniform $\delta^{18}O_{re}$ values (Fig. 7(d)) as well as
the uniform $C_{atm}$ irrespective of the population densities (Fig. 7(b)) imply that the
denitrification in riverbed sediments was minor for nitrate in the streams. Rather, the $NO_3^-{}_{re}$
must be enriched in $^{15}N$ from its initial production through nitrification within the catchments
with high population densities. In addition, the small differences in $\delta^{18}O$ values of $NO_3^-{}_{re}$
between those values irrespective of the population densities in the catchment area (Fig. 7)
imply that the essential parameters for determining the $\delta^{18}O$ values of nitrate during
nitrification, such as the $\delta^{18}O$ values of $H_2O$ and pH of soils (Buchwald et al. 2012; Fang et al.
2012), should be similar between them.
Based on the $\delta^{15}N$ values of total nitrate eluted from catchments with high population
densities, as well as the positive correlation between the $\delta^{15}N$ values of total nitrate and
population densities, Ohte et al. (2010) proposed sewage effluent as the dominant source
contributing to the increase in the $\delta^{15}N$ values of total nitrate eluted from such catchments.
The $\delta^{15}N$ and $\delta^{18}O$ values of $NO_3^-{}_{re}$ newly estimated in this study, +9.2 ± 1.3‰ or more and
−2.2 ± 1.1‰, respectively, also imply that the dominant source contributing to the increase in
the $\delta^{15}N$ values of total nitrate had been produced through nitrification in which the source N
of the nitrate had already been enriched in $^{15}N$. Although the $\delta^{15}N$ and $\delta^{18}O$ values of $NO_3^-{}_{re}$
eluted from the high population density catchments, $\delta^{15}N_{re}$ = +9.2 ± 1.3‰ and $\delta^{18}O_{re}$ = −2.2 ±
1.1‰, were a few ‰ lower than the $\delta^{15}N$ and $\delta^{18}O$ values of total nitrate in the sewage
effluent determined in past studies (Aravena et al. 1993; Wankel et al. 2006; Widory et al.





2005; Xue et al. 2009), the slight deviations in the reported $\delta^{15}N$ and $\delta^{18}O$ values from our
results can be explained by the following factors: (1) a slight contribution of $NO_3^-{}_{atm}$, and (2)
the progress of denitrification subsequent to production. We concluded that sewage effluent
was the most probable pollution source of nitrate to explain the observed concentrations and
isotopic compositions of nitrate eluted from the catchments with high population densities,
particularly for those more than 1000 km$^{-2}$.

### 3.5   Seasonal variation

Although the annual average values of $\Delta^{17}O$ and $\delta^{18}O$ in each river, $\overline{\Delta^{17}O}$ and $\overline{\delta^{18}O}$,
respectively, showed linear correlation as presented in Fig. 6, the same results were not
always attained for those in each season. Particularly for those obtained during June and
August (i.e. summer), some of the streams showed significant deviations in $\delta^{18}O$ of more than
5‰ from the hypothetical mixing line between $NO_3^-{}_{atm}$ ($\Delta^{17}O$ = +26.3 ‰ and $\delta^{18}O$ = 79.8 ‰)
and $NO_3^-{}_{re}$ ($\Delta^{17}O$ = 0 ‰ and $\delta^{18}O$ = −2.9 ‰; Fig. 6). Even though the values of $\Delta^{17}O$ and
$\delta^{18}O$ of $NO_3^-{}_{atm}$ showed seasonal variation, as presented in Fig. 3, it could not explain the
large deviations from the mixing line based on the seasonal changes in $NO_3^-{}_{atm}$. Rather, we
must assume some seasonal changes in the biogeochemical nitrogen cycles within each
catchment area to explain the relationship because different from that of the $\Delta^{17}O$ values, the
$\delta^{18}O$ values of nitrate can vary during biogeochemical processing within each catchment area.
As a result, we can evaluate the seasonal changes in the biogeochemical processing within
each catchment area by using the seasonal changes in the relationship between $\Delta^{17}O$ and $\delta^{18}O$
shown in Fig. 8.
The increases in the number of data plotted on the highly $^{18}O$-enriched region of more than a
few ‰ in $\delta^{18}O$ from the lines imply that partial nitrate removal through assimilation or
denitrification was active within each catchment area in June and August. The areal
differences in the $^{18}O$ enrichment also support this hypothesis. As presented in Fig. 8, $^{18}O$
enrichments were common in samples obtained at the southern streams having high
population densities in each catchment area, as shown in the figure by white squares. We can
anticipate elevated loading of both nutrients and organic matter of anthropogenic origin in
these catchments, both of which naturally enhance both assimilation and denitrification.
On the contrary, most samples obtained during March and October were distributed on the
hypothetical mixing line between $NO_3^-{}_{atm}$ and $NO_3^-{}_{re}$ as presented in Fig. 8. We concluded





that in winter, the range of isotopic fractionation subsequent to production, such as partial
removal through assimilation or denitrification, was generally small for the major portion of
nitrate eluted from the watershed and fed into the lake. Therefore, the annual average values
(i.e. $\overline{\delta^{18}O}$ and $\overline{\Delta^{17}O}$) of streams were distributed on the hypothetical mixing line, as shown in
Fig. 6, because the nitrate influx in winter occupied a major portion of the annual nitrate
influx. Active removal of nitrate from the streams through denitrification/assimilation in
summer was also responsible for the small relative importance of nitrate influx into the lake in
summer. In conclusion, the relationship between $\Delta^{17}O$ and $\delta^{18}O$ of nitrate eluted from a
catchment area is a useful indicator for evaluating the biogeochemical processing within the
catchment area, including the seasonal change.

## 3.6   Areal and temporal $\Delta^{17}O$ variation

By using the $\delta^{18}O$ values of nitrate as tracers, Ohte et al. (2010) found that the mixing ratios
of unprocessed $NO_3^-{}_{atm}$ within the total nitrate pool were high in the northern streams of the
watershed in winter, from November to late April. Our present results further verified the past
results shown in Fig. 4, which adds more robust evidence through the use of the $\Delta^{17}O$ tracer
for $NO_3^-{}_{atm}$.
Based on the high accumulation rate of snow in the catchment zones of the northern streams,
Ohte et al. (2010) concluded that high loading of unprocessed $NO_3^-{}_{atm}$ via snow in the
catchment zones increased the stored unprocessed $NO_3^-{}_{atm}$ in the snowpack, which was
subsequently released into the streams during the melting seasons. This process enhanced the
mixing ratio of unprocessed $NO_3^-{}_{atm}$ within the total nitrate pool during the melting season,
which was also reported for streams worldwide (Kendall et al. 1995; Ohte et al. 2004; Ohte et
al. 2010; Pellerin et al. 2012; Piatek et al. 2005; Tsunogai et al. 2014). However, the
contribution of nitrate from anthropogenic sources could be smaller in this area owing to
lower population densities in the catchments (Table 1). Because a major portion of possible
anthropogenic nitrate in the catchments must be occupied by $NO_3^-{}_{re}$ (Ohte et al. 2010), a
lower $NO_3^-{}_{re}$ supply from anthropogenic sources in each catchment area could elevate the
mixing ratio of unprocessed $NO_3^-{}_{atm}$ within the total nitrate pool even if absolute
concentration of $NO_3^-{}_{atm}$ ($C_{atm}$) was uniform in the streams.
To determine the $C_{atm}$ variability among the streams, the $C_{atm}$ values estimated in this study
were plotted as a function of population densities in Fig. 7(b). The $C_{atm}$ was almost uniform at



2.3 ± 1.1 µmol L$^{-1}$ irrespective of changes in the population density of each catchment area.
However, a clear $C_{total}$ enrichment trend was noted in accordance with the increase in
population densities of the catchments (Fig. 7(a)). Similar $C_{total}$ enrichment trends have been
reported in previous studies (Ohte et al. 2010).
The northern streams such as Nos. 3, 4, and 5 were enriched in $C_{atm}$, showing $C_{atm}$ annual
average values of 5.3 ± 0.9, 2.9 ± 0.6, and 4.3 ± 0.8 µmol L$^{-1}$, respectively, and $\Delta^{17}O$ values
of +3.1‰, +1.9‰, and +2.9‰, respectively. These results support the previous observation of
the streams determined by using the $\delta^{18}O$ tracer. Similar $C_{atm}$ enrichment of about 3 µmol L$^{-1}$
or more, however, was also found in streams in other areas, such as Nos. 14 ($C_{atm}$ = 3.3 ± 0.9
µmol L$^{-1}$), 25 (3.2 ± 0.7 µmol L$^{-1}$), and 21 (4.2 ± 0.8 µmol L$^{-1}$), while showing lower $\Delta^{17}O$
values of +0.9‰, +1.5‰, and +2.0‰, respectively, thus showing low mixing ratios of
unprocessed $NO_3^-{}_{atm}$ within total nitrate. We concluded that the difference in the addition of
anthropogenic nitrate composed of $NO_3^-{}_{re}$ in the catchments was primarily responsible for the
difference in the mixing ratio of unprocessed $NO_3^-{}_{atm}$ within the total nitrate pool, as well as
$C_{total}$ variation in accordance with the population densities of the catchment area as shown in
Fig. 1. That is, a small contribution of anthropogenic nitrate in the catchments in the northern
rivers was primarily responsible for the low $C_{total}$ and thus the high mixing ratio of
unprocessed $NO_3^-{}_{atm}$ within the total nitrate pool, or $C_{atm}/C_{total}$ ratios, in the northern streams
of the watershed.
Although the difference in the accumulation rate of snow between each catchment zone was
not the major factor controlling the $C_{atm}/C_{total}$ ratios, the concentrated release of $NO_3^-{}_{atm}$
stored in the snowpack during the melting seasons should be one of the important factors in
determining the $C_{atm}$ variation among the streams. Most of the $C_{atm}$-enriched streams, such as
Nos. 3, 4, 5, 14, and 25, originated from forested catchment in high elevations of more than
800 m above sea level; thus, we can anticipate heavy snowpack in winter of each headwater.
Moreover, the maximum $C_{atm}$ values in these streams were found in March, which is the
season of snowmelt (Fig. 5). On the contrary, most of the $C_{atm}$-depleted streams such as Nos.
29 (0.9 ± 0.3 µmol L$^{-1}$), 19 (0.8 ± 0.2 µmol L$^{-1}$), 23 (0.5 ± 0.2 µmol L$^{-1}$), and 30 (0.6 ± 0.2
µmol L$^{-1}$) originated from low elevations having urban and sub-urban catchment areas (Table
1). As a result, the concentrated release of stored $NO_3^-{}_{atm}$ in the snowpack to the forest floor
in the catchment zone during the melting seasons is strongly responsible for the $C_{atm}$





enrichment for some of the streams, particularly that in the streams during March as presented
in Fig. 1.
The only exception is stream No. 21, located in the southernmost part of the watershed, which
showed a high annual average $C_{atm}$ of $4.2 \pm 0.8$ µmol $L^{-1}$. This small stream originated from a
low elevation of about 200 m having a small catchment area of 4 km$^2$. In addition, although
the other $C_{atm}$-enriched streams showed the maximum $C_{atm}$ in March, that in No. 21 was
highest in August, showing extraordinarily high $C_{atm}$ of more than 10 µmol $L^{-1}$. It is unlikely
that the $NO_3^-{}_{atm}$ stored in the snowpack in winter was the major source of $NO_3^-{}_{atm}$ in this
stream.
The catchment zone of stream No. 21 showed the highest population density within the
catchments of the streams studied (Table 1). About one-third of the catchment includes
residential areas. Artificial drainage systems in urban or residential areas and agricultural
lands in humid temperate regions are usually designed to drain rainwater efficiently into
streams (Takimoto et al. 1994). As a result, a significant portion of $NO_3^-{}_{atm}$ deposited into the
catchment area was deposited onto paved surfaces and was then drained directly into the
stream via storm sewers without penetrating the ground. Thus, no interaction occurred with
soils, as presented in Fig. 1. Because biogeochemical interactions within soils are the major
sink for $NO_3^-{}_{atm}$ and thus for $^{17}O$ anomalies of nitrate (Nakagawa et al. 2013; Tsunogai et al.
2014), the development of such sewage systems in urban/suburban areas is strongly
responsible for the high $C_{atm}$ in stream No. 21. Similar bypassing effects of $NO_3^-{}_{atm}$ from soil
contacts by paved surfaces have been suggested in urban/suburban watersheds by using $\delta^{18}O$
values of nitrate as tracers (Burns et al. 2009; Kaushal et al. 2011). We further verified that
the sewage systems in urban/suburban catchments changed the fate of the $NO_3^-{}_{atm}$ deposited
onto land to some extent.
The observed uniform $C_{atm}$ irrespective of population densities and headwater elevations
shown in Fig. 7 implies that the influences of snow packs and paved surfaces were still minor
in determining the $C_{atm}$ values in the streams. Rather, the observed stable $C_{atm}$ implies that
most of $NO_3^-{}_{atm}$ in the streams had been stored in groundwater/subsurface runoff in the
watershed having similar $C_{atm}$ concentrations and then gushed to the surface at respective
headwater zones of various elevations and various land uses as presented in Fig. 1.
When using the $\delta^{18}O$ tracer, it was difficult to determine the precise absolute concentration of
$NO_3^-{}_{atm}$ ($C_{atm}$) in each stream water as presented in this study and to determine whether the





absolute concentration of $NO_3^-{}_{atm}$ was stable among the streams. By using the $\Delta^{17}O$ values,
however, we can determine the precise $C_{atm}$ in each stream for each season; thus, we can
clarify the fate of $NO_3^-{}_{atm}$.
**3.7    Differences in outflows from inflows**
The concentrations and $\delta^{15}N$, $\delta^{18}O$, and $\Delta^{17}O$ values of nitrate in the outflow (Seta River; No.
33) are also presented in Fig. 4. In a manner similar to the inflow streams (i.e. by using Eqs.
(3) to (5)), we estimated the annual average concentration of total nitrate in the outflow river (
$\overline{C_{total}}$) to be 13.3 μmol $L^{-1}$, the annual average $\delta^{15}N$ values ($\overline{\delta^{15}N}$) to be +13.1‰, the annual
average $\delta^{18}O$ values ($\overline{\delta^{18}O}$) to be +1.5‰, and the annual average $\Delta^{17}O$ values ($\overline{\Delta^{17}O}$) to be
+0.9‰, as presented in Fig. 4. Moreover, in a manner similar to that used for the inflow
streams (i.e. by using Eqs. (2), (6), and (7)), we estimated the annual average concentration of
$NO_3^-{}_{atm}$ in the outflow river ($\overline{C_{atm}}$) to be 0.4 ± 0.1 μmol $L^{-1}$, the annual average $\delta^{15}N_{re}$ values (
$\overline{\delta^{15}N_{re}}$) to be +13.7 ± 0.2‰, and the annual average $\delta^{18}O_{re}$ values ($\overline{\delta^{18}O_{re}}$) to be −1.2 ± 0.6‰,
as presented in Fig. 5. Similar to that for inflows, the $\Delta^{17}O$ values were typical for nitrate in
the natural stream waters. The striking features of the outflow in comparison with the inflows
were the depletions of both $C_{total}$ and $C_{atm}$ as well as the $^{15}N$ enrichment in the outflow
compared with that in the inflow (Figs. 4 and 5). Because the denitrification/assimilation
processes remove both nitrate and $NO_3^-{}_{atm}$ and preferentially consumes $^{14}N$ during the
removal, the progress of denitrification/assimilation in the lake water column can be strongly
responsible for the removal of both nitrate and $NO_3^-{}_{atm}$, as well as the $^{15}N$ enrichment of
nitrate in the outflow compared with the inflow. If this were the case in Lake Biwa, the total
nitrate efflux must have been smaller than the total nitrate influx. To quantitatively verify this
hypothesis and to evaluate the influences of the stagnated flow into the lake on nitrate, we
estimated the total influx through all of the inflow streams for nitrate and $NO_3^-{}_{atm}$, $\Delta N_{in}$, and
$\Delta A_{in}$, respectively, and the total efflux for nitrate and $NO_3^-{}_{atm}$, $\Delta N_{out}$, and $\Delta A_{out}$, respectively,
as well as the flow-weighted average $\delta^{15}N$, $\delta^{18}O$, and $\Delta^{17}O$ values of all inflows and outflows,
to discuss their changes in the lake.
The $\Delta N_{in}$ and $\Delta A_{in}$ in each interval between the observation n and the next observation n + 1
(i.e. each season) and the flow-weighted average $\delta^{15}N$, $\delta^{18}O$, and $\Delta^{17}O$ values of the inflows
(δ(n)) during each interval between the observation n and the next observation n + 1 were
determined by using the following equations:



$$\alpha = \frac{Q_{in}}{\sum\limits_{i} q_i},$$ (9)
$$\Delta N_{in}(n) = \sum\limits_{i} C_i \cdot f_i \cdot \Delta t_i \cdot \alpha,$$ (10)
$$\delta(n) = \frac{\sum\limits_{i} \delta_i \cdot C_i \cdot f_i \cdot \Delta t_i}{\sum\limits_{i} C_i \cdot f_i \cdot \Delta t_i},$$ (11)
$$\Delta A_{in}(n) = \Delta N_{in}(n) \cdot \frac{\Delta^{17}O_{in}(n)}{\Delta^{17}O_{atm}},$$ (12)
$$\Delta N_{in} = \sum\limits_{n=1}^{4} \Delta N_{in}(n),$$ (13)
$$\Delta A_{in} = \sum\limits_{n=1}^{4} \Delta A_{in}(n),$$ (14)
$$\delta = \frac{\sum\limits_{n=1}^{4} \delta(n) \cdot \Delta N_{in}(n)}{\sum\limits_{n=1}^{4} \Delta N_{in}(n)},$$ (15)
where $Q_{in}$ denotes the annual gross influx of water into the lake; $C_i$ and $\delta_i$ denote the
concentration and isotopic values ($\delta^{15}N$, $\delta^{18}O$, or $\Delta^{17}O$) of nitrate on each stream $i$ during each
observation $n$, respectively; $f_i$ denotes the flow rate of each stream $i$ during each observation
$n$; and $\Delta t_n$ denotes the time interval between the observation $n$ and the next observation $n + 1$.
For $Q_{in}$, we used the annual influx of water estimated by Kunimatsu et al. (1995), in which the
influx via streams and that via groundwater were included. To include the influx of nitrate via
groundwater and the other minor streams not measured in this study during the calculations,
we used the correction factor $\alpha$ in Eq. (9), whereby we assumed that both the average
concentration and average isotopic compositions of the inflows determined in this study
represented those of all inflows into the lake, while assuming an error range of 20% on $\alpha$.
By using the aforementioned equations, we estimated the total influx of nitrate to the lake
($\Delta N_{in}$) for each interval, together with the average $\delta^{15}N$, $\delta^{18}O$, and $\Delta^{17}O$ values of nitrate
during each interval, as presented in Table 2. Moreover, by using the values of $\Delta N_{in}$ during
each interval, as well as their $\delta^{15}N$, $\delta^{18}O$, and $\Delta^{17}O$ values, we estimated the total influx of
$NO_3^-{}_{atm}$ to the lake ($\Delta A_{in}$) and their average $\delta^{15}N_{re}$ and $\delta^{18}O_{re}$ values for each interval, as





presented in Table 2, by using Eqs. (12), (6), and (7). Furthermore, we estimated the annual
total influx and their annual average values, as shown in Table 2.
The estimated annual average $\Delta^{17}O$ value of inflows, +1.3‰, corresponded to the average
mixing ratio of $NO_3^-{}_{atm}$ within total nitrate was 5.1 ± 0.5%. We concluded that about 5% of
the total nitrate in the inflows originated directly from the atmosphere; therefore, the
remainder of the nitrate was of remineralized origin ($NO_3^-{}_{re}$) likely produced through
nitrification within the catchments, as discussed in section 3.4. In addition, we estimated the
annual total influx of nitrate to the lake ($\Delta N_{in}$) to be 199 ± 40 Mmol and that of $NO_3^-{}_{atm}$
($\Delta A_{in}$) to be 10.1 ± 2.0 Mmol.
Moreover, we estimated the total efflux of nitrate from the lake via the outflows ($\Delta N_{out}$) and
that of $NO_3^-{}_{atm}$ ($\Delta A_{in}$) for each interval by using Eqs. (9) to (15) in which $\Delta N_{in}$ was replaced
with $\Delta N_{out}$, which is the gross efflux of nitrate from the lake via the streams and groundwater
during the interval between the observation n and the next observation n + 1. Additionally, $Q_{in}$
was replaced with $Q_{out}$, which is the annual gross efflux of water. To include the minor efflux
of nitrate to $\Delta N_{out}$, such as that via canals, we used the correction factor γ instead of α in Eqs.
(9) and (10), whereby we assumed that both the concentration and isotopic compositions of
the natural outflow determined for each season in this study represented all outflows. For $Q_{out}$,
we used the annual efflux of water from Lake Biwa estimated by Kunimatsu et al. (1995),
which included the efflux via a natural river (Seta River, No. 33) and that via canals.
Compared with the annual $\Delta N_{in}$, 199 ± 40 Mmol, and annual $\Delta A_{in}$, 10.1 ± 2.0 Mmol, both the
annual $\Delta N_{out}$, 67 ± 13 Mmol, and annual $\Delta A_{out}$, 2.2 ± 0.4 Mmol, were significantly smaller by
about 34% and 22%, respectively. Hence, Lake Biwa acts as a net sink for both nitrate and
$NO_3^-{}_{atm}$, as previously implied from the $^{15}N$ enrichment in outflows. Considering that nitrate
occupied about 70% of the total fixed N pool in the inflows and about 40% of the total fixed
N pool in the outflows (Shiga_Prefecture 2015), Lake Biwa also acts as a net sink for fixed N.
Similar results had been obtained in past studies that discussed fixed N input/output of the
lake (Kunimatsu 1995; Tezuka 1985; Tezuka 1992; Yamada et al. 1996). As implied in the
significant $^{15}N$ enrichment in the remineralized portion of nitrate ($\delta^{15}N_{re}$) in the outflow,
+13.7 ± 0.2‰, compared to that in the inflow, +5.6 ± 0.3‰, partial removal of nitrate through
either assimilation or denitrification to result in $^{15}N$ enrichment of residual nitrate is strongly
responsible for the 8.1 ± 0.5‰ increase in $\delta^{15}N_{re}$, as well as the net removal of both nitrate
and $NO_3^-{}_{atm}$ from the lake.





On the contrary, the $\delta^{18}O$ differences in the remineralized portion of nitrate ($\delta^{18}O_{re}$) between
the inflows and outflows were significantly smaller than $\delta^{15}N_{re}$, at an annual average of only
$1.6 \pm 1.5‰$ (Table 2). If nitrate in outflows is the residual of assimilation/denitrification in the
lake, $\delta^{18}O_{re}$ should increase as well (Granger et al. 2008; Granger et al. 2004). The much
smaller $\delta^{18}O_{re}$ difference implies that nitrate supplied directly from inflows occupied a small
portion of nitrate in the outflows and that most of nitrate with high $\delta^{15}N$ values in the outflows
had been produced through nitrification in the lake water column in which the fixed N had
previously been enriched in $^{15}N$. Isotopic fractionations during fixed N cycling in the lake
such as denitrification or assimilation and the subsequent removal of $^{15}N$-depleted organic N
during sedimentation (Fig. 1) should be responsible for the $^{15}N$ enrichment of the total fixed N.
That is, most of nitrate fed into the lake via the inflows had been removed at least once from
the lake water column and been involved into the total fixed N cycling in the lake, in which
the $^{15}N$-enriched nitrate in the outflow had been produced (Fig. 1). The stagnation of flow in
the lake encouraged primary production and thus the net removal of total fixed N through
either denitrification or sedimentation, which resulted in $^{15}N$ enrichment of total fixed N pool
compared with that in the inflows. Further studies on N cycling in the lake are needed to
verify these results.
**4    Concluding Remarks**
In this study, we applied the $\Delta^{17}O$ tracer of nitrate to determine accurate and precise mixing
ratios of unprocessed $NO_3^-{}_{atm}$ within the total nitrate value for more than 30 streams in the
Lake Biwa watershed basin. Although the nitrate concentration changed from 12.7 to 106.2
$\mu mol\ L^{-1}$ between the inflow streams and the mixing ratio of $NO_3^-{}_{atm}$ within total nitrate also
changed from 1.8% to 11.8%, the absolute concentration of $NO_3^-{}_{atm}$ ($C_{atm}$) in each stream
water was almost stable at $2.3 \pm 1.1\ \mu mol\ L^{-1}$ irrespective of the changes in population
density and land use among the catchment areas. We concluded that changes in population
density and land use in each catchment area had little impact on $C_{atm}$, and the total nitrate
concentration was determined primarily by the extent of the additional $NO_3^-{}_{re}$ contribution
mostly from anthropogenic sources. Relying on only the $\delta^{15}N$ and $\delta^{18}O$ tracers of nitrate, it
was difficult to determine the precise $C_{atm}$ in stream water and whether $C_{atm}$ was uniform
among the streams. By using the $\Delta^{17}O$ values, we can estimate accurate and precise $C_{atm}$ in





each stream for each season; thus, we can clarify the fate of $NO_3^-{}_{atm}$ deposited into the
catchments.
Moreover, additional measurements of the $\Delta^{17}O$ values of nitrate together with $\delta^{15}N$ and $\delta^{18}O$
enabled us to exclude the contribution of $NO_3^-{}_{atm}$ from the determined $\delta^{15}N$ and $\delta^{18}O$ values
and to use the corrected $\delta^{15}N$ and $\delta^{18}O$ values, $\delta^{15}N_{re}$ and $\delta^{18}O_{re}$, to evaluate the source and
behaviour of $NO_3^-{}_{re}$ in each stream. Based on the correction, we successfully estimated the
$\delta^{15}N$ and $\delta^{18}O$ values of $NO_3^-{}_{re}$ in the streams to be $+4.4 \pm 1.8‰$ and $-2.3 \pm 0.9‰$,
respectively, for that produced through nitrification in naturally occurring soil organic matter
and $+9.2 \pm 1.3‰$ and $-2.2 \pm 1.1‰$, respectively, for that supplied from anthropogenic
sources, most of which had been occupied by sewage effluent. In addition, the low and
uniform annual average $\delta^{18}O_{re}$ values of $NO_3^-{}_{re}$ in the streams implies that the denitrification
in the riverbed sediments was minor in the streams.
Furthermore, we clarified the seasonal changes in the range of isotopic fractionation through
partial nitrate removal via assimilation or denitrification by using the relationship between
$\Delta^{17}O$ and $\delta^{18}O$ of nitrate in the streams. The changes were small in winter in all of the
catchment area but large in summer in some catchments. Therefore, the relationship between
$\Delta^{17}O$ and $\delta^{18}O$ of nitrate eluted from a catchment area is a powerful indicator for evaluating
the biogeochemical nitrogen cycles within a catchment area, including the seasonal changes.
In summary, interpretations on the stable isotopic compositions of nitrate ($\delta^{15}N$ and $\delta^{18}O$)
without $\Delta^{17}O$ values can be often misleading when assuming (1) $\delta^{18}O$ values simply reflect
the mixing ratio of $NO_3^-{}_{atm}$ within total nitrate, (2) the mixing ratios of $NO_3^-{}_{atm}$ within the
total $NO_3^-$ is minimum for whole/specific samples studied, or (3) the mixing ratio of $NO_3^-{}_{atm}$
within total nitrate is uniform in whole/specific samples studied. In using the stable isotopic
compositions of nitrate in freshwater environments to trace the source and fate of the nitrate,
particularly the $\delta^{18}O$ value, determination of the $\Delta^{17}O$ values is essential.
**Acknowledgements**
We are grateful to Kosuke Ikeya, Hiroki Sakuma, Sho Minami, Kenta Ando, Shuichi Hara,
Toshiyuki Matsushita, Takahiro Mihara, Teresa Fukuda, Yoshiumi Matsumoto, Rei Nakane,
Lin Cheng, Yuuko Nakano, and other present and past members of the Biogeochemistry
Group, Nagoya University, for their valuable support throughout this study. We thank Drs.



Shin-ichi Nakano, Tadatoshi Koitatabashi, Yukiko Goda, and other staff of the Center for
Ecological Research, Kyoto University, for their valuable support during the field study in the
Lake Biwa watershed basin. We also thank the members of the Machine Shop of Nagoya
University Technical Center for their valuable support in developing the sampling and
analytical devices used in this study. This work is supported by a Grant-in-Aid for Scientific
Research from the Ministry of Education, Culture, Sports, Science, and Technology of Japan
under grant numbers 24651002, 26241006, and 15H02804.

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




1    Table 1 List of studied streams.

| No. | Name | Loc.[#] | Basin Area[*] (km$^2$) | Population Density[*] (km$^{-2}$) | Residential[*] (%) | No. | Name | Loc.[#] | Basin Area[*] (km$^2$) | Population Density[*] (km$^{-2}$) | Residential[*] (%) |
|---|---|---|---|---|---|---|---|---|---|---|---|
| ***Inflow*** | | | | | | | | | | | |
| 31 | Tenjin | West | 10 | 539 | 7.3 | 14 | Seri | East | 74 | 462 | 8.1 |
| 30 | Mano | West | 23 | 1048 | 15.6 | 15 | Inukami | East | 102 | 109 | 2.8 |
| 29 | Wani | West | 17 | 186 | 5.4 | 16 | Ajiki | East | 15 | 1002 | 20.1 |
| 28 | U | West | 7 | 66 | 0.8 | 17 | Uso | East | 84 | 411 | 11.0 |
| 1 | Kamo | West | 47 | 89 | 1.2 | 18 | Bunroku | East | 14 | 595 | 11.7 |
| 2 | Ado | West | 306 | 27 | 0.7 | 19 | Nomazu | East | 7 | 758 | 20.2 |
| 3 | Ishida | North | 60 | 84 | 2.0 | 20 | Echi | East | 211 | 110 | 2.5 |
| 4 | Momose | North | 13 | 65 | 0.6 | 27 | Hino | South | 226 | 338 | 8.4 |
| 5 | Chinai | North | 51 | 44 | 1.8 | 26 | Yanomune | South | 42 | 859 | 15.7 |
| 6 | Ohura | North | 39 | 98 | 2.7 | 25 | Yasu | South | 391 | 324 | 6.5 |
| 7 | Oh | North | 20 | 55 | 1.6 | 24 | Yamaga | South | 6 | 2540 | 33.7 |
| 8 | Yogo | North | 7 | 141 | 3.4 | 23 | Sakai | South | 2 | 979 | 27.7 |
| 9 | Chonoki | North | 10 | 412 | 13.5 | 22 | Hayama | South | 34 | 2048 | 29.7 |
| 10 | Ta | North | 36 | 301 | 9.6 | 34 | Kusatsu | South | 48 | 370 | 30.6 |
| 11 | Ane | North | 372 | 61 | 1.7 | 21 | Nagaso | South | 4 | 3174 | 31.4 |
| 12 | Yone | North | 15 | 2047 | 34.5 | 32 | Fujinoki | South | 4 | 1805 | 20.8 |
| 13 | Amano | North | 111 | 226 | 5.8 | | | | | | |
| | | | | | | ***Outflow*** | | | | | |
| | | | | | | 33 | Seta | South | 3848 | 323 | − |

2    [#] Category of location classified by Ohte et al. (2010).

3    [*] Data source: Ohte et al. (2010).



Table 2 Estimated gross influx/efflux of total nitrate (ΔN) and atmospheric nitrate (ΔA) via
inflows/outflows during each observation interval, together with the average $\delta^{15}N$, $\delta^{18}O$, and
$\Delta^{17}O$ values of total nitrate and remineralized portions of nitrate ($\delta^{15}N_{re}$ and $\delta^{18}O_{re}$) in the
inflows/outflows during each interval.

|  | Spring (n=1 to 2) | Summer (n=2 to 3) | Autumn (n=3 to 4) | Winter (n=4 to 5) | Annual (n=1 to 5) |
|---|---|---|---|---|---|
| Duration (days) | 94 | 49 | 77 | 145 | 365 |
| ***Inflow*** | | | | | |
| $\Delta N_{in}$ ($10^6$ mol) | 69 ± 14 | 3 ± 1 | 13 ± 3 | 114 ± 2 3 | 199 ± 40 |
| $\Delta A_{in}$ ($10^6$ mol) | 6.4 ± 1.3 | 0.1 | 0.8 ± 0.2 | 2.8 ± 0.6 | 10.1 ± 2.0 |
| $10^3$ $\delta^{15}N$ | +4.0 | +6.8 | +5.6 | +5.6 | +5.1 |
| $10^3$ $\delta^{18}O$ | +6.1 | −0.8 | +3.3 | −1.5 | +1.4 |
| $10^3$ $\Delta^{17}O$ | +2.5 | +0.8 | +1.7 | +0.6 | +1.3 |
| $10^3$ $\delta^{15}N_{re}$ | +4.8 ± 0.7 | +7.1 ± 0.2 | +6.3 ± 0.5 | +5.9 ± 0.2 | +5.6 ± 0.3 |
| $10^3$ $\delta^{18}O_{re}$ | −1.5 ± 1.8 | −3.2 ± 0.5 | −2.0 ± 1.2 | −3.5 ± 0.4 | −2.8 ± 0.9 |
| ***Outflow*** | | | | | |
| $\Delta N_{out}$ ($10^6$ mol) | 24 ± 5 | 6 ± 1 | 5 ± 1 | 32 ± 6 | 67 ± 13 |
| $\Delta A_{out}$ ($10^6$ mol) | 1.4 ± 0.1 | 0.1 | 0.2 | 0.4 ± 0.1 | 2.2 ± 0.4 |
| $10^3$ $\delta^{15}N$ | +7.3 | +11.4 | +10.4 | +18.0 | +13.1 |
| $10^3$ $\delta^{18}O$ | +3.4 | +4.8 | +3.0 | −0.7 | +1.5 |
| $10^3$ $\Delta^{17}O$ | +1.6 | +0.4 | +1.4 | +0.4 | +0.9 |
| $10^3$ $\delta^{15}N_{re}$ | +8.1 ± 0.4 | +11.7 ± 0.1 | +11.2 ± 0.4 | +18.3 ± 0.1 | +13.7 ± 0.2 |
| $10^3$ $\delta^{18}O_{re}$ | −1.5 ± 1.1 | +3.6 ± 0.3 | −1.2 ± 0.9 | −1.9 ± 0.2 | −1.2 ± 0.6 |





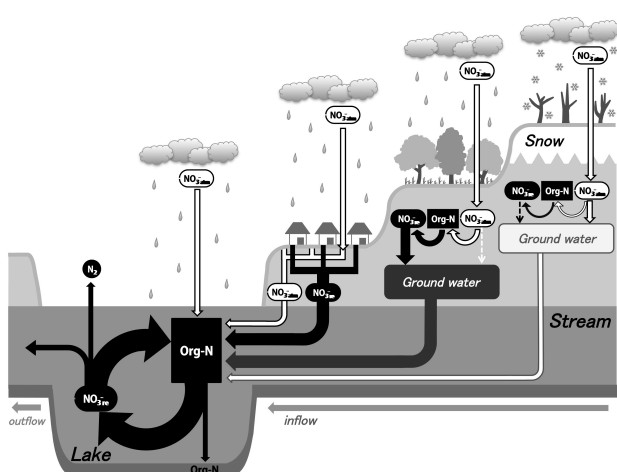

2 Figure 1. Schematic diagram showing the biological processing of atmospheric nitrate

3 (NO$_3^-{}_{atm}$) and remineralized nitrate (NO$_3^-{}_{re}$) in the watershed with catchments of varying land

4 uses and in the lake water column.




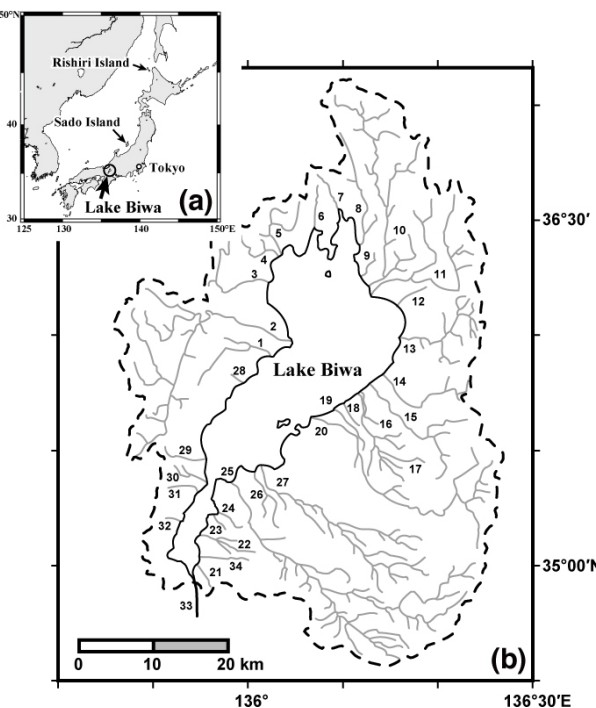

Figure 2.  (a) Map showing the location of Lake Biwa watershed basin in Japan and Sado
Island, where the Sado-seki National Acid Rain Monitoring Station is located. (b) Map
showing the boundary of the Lake Biwa watershed basin (dashed line) and the locations of the
inflows (represented by numbers) and outflow (Seta River, No. 33) studied in this paper
(modified from Ohte et al. 2010).





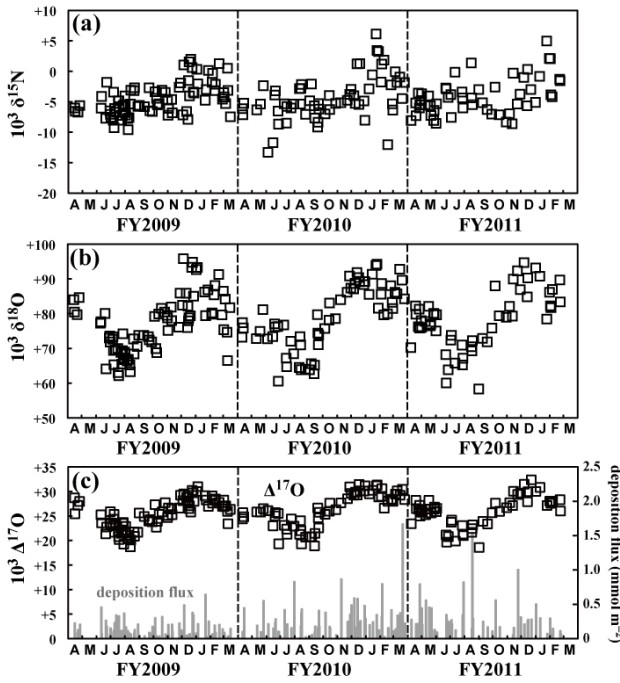

2    Figure 3. Temporal variations in the values of (a) $\delta^{15}N$, (b) $\delta^{18}O$, and (c) $\Delta^{17}O$ of nitrate in wet

3    deposition recorded at the Sado-seki National Acid Rain Monitoring Station. Total deposition

4    flux of nitrate during each rain event was also presented (c).





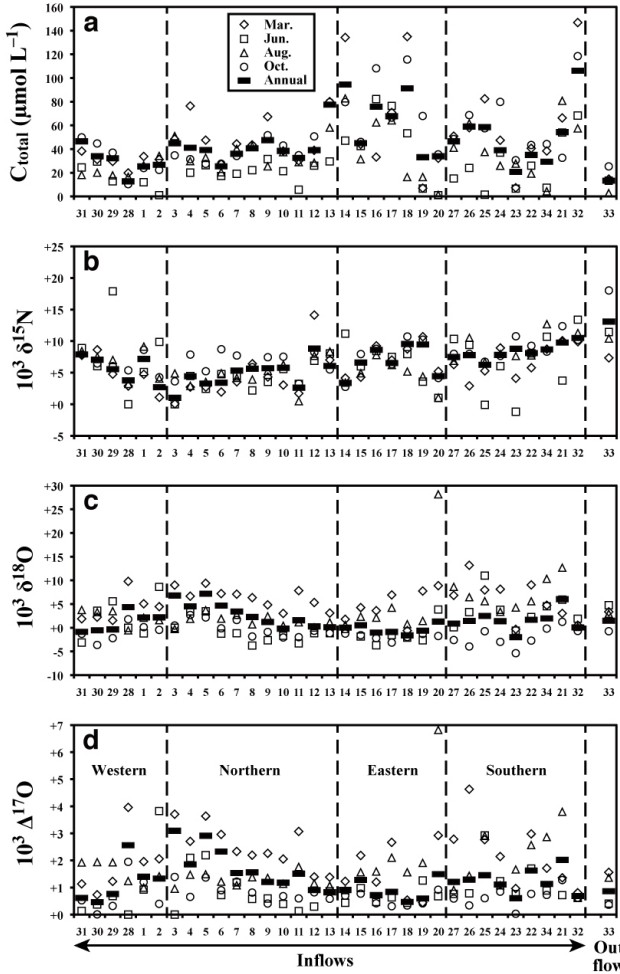

Figure 4. Distribution of (a) total concentrations, (b) $\delta^{15}N$, (c) $\delta^{18}O$, and (d) $\Delta^{17}O$ for nitrate in
inflow streams showing various station numbers and the outflow at station No. 33 in Lake
Biwa watershed in March (diamonds), June (squares), August (triangles), and October
(circles) 2013 together with the annual averages for each river (black bars).





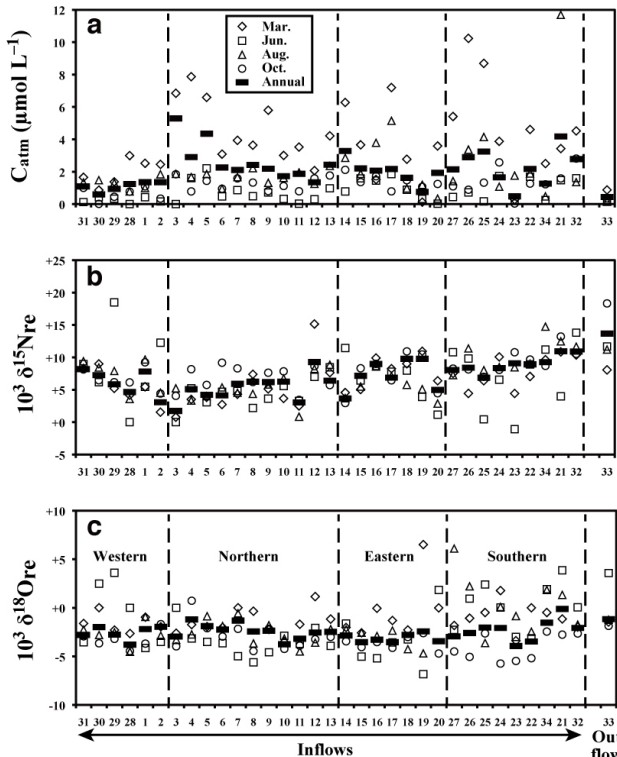

Figure 5. Distribution of concentrations of (a) atmospheric nitrate and (b) $\delta^{15}N$ and (c) $\delta^{18}O$ for remineralized nitrate ($\delta^{15}N_{re}$ and $\delta^{18}O_{re}$, respectively) in inflow streams shown by various station numbers and the outflow at station No. 33 of Lake Biwa watershed in March (diamonds), June (squares), August (triangles), and October (circles) 2013 together with the annual averages for each river (black bars).



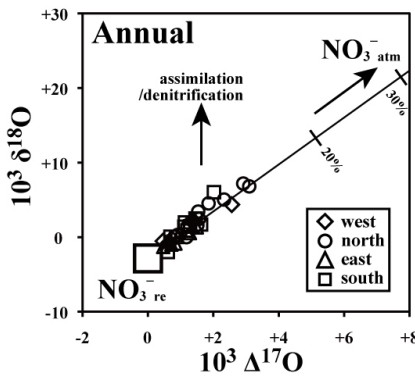

Figure 6. Relationship of the annual average values of $\Delta^{17}O$ and $\delta^{18}O$ of $NO_3^-$ in the inflow
streams. The symbols represent the location of each river (west: diamonds; north: circles;
east: triangles; south: squares). A hypothetical mixing line between atmospheric nitrate
($NO_3^-{}_{atm}$) and remineralized nitrate ($NO_3^-{}_{re}$) is shown together with the end member value of
$NO_3^-{}_{re}$ (large white square) and the 20% and 30% mixing ratios of $NO_3^-{}_{atm}$ on the line.



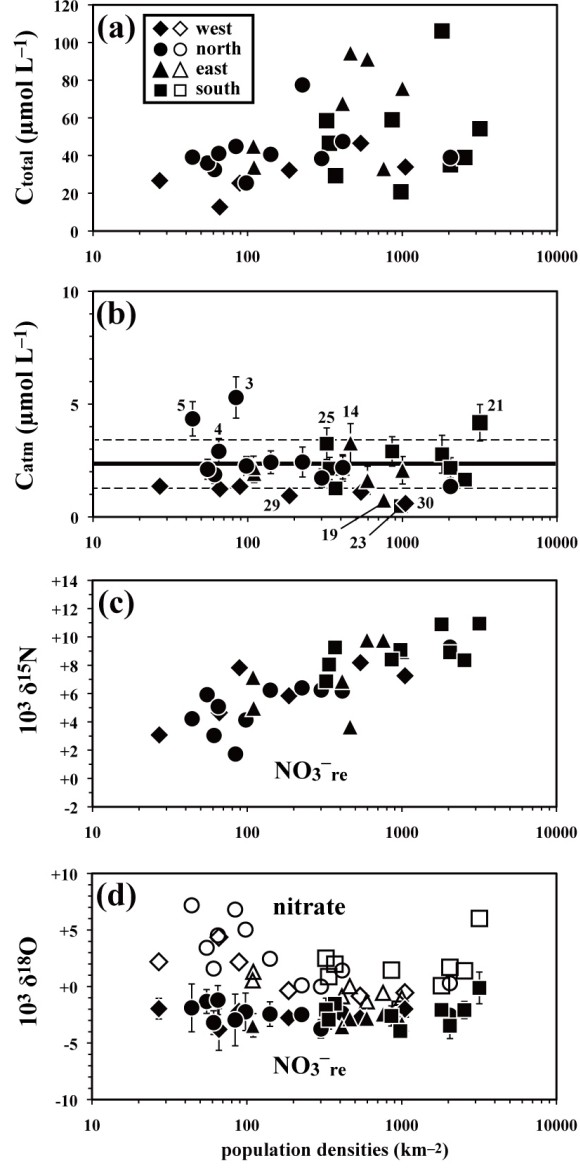

Figure 7. Annual average concentration of (a) nitrate ($C_{total}$) and that of (b) $NO_3^-{}_{atm}$ ($C_{atm}$) in each inflow stream plotted as a function of the population density in each catchment, together with (c) the annual average value of $\delta^{15}N$ and (d) that of $\delta^{18}O$ for remineralized $NO_3^-$ ($NO_3^-{}_{re}$). (d) The annual average $\delta^{18}O$ values of nitrate are also presented. The symbols represent the location of each river (west: diamonds; north: circles; east: triangles; south: squares). The uncertainties are presented only for those larger than the symbols.





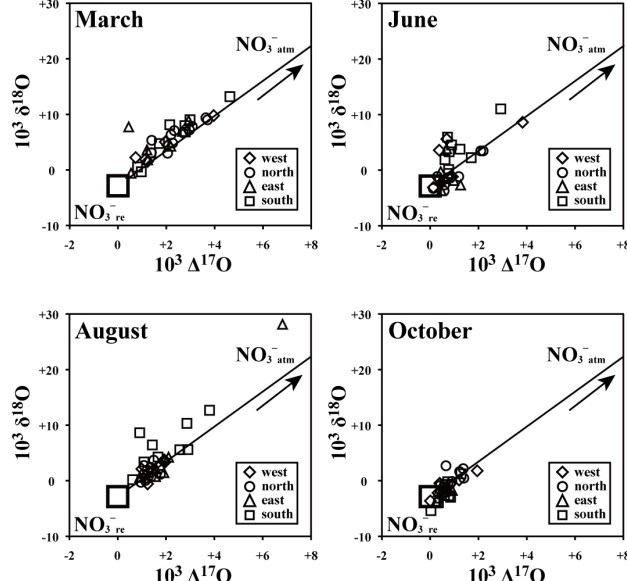

Figure 8. Temporal changes in the relationship between the values of $\Delta^{17}O$ and $\delta^{18}O$ of total nitrate in the stream water. The symbols are the same as those in Figure 6. A hypothetical mixing line between atmospheric nitrate ($NO_3^-{}_{atm}$) and remineralized nitrate ($NO_3^-{}_{re}$) is also shown together with the end member value of $NO_3^-{}_{re}$ (large white square).