# Peer review of "Accurate and precise quantification of atmospheric nitrate in streams draining land of various uses by using triple oxygen isotopes as tracers"

_Biogeosciences, 2015_

## Referee Comment (RC1) · Anonymous Referee #1 · 15 Feb 2016

**1 General comments**

This paper by Tsunogai et al. builds on previous research by Ohte et al. (2010) at Lake Biwa. The authors have quantified the relative contribution of atmospheric nitrate to stream nitrate using oxygen triple isotope measurements, their seasonal variability and interannual variability. In conjunction with Lake isotope measurements, the data are used to present a nitrate budget for Lake Biwa as a whole.

The study is comprehensive in its spatial coverage and includes data from 32 streams.
[Figure]

Temporal resolution (4 sampling periods over 1-year interval) is somewhat limited, but in light of the residence time of nitrate in the Lake (5.5 years), it is sufficient.

Compared to the work of Ohte et al. (2010), the main advance of the present study is the inclusion of oxygen triple isotope measurements. Other aspects (e.g. the correlation between nitrogen isotope ratios and population density) have been noted before.

The paper is generally well-written and complete, with a few exceptions, notably the Conclusions section disregard large fractions of the discussion (e.g. on seasonal and interannual variations as well as the Lake Biwa nitrate budget). The presentation quality (except for the figures) is mostly good, but there are a number of grammatical errors. Language copy-editing will be required.

Before the paper could be accepted for publication in Biogeosciences, several major areas need to be improved as detailed below.

In line with the Biogeosciences data policy (http://www.biogeosciences.net/about/data_ policy.html) , all data (water fluxes, nitrate and nitrite concentrations, isotope values, etc.)  should be publicly available, preferably by deposition in a data repository, or alternatively as electronic supplementary information. Please add a section on "Data availability".

The figures are somewhat antediluvian in their appearance due to the lack of colours. While the stated aim to make them compatible with black and white printer is commendable, most readers will either view them on a colour display or printed in colour. The figures should be redrawn in colour. The current use of different shapes to distinguish time series can be retained, for the benefit of the small number of readers without access to colour displays and printers, and for those with impaired colour vision.

The figures in the supplementary information are discussed at crucial points and called out in the text. Fragmentation of the text into different documents is undesirable. They should be merged with the main text. The supplementary information should be used

instead to present the full dataset in a table (water fluxes, nitrate and nitrite concentrations, isotope values, etc.), unless the authors can give a URL or DOI at a repository, at which they have lodged the data.

The treatment of systematic uncertainties and propagation of measurement uncertainties to derived properties is insufficient and/or has been insufficiently well presented. In particular, variations in the isotopic composition of the atmospheric nitrate end member (delta17O, delta15N and delta18O) should be better documented and listed in a table. The resultant uncertainties in the calculation of the atmospheric nitrate fraction and the delta15N and delta18O values of the remineralised nitrate should be included in the text and figures of the paper.

A major systematic uncertainty arises from the current disregard of nitrate sinks and associated isotopic fractionation (section 3.5). Specifically, the delta15N and delta18O values of so-called "remineralised nitrate" may have undergone large modifications due to nitrate assimilation or denitrification. These processes have been currently ignored. While their effect on delta17O is most likely negligible, they cannot be discounted for delta15N and delta18O. In other words, assimilation and denitrification could have enriched "remineralised nitrate" in both 15N and 18O isotopes. The possibility of this should be explored further, perhaps in the first instance using delta15N and delta18O scatter plots of the "remineralised nitrate" component. If there is a noticeable potential influence, this should be quantified as good as possible. The authors may also want to consider renaming the "remineralised nitrate" component as "residual nitrate" or another less prejudiced term.

**2  Specific comments**

The term "17O anomalies" has been criticised for its subjectivity and biased nuances. A more neutral term, which has been adopted by many authors, is "17O excess" (similar

to deuterium excess and thermodynamic "excess" properties; a negative "17O excess" reflects a deficit). Alternatively, "delta17O values" would also be acceptable.

The term "mixing ratios" is ambiguous and should be replaced by "mole fractions" or "mass fractions".

Averages and uncertainties should be enclosed in round brackets so that the unit applies to both, e.g. p. 2/3 "(2.3±1.1) $\mu$mol L-1", in line with standard practice in natural sciences.

In line with international conventions, chemical phase information should be added in round brackets after the chemical formula, i.e. NO3-(atm) and NO3-(re), not as subscript index. Alternatively, text abbreviations should be used, e.g. AN for atmospheric nitrate and RN for remineralised / residual nitrate.

Title: The words "accurate and precise" should be dropped from the title. The claimed "accuracy" would require an absolute measurement method and/or validation by an independent laboratory, neither fo which have been performed. "Precision" is not a meaningful metrological term. In any case, the quantification method has known significant systematic uncertainties, e.g. the isotopic composition of atmospheric nitrate deposition, temporal undersampling, etc., so neither the qualifications "accurate" nor "precise" apply.

Abstract: A sentence should be inserted at the beginning of the abstract that explains the motivation for the present study. The current first sentence of the abstract is too long and should be split into two.

Abstract p. 2, lines 14 to 18 should be moved up to before "We conclude ..." in line 4.

Introduction: The discussion misses fertiliser inputs and groundwater recharge as nitrate sources - see also p. 12.

4/9: Use of beta = 0.5247 is unusual. Most other research groups use 0.528, based on the meteoric water relationship (e.g. Savarino, Hastings, Michalski). The references

given by the authors either do not express a preference (Miller 2002) or also use a coefficient of nearly 0.528 (Kaiser et al. 2007).

4/12: This statement only applies to mass-dependent fractionation processes following a slope of 0.5247. It also does not apply to mixing applications because mixing follows linear, not power-law relationships. As in the present case, the authors are primarily using delta17O as a mixing tracer, a linear definition of delta17O would be preferable. At the least, the potential systematic error due to fractionating processes following other mass-dependent relationships (e.g. beta = 0.5) and the effect of linear mixing on the non-linear delta17O definition used here should be quantified and carried forward to the subsequent discussion.

Sections 2.1 and 2.2 should be merged. Sections 2.1.2 and the last paragraph of sections 2.2 are highly repetitive.

The use of Japanese "financial years" should be abandoned in favour of the use of calendar years as mandated by the manuscript submission guidelines for Biogeosciences. All dates should be revised accordingly, as well as the calculation of annual averages.

7/7-14: This discussion is confusing. At the least, it needs to be made clear that there is no 5th sample and that the authors have instead used the data from the previous year (March 2013) to calculate changes over the so-called "winter period", i.e. the difference between October 2013 and March 2013. Rather than using "sampling" numbers, the calculation scheme could perhaps be better presented in a table.

7/28-8/2: The H2O isotope measurements referenced here have not been presented in the manuscript. Please include the results as a figure or table and include the data in the supplementary information.

8/3-11: There are too many references given here. Please give just the one that documents the method used here.

The azide method is known to require larger isotope exchange corrections for oxygen

isotope ratios. How large was the required exchange correction applied to the raw measurements?

8/12-17: It is not appropriate to use ion chromatography as a reference method to determine the efficacy of N2O conversion. Ion chromatography can also be affected by measurement biases. Nitrate standards should be used instead to check the azide method is working properly. Please explain how many samples have been excluded based on the ion chromatographically determined concentrations.

8/24: What "error" do these values represent? Are they the standard deviations of 3 replicate measurements?

8/28: Please replace "approximation" by "definition". The definition of delta17O can be made in any arbitrary fashion; there is no approximation in a definition. However, of course, the interpretation of the resultant delta17O values may change, depending on the definition.

9/4: A 5 % contribution of nitrite leads to a significant bias in the delta18O value and cannot be neglected. The bias could be as large as -1.3 ‰ because the isotopic fractionation associated with conversion of nitrite to N2O is about 27 ‰ lower than for conversion of nitrate to N2O (e.g. Casciotti et al. 2007). All measurements need to be corrected for this bias, or re-analysed after NO2- removal (e.g. following the sulfamic acid protocol of Granger Sigman 2009).

9/11-13: It is unclear how the monthly stream flow measurements are used together with the less frequent nitrate concentration and isotope measurements. Do you ignore the flow measurements from the months when no sampling has occurred? Do you interpolate the concentrations to match each flow measurement with a corresponding nitrate concentration or isotope delta value? It seems that you ignore major parts of the flow (based on the counting index of 4). How much stream flow is "missed" due to this temporal undersampling?

[Figure]

10/13: Please replace "isotopic compositions" with "excess" - see above. "Composition" is not an extensive quantity.

11/13-16: What are uncertainties / variability in the atmospheric end member values? Which values and which uncertainty did you actually use when you calculated the delta values of residual nitrate (see also p. 12, l. 26-31)? Which error do you estimate the systematic neglect of dry deposition has caused?

14/4: What do you mean by "almost correlated"? Please give a quantitative measure of what "almost correlated" constitutes.

15/28-31: The last sentence does not make sense at the end of a paragraph. It should appear at the beginning of a new paragraph. You should then explain how you verified that the "remineralised" portion was actually responsible for the positive correlation with delta15N and population density.

16/14: It is unclear why a uniform Catm (atmospheric nitrate contribution to the total nitrate concentration) is indicative for low denitrification. This could be offset of atmospheric inputs of equal magnitude. A more sensitive approach would be to look for a correlation (or absence thereof) between delta15N and delta18O of "remineralised" nitrate.

17/23: delta15N measurements should be used to verify this hypothesis, especially in combination with delta18O (scatter plots).

22/17: Which value was used for alpha?

23/15: Which value was used for gamma?

23/25: Please speculate on the fate of nitrogen. Are they lost due to gaseous emissions (denitrification, anammox?) Sedimentary deposition? Eutrophication (secular nitrate concentration increases)?

24/29-25/2: This is not a conclusion, this is the premise/rationale of the present paper.

These lines should be deleted.

25/19-25/25: This is not a conclusion, this is the premise/rational of the present paper. These lines should be deleted.

The section Conclusions is incomplete. Seasonal and interannual changes in stream nitrate concentrations and isotopic composition are missing. Also, the substantive section on the lake nitrogen budget is not reflected by a corresponding conclusion. I would expect a statement on the apparent nitrogen sink and possible identification including any relevant past or future work.

**3  Technical corrections**

The internationally recommended symbol "a" should be used to abbreviated "year", not "yr" and "y", which have both been used in the present paper.

1/25: The word "However" does not apply - there is nothing contradictory in this sentence.

2/1: Please insert "stream" between "average" and "nitrate".

2/3: Please delete "in the streams".

10/1: Please delete "each".

12/5: Please cross-reference the location of Rishiri island in Fig. 1.

12/16: What "recession period"?

14/12: "strongly" should be deleted.

17/8: Please replace "river" with "stream".

22/1: A different symbol than alpha should be chosen because alpha is usually used

to denote isotopic fractionation factors.

Figure 7b: The numbers should be explained (stream numbers?).
* * *

---

## Referee Comment (RC2) · Anonymous Referee #2 · 24 Mar 2016

General comments

This paper presents interesting data on the quantification of unprocessed nitrate from atmospheric wet deposition in waters of 33 streams, discharging to Lake Biwa, Central Japan, by means of nitrogen (14/15N) and triple oxygen (16/17/18O) stable isotope analysis of nitrate. Stream waters were sampled four times in March, June, August and October 2013. Total nitrate inflow into the lake as well as nitrate stable isotope signatures were averaged over the year by interpolation between the four sampling dates. The main outcome was that unprocessed atmospheric nitrate made up about

5% of the total nitrate transported into Lake Biwa by the 33 streams, that the lake removed a substantial portion of the nitrate before the water left the lake again, and that both quantity and anthropogenic source signature of nitrate increased in streams discharging areas with increasing population density.

While the topic is highly relevant and timely, and the data presented contain valuable information, the paper suffers severely from several weaknesses. Firstly, it is way too long, starting with the Abstract and ending with the Conclusions. The reader gets lost in the many detailed descriptions of results, while the description of methods is partly incomplete. For example, the method how the triple oxygen stable isotope analysis has been done is not described, even not briefly. Also statistical and data evaluation methods are not described. But most importantly, the conclusions with respect to the effect of different land uses on the fate of atmospheric nitrate (which was the main motivation of the study) were based on many assumptions and uncertain values, especially by excluding more than two sources of nitrate (from atmosphere and from nitrification only). Furthermore, there were no statistical data provided that proved an unambiguous relationship between land use in the different stream catchments and the signature of nitrate in the stream water. This weakens the key message of the paper on the effect of land use and population density on the fate of atmospheric nitrate, and needs to be rectified before the paper becomes acceptable for publication.

Please see below for specific comments.

Technical corrections can be found in the annotated pdf.

Specific comments

Abstract, Results and Discussion, and Conclusions should be shortened significantly, focusing on the main outcome of the paper. It would be good if Lake Biwa were mentioned in the Abstract.

The number of references should be reduced to about 50 (from almost 80). The number

of figures should be reduced to about 6.

Title: I suggest deleting "Accurate and precise" from the title, as it suggests a very high accuracy and precision of the data presented in the paper, which is not the case (the fraction of unprocessed atmospheric nitrate in relation to total nitrate in the stream waters of about 5% had a relative error of 10%, and average $\delta$15N and $\delta$18O values were assigned with an absolute error of +/- 10‰ which is really large). Secondly, the reader might wonder why there is a differentiation between accurate and precise, which occurs also several times in the paper, but which is never explained, also not the way how to achieve both high accuracy and precision.

Data evaluation and regression methods as well as statistics are not described.

The English should be checked by a native speaker.

p. 2, l. 23-24: "important to primary production and thus eutrophication": primary production does not in itself lead to eutrophication, but only a mismatch between primary production and heterotrophic consumption, usually induced by excess nutrient load. I suggest rewording to "important to primary production, and an excess of nitrate can lead to eutrophication downstream".

p. 2, l. 28-29: I would separate assimilation by plants and microbes and denitrification by microbes in two separate processes, as they are of completely different nature.

p. 3, l. 4-10: Here you cite 25 (!) references for one statement, overshooting by far. Please reduce to the 5-6 most important papers.

p. 3, l. 10-12: As you use the bold statement "…can be quantified through a simple isotope mass balance approach", you should give ranges reported in the literature for the two isotope ratios for the different sources to allow the reader to assess the feasibility of the simple isotope mass balance approach.

p. 3, l. 22-23: "the mixing ratios of unprocessed NO3−atm within total nitrate are minimum or uniform for whole or specific stream water samples": Meaning of this sentence is unclear. Please reword.

p. 4, l. 6: "By using the $\Delta 17O$ signature...": This term should be introduced and explained, not only by an equation, but also in words.

p. 4, l.12-13: "In addition, $\Delta 17O$ is stable during the mass-dependent isotope fractionation processes within surface ecosystems.": Yes, but only if there is no oxygen exchange with the surrounding water, otherwise the $\Delta 17O$ information gets lost. That is the reason why only UNPROCESSED atmospheric nitrate can be traced, not the further processing of atmospheric nitrate itself.

p. 4, l. 14-15: "Therefore, although the atmospheric $\delta 15N$ or $\delta 18O$ signature can be overprinted by biogeochemical processes subsequent to deposition, $\Delta 17O$ can be used as a robust tracer...": Again, also $\Delta 17O$ can be "overprinted" by oxygen exchange, not only $\delta 15N$ or $\delta 18O$ of nitrate.

p. 5, l. 14: "$NO3-atm$ is stable": I disagree. Nitrate from atmospheric deposition can and will be processed after deposition. Therefore, it cannot be considered as stable.

p. 5, l. 18-20: "Moreover, we exclude the contribution of $NO3-atm$ in the determined $\delta 15N$ and $\delta 18O$ values to estimate the corrected $\delta 15N$ and $\delta 18O$ values for accurate evaluation of the source and behaviour of $NO3-re$.": Totally unclear what that means. Please explain more clearly.

p. 5, l. 21-22: "influences of flow stagnation into the lake on nitrate": What influence is meant here? On nitrate concentration? On isotope ratios? On total amount?

p. 6, l. 6-14: This paragraph should be moved to the end of the introduction as part of the motivation for the study.

p. 7, l. 7: "To calculate the annual influx/efflux of nitrate via each stream... we used the sampling number n": Unclear how the annual influx/efflux of nitrate was calculated using the sampling number. Please provide a more detailed description of the calculation. How were peak flow events after strong precipitation events or after snow melt (if there was) taken into account? Frequently, the solute composition of stream water is significantly altered during peak flow events, and the total annual discharge is often dominated by peak flow events.

p. 8, l. 4-6: The principle of the method should be briefly described, despite the references.

p. 8, l. 15: There is no mention of the method by which the 17O signatures of nitrate were determined. This need to be done here or above.

p. 8, l. 24: How do you define error here and elsewhere in the manuscript? Standard error of the mean? Standard deviation? Or else?

p. 9, l. 3-4: "showed NO2−/NO3 − ratios of less than 5%; thus, the results were used with no corrections.": How does that translate in the worst case to uncertainty of the nitrate isotope values?

p. 9, l. 8: "flow-weighted": There is no mention of flow measurements further up in the Materials and Methods section. This needs to be done, and the uncertainty of interpolating nitrate concentrations between four sampling dates only for a whole year needs to be addressed.

p. 9, l. 19: "For small streams with no data for the flow rate, we used a small and stable flow rate of 0.1 m3/s for fn.": For how many of the 33 streams was that the case?

p. 9, l. 21f.: The calculation of the $\delta$15N and $\delta$18O values of remineralized nitrate with a two end-member mixing model with atmospheric nitrate as second end member falls short of taking into account also other sources of nitrate, e.g. fertilizer or sewage water.

p. 10, l. 17: What is a "clear normal correlation"? Please specify.

p. 11, l. 6-7: "The present results imply seasonal and regional changes in the $\delta$18O/$\Delta$17O ratios of tropospheric ozone and in the OH radical.": Are there any references that back up this assumption?

p. 11, l. 8: "On the basis of both the temporal variation in the depositional flux of $NO_3-atm...$": No temporal/seasonal variation of the depositional nitrate flux has been described further up, and in Fig. 3c there is no clear seasonal pattern of the depositional nitrate flux visible, in contrast to the $\Delta17O$ values.

p. 11, l. 18-19: "additional corrections could be needed": Were they required? And if yes, how exactly were these corrections done?

p. 11, l. 19-20 and 24: What do you mean with "NOx oxidation channel"? Pathway?

p. 12, l. 3: "correct for difference in arrival frequency ": What do you mean with "difference in arrival frequency? Please rephrase in an understandable way. And has it been corrected for in the present work?

p. 12, l. 9-12: This statement is too vague and weak. It needs to be backed up with literature, or it should be abandoned.

p. 12, l. 10: "by allowing an appropriate range of errors presented later": This "range of errors should be specified here at its first mention.

p. 12, l. 14-18: The residence time of atmospheric nitrate could vary significantly between your different catchments with different land uses. How do you know whether the residence time was similar in all of your catchments to that of forested catchments reported elsewhere?

p. 12, l. 21: "we used the obtained $\Delta17O_{avg}$": At this stage it is not clear how the $\Delta17O_{avg}$ was obtained.

p. 12, l. 22-23: "...by allowing the error range of 3.0‰ considering the whole factor change of $\Delta17O_{atm}$ from $\Delta17O_{avg}$.": What does that mean? Please describe in an understandable way. Why exactly 3.0‰ and not 2‰ 1‰ or any other value?

p. 12, l. 28-30: "As a result, while using the $\delta15N_{avg}$ and $\delta18O_{avg}$ values as $\delta15N_{atm}$ and $\delta18O_{atm}$, we assumed much larger error range on the values; i.e. $\pm$ 10‰ for both
$\delta15N$ and $\delta18O$." Unclear, how this error was determined. Please describe in more detail.

p. 13, l. 10-12: "The spatially continuous variation in the values... imply that the values may represent land use changes in each catchment area.": The annual average values of $\delta15N$ and $\delta18O$ vary by 10‰ at the most. Given the uncertainty range of these values of +/- 10‰ (as stated on p. 12, l. 30), how do you want to discern any significant differences here, not to mention to derive any statements about land-use effects on the processing of atmospheric nitrate in the different catchments?

p. 14, l. 24: "determined recently": By whom? No reference provided.

p. 15, l. 2-4: "We concluded that the $\delta18O$ value of NO3−re produced through nitrification in the temperate watershed having $\delta18O(H2O)$ values of $-7.8 \pm 1.0$‰ was $-2.9 \pm 1.2$‰ and that we should use such a low $\delta18O$ value...": Did the soil and/or stream water have this $\delta18O(H2O)$ values of $-7.8 \pm 1.0$‰ If yes, please make this clear in this sentence. If not, then the basis for this conclusion is not clear.

p. 15, l. 8: "Although the $\Delta17O$ values of nitrate were stable during the biogeochemical processing": Again, if nitrate is biogeochemically processed, then also the $\Delta17O$ gets lost.

p. 15, l. 13-16: "We concluded that the range of isotopic fractionations . . . was generally small": The basis of this conclusion remains unclear. Please explain in more detail.

p. 15, l. 16-19: "This result also supports our assumption in section 3.1 such that the actual $\delta15N$ and $\delta18O$ values of NO3−atm in each stream water sample . . . correlate with the $\delta15Navg$ and $\delta18Oavg$ estimated at Sado-seki monitoring station within an error of $\pm10$‰'": This refers to the previous sentence, which does not report a result but a conclusion, the basis of which remained unclear. That is, the statement made in section 3.1 has been based on very weak grounds.

p. 15, l. 30-31: ". . .responsible for the positive correlation between the $\delta15N$ values of total nitrate and population density.": Was this correlation significant? I could not find any statistical information.

p. 17, l. 1-3: "...the slight deviations in the reported $\delta$15N and $\delta$18O values from our results can be explained by the following factors...": Could also different sources of ammonium for nitrification could have played a role (soil, sewage water, fertilizer)?

p. 18, l. 17f.: It is unclear whether this snow signal of atmospheric nitrate could be captured by the sampling design of only four samplings per year.

p. 23, l. 3: "The estimated annual average $\Delta$17O value of inflows, +1.3‰...": Unclear, where this value comes from. Please explain.

p. 23, l. 4: "...average mixing ratio of NO3−atm within total nitrate of 5.1 ± 0.5%...": This value shows only up here and in the abstract, but it is unclear how and when it was calculated.

p. 23, l. 6-7: "...the remainder of the nitrate was of remineralized origin (NO3−re) likely produced through nitrification within the catchments...": Again, what about direct input of nitrate via fertilizer and/or sewage water without remineralization?

p. 23, l. 25: "Lake Biwa also acts as a net sink for fixed N": The question is what happens with the processed nitrate? Very likely most of it is denitrified and lost to the atmosphere as N2O and/or N2. Thus, the statement that Lake Biwa acts as a net sink for fixed N is questionable.

Table 1: This table should also include the dominating land use in the respective catchment.

Please also note the supplement to this comment:
http://www.biogeosciences-discuss.net/bg-2015-627/bg-2015-627-RC2-supplement.pdf

[Figure]

**Supplement:**

[revised manuscript text omitted]

---

## Author Comment (AC1) · 21 Apr 2016

Reply to your comment (Referee #1).

Thank you very much for your valuable comments on our manuscript. We would like to reply by responding to each of your comments and questions.

> The presentation quality (except for the figures) is mostly good, but there are a number of grammatical errors. Language copy-editing will be required.

The English of the manuscript was thoroughly edited by Editage English editing service

(http:// www.editage.jp/) prior to initial submission. We intend to have them edit the English again prior to submission of the revised manuscript.

> In line with the Biogeosciences data policy, all data (water fluxes, nitrate and nitrite concentrations, isotope values, etc.) should be publicly available, preferably by deposition in a data repository, or alternatively as electronic supplementary information. Please add a section on "Data availability".

We would like to present all the data in supplementary tables of the revised MS.

> The figures are somewhat antediluvian in their appearance due to the lack of colours. While the stated aim to make them compatible with black and white printer is commendable, most readers will either view them on a colour display or printed in colour. The figures should be redrawn in colour. The current use of different shapes to distinguish time series can be retained, for the benefit of the small number of readers without access to colour displays and printers, and for those with impaired colour vision.

Although, we cannot agree with you on this point, we would like to add colour to the figures during revision in response to your request.

> The figures in the supplementary information are discussed at crucial points and called out in the text. Fragmentation of the text into different documents is undesirable. They should be merged with the main text. The supplementary information should be used instead to present the full dataset in a table (water fluxes, nitrate and nitrite concentrations, isotope values, etc.), unless the authors can give a URL or DOI at a repository, at which they have lodged the data.

We would like to move Fig. S1 from the supplementary information to the main text in the revised manuscript, in response to your request.

> The treatment of systematic uncertainties and propagation of measurement uncertainties to derived properties is insufficient and/or has been insufficiently well presented. In particular, variations in the isotopic composition of the atmospheric nitrate

end member (delta17O, delta15N and delta18O) should be better documented and listed in a table. The resultant uncertainties in the calculation of the atmospheric nitrate fraction and the delta15N and delta18O values of the remineralised nitrate should be included in the text and figures of the paper.

The measurement uncertainties in the isotopic composition of nitrate (delta15N, delta18O, and Delta17O) in each sample were presented in Section 3.1 (L24–25/P8). The uncertainties in the isotopic composition of the atmospheric nitrate end member (delta15Natm, delta18Oatm, and Delta17Oatm) were presented and discussed in detail in Section 3.1 (from L17/P11 to L2/P13) and 3.3 (L14–18/P14).

The uncertainties in both absolute concentration of atmospheric nitrate (Catm) and the isotopic composition of the remineralized nitrate end member (delta15Nre, and delta18Ore) were simply calculated based on the propagation law of the errors, mostly derived from the errors in the values of delta15Natm, delta18Oatm, and Delta17Oatm in equations (2), (6) and (7). We would like to emphasize this in the revised MS.

The uncertainties in the data points were presented in the figures as well. The data points without error bars corresponded to errors smaller than the symbols. The only exception was Fig. 5, simply because the figure would be too complicated if we added error bars. We would like to emphasize this in the figure caption. Besides, we would like to add the errors in the tables (especially for those in supplement) of the revised MS in response to your request.

> A major systematic uncertainty arises from the current disregard of nitrate sinks and associated isotopic fractionation (section 3.5). Specifically, the delta15N and delta18O values of so-called "remineralised nitrate" may have undergone large modifications due to nitrate assimilation or denitrification. These processes have been currently ignored. While their effect on delta17O is most likely negligible, they cannot be discounted for delta15N and delta18O. In other words, assimilation and denitrification could have enriched "remineralised nitrate" in both 15N and 18O isotopes. The possibility of this

should be explored further, perhaps in the first instance using delta15N and delta18O scatter plots of the "remineralised nitrate" component. If there is a noticeable potential influence, this should be quantified as good as possible.

First of all, we did not disregard isotopic fractionations owing to partial removal through either nitrate assimilation or denitrification. Rather, we found significant isotopic fractionations owing to partial removal in a few streams in June and August, as presented in Section 3.5.

On the annual average, however, it was difficult to assume significant isotopic fractionations through assimilation or denitrification for the major portion of nitrate eluted from the watershed (see our discussions from L19/P14 to L19/P15), due to the low and uniform delta18O values of remineralized nitrate (i.e., delta18Ore) as presented in Section 3.3.

> The authors may also want to consider renaming the "remineralised nitrate" component as "residual nitrate".

We have been using "remineralized nitrate" since 2010 (Tsunogai et al., Atmos. Chem. Phys., 2010) to indicate nitrate produced through biogeochemical processing (including artificial processing) at surface (such as nitrification) showing Delta17O values almost 0‰.While it was very difficult for us non-native speakers to choose an appropriate English term, we discussed your recommendation for more than a week. Finally, we concluded that "remineralized nitrate" would be better to use, for the following reasons: (1) nitrate other than unprocessed atmospheric nitrate (such as those in soils, agricultural land, manure, and sewage), are mostly "remineralized nitrate" produced through microbial nitrification. (2) Almost the only exception [i.e. not (1)] within "remineralized nitrate" was fertilizer nitrate, which is produced in chemical plants through inorganic oxidation of ammonium and eluted from agricultural land without processing. This cannot be a major portion of "remineralized nitrate" in the streams, at least in Japan, where the rate of fertilizer application to agricultural land is strictly controlled. (3) While the

production reactions of fertilizer nitrate in chemical plants are different from the exact microbial nitrification, oxidation of ammonium can be classified as "nitrification" and the "nitrification" of N2-derived ammonium can be classified as "remineralization": producing inorganic nutrients from the other components. (4) Because we used "remineralized nitrate" again and again to indicate nitrate produced through biogeochemical processing (including artificial processing) at the surface (such as nitrification) showing Delta17O values of almost 0‰ in past studies (Tsunogai et al., 2011; 2014, Nakagawa et al., 2013), we would like to avoid confusion by using a different one here. (5) Because "residual nitrate" has often been used to indicate the residue of denitrification/assimilation reactions in a system, the term "residual nitrate" would be confusing. With all this in mind, we have continued use of the term "remineralized nitrate" with added clarification in the definition presented above.

> The term "17O anomalies" has been criticised for its subjectivity and biased nuances. A more neutral term, which has been adopted by many authors, is "17O excess" (similar to deuterium excess and thermodynamic "excess" properties; a negative "17O excess" reflects a deficit). Alternatively, "delta17O values" would also be acceptable.

We would like to use either "17O excess" or "delta17O values" in the revised MS.

> The term "mixing ratios" is ambiguous and should be replaced by "mole fractions" or "mass fractions".

We would like to use "mole fractions" in the revised MS.

> Averages and uncertainties should be enclosed in round brackets so that the unit applies to both, e.g. p. 2/3 "(2.3±1.1) $\mu$mol L-1", in line with standard practice in natural sciences.

We would like to enclose averages and uncertainties in round brackets in the revised MS, as suggested.

> In line with international conventions, chemical phase information should be added

in round brackets after the chemical formula, i.e. NO3-(atm) and NO3-(re), not as sub-script index. Alternatively, text abbreviations should be used, e.g. AN for atmospheric nitrate and RN for remineralised / residual nitrate.

We would like to use NO3-(atm) and NO3-(re) in the revised MS, as suggested.

> Title: The words "accurate and precise" should be dropped from the title. The claimed "accuracy" would require an absolute measurement method and/or validation by an independent laboratory, neither fo which have been performed. "Precision" is not a meaningful metrological term. In any case, the quantification method has known sig-nificant systematic uncertainties, e.g. the isotopic composition of atmospheric nitrate deposition, temporal undersampling, etc., so neither the qualifications "accurate" nor "precise" apply.

Traditional quantification of atmospheric nitrate using the d18O values of nitrate has been done in many of the past studies. Compared with the traditional method using d18O values of nitrate only, our quantification of atmospheric nitrate using Delta17O was more accurate and more precise. Ohte et al. (2010) studied the same watershed using d18O values of nitrate, for instance, but could not quantify the concentrations of atmospheric nitrate in the streams. We used the words "accurate and precise" so as to differentiate our results from those of past studies.

> Abstract: A sentence should be inserted at the beginning of the abstract that explains the motivation for the present study. The current first sentence of the abstract is too long and should be split into two. Abstract p. 2, lines 14 to 18 should be moved up to before "We conclude ..." in line 4.

We would like to revise the abstract as suggested.

> Introduction: The discussion misses fertiliser inputs and groundwater recharge as nitrate sources - see also p. 12.

The inputs of fertilizer are included in "In addition to natural processes, anthropogenic

processes could have a significant impact on the nitrate dynamics within each catchment area, particularly for those eluted from urban or agricultural catchment zones (L29–31/P2)". We would like to revise this part to avoid misleading readers.

While groundwater is an important reservoir for nitrate, the nitrate in the groundwater had been derived from either internal microbial nitrification or addition from external sources including anthropogenic (fertilizer, manure, sewage etc.). Because all these were taken into account as the sources of nitrate in streams, we did not rate "the groundwater recharge/discharge" as the source of nitrate in this text to avoid double counting the sources. We would like to revise this part to avoid misleading readers.

> 4/9: Use of beta = 0.5247 is unusual. Most other research groups use 0.528, based on the meteoric water relationship (e.g. Savarino, Hastings, Michalski). The references given by the authors either do not express a preference (Miller 2002) or also use a coefficient of nearly 0.528 (Kaiser et al. 2007).

We are very sorry but 0.5247 was a typographic error. We changed the beta of nitrate in our calculation from 0.516 to 0.5279 in 2008, but typed 0.5247 in text. Thank you for pointing this out. We would like to revise this part.

> 4/12: This statement only applies to mass-dependent fractionation processes following a slope of 0.5247. It also does not apply to mixing applications because mixing follows linear, not power-law relationships. As in the present case, the authors are primarily using delta17O as a mixing tracer, a linear definition of delta17O would be preferable.

We cannot agree with you on this point. First of all, if either definition would be used, the difference in the final results would be much smaller in the case of nitrate. As a result, we would prefer to use the same definition as used in past studies (Tsunogai et al., 2010; 2011; 2014; Nakagawa et al., 2013) to avoid confusion. Second, atmospheric nitrate in steam nitrate had been isotopically fractionated, around +7‰ on the average (Fig. 6), and much more in some of the samples (Fig. 8). The most important merit

in using Delta17O tracer of nitrate is that we can assume the same Delta17O values during the isotopic fractionation processes, such as partial removal through assimilation/denitrification. The definition of the linear approximation would have abandoned this merit in principle.

Of course we should take into account the non-linear variation during mixing and thus we should correct to that extent. Similar corrections would be needed for the linear definition as well, in principle, by extrapolating the extent of fractionations. We prefer the definition using the power law, assuming appropriate ranges of errors during mixing, as done in this paper.

> At the least, the potential systematic error due to fractionating processes following other mass-dependent relationships (e.g. beta = 0.5) and the effect of linear mixing on the non-linear delta17O definition used here should be quantified and carried forward to the subsequent discussion.

We would like to add this in the revised MS.

> Sections 2.1 and 2.2 should be merged. Sections 2.1.2 and the last paragraph of sections 2.2 are highly repetitive.

We would like to make the suggested revision.

> The use of Japanese "financial years" should be abandoned in favor of the use of calendar years as mandated by the manuscript submission guidelines for Biogeosciences. All dates should be revised accordingly, as well as the calculation of annual averages.

We cannot agree with you to change the starting month for calculating each annual average from April to January. First of all, we could not find such mandate to obligate us to use January as the starting month of statistics in any part of the manuscript submission guidelines for Biogeosciences. Although the reason we started sampling on April was related to management, not science, starting from January has no scientific basis either. Besides, if we started the calculation from January 2010 and ended at

December 2011, the other data (from April to December 2009 and from January to March 2012) would have been excluded from calculation of annual averages without scientific reason. We do not think such statistical treatment is scientifically appropriate. If you do not like the term Japanese "financial year (FY)" with the provided in our paper, please recommend a more appropriate term to express a one-year period from April to March.

> 7/7-14: This discussion is confusing. At the least, it needs to be made clear that there is no 5th sample and that the authors have instead used the data from the previous year (March 2013) to calculate changes over the so-called "winter period", i.e. the difference between October 2013 and March 2013. Rather than using "sampling" numbers, the calculation scheme could perhaps be better presented in a table.

We would like to make the suggested revision.

> 7/28-8/2: The H2O isotope measurements referenced here have not been presented in the manuscript. Please include the results as a figure or table and include the data in the supplementary information.

We would like to add the delta18OH2O data in the supplements of the revised MS.

> 8/3-11: There are too many references given here. Please give just the one that documents the method used here.

We would like to reduce the number of citations, as suggested.

> The azide method is known to require larger isotope exchange corrections for oxygen isotope ratios. How large was the required exchange correction applied to the raw measurements?

The exchange corrections were about 20% (Tsunogai et al., Biogeosciences, 2011).

> 8/12-17: It is not appropriate to use ion chromatography as a reference method to determine the efficacy of N2O conversion. Ion chromatography can also be affected

by measurement biases. Nitrate standards should be used instead to check the azide method is working properly.

The measurement biases in nitrate concentrations using ion chromatography are estimated to be less than 5%, at least for the usual freshwater samples. (We would like to clarify this in the revised MS). Besides, because both the conversion efficiency from nitrate to N2O and the extent of fractionation could be a function of the sample matrix in some cases (Nakagawa et al., Biogeosciences, 2013), it must be better to use the nitrate concentration in each sample determined by Ion chromatography just after each sampling to check the conversion efficiencies, together with the other factors presented.

> Please explain how many samples have been excluded based on the ion chromatographically determined concentrations.

The samples excluded due to lower conversion efficiency were about 10% for all the analyses (measured by one of the graduate school students in our lab). Because this result is a function of the proficiency of the analyst, the value varied.

> 8/24: What "error" do these values represent? Are they the standard deviations of 3 replicate measurements?

They are the standard errors of the mean, in which the error of each single analysis was divided by the square root of the number of analyses. We have clarified this in the revised MS.

> 8/28: Please replace "approximation" by "definition". The definition of delta17O can be made in any arbitrary fashion; there is no approximation in a definition. However, of course, the interpretation of the resultant delta17O values may change, depending on the definition.

We would like to make the suggested revision.

> 9/4: A 5 % contribution of nitrite leads to a significant bias in the delta18O value

and cannot be neglected. The bias could be as large as -1.3 ‰ because the isotopic fractionation associated with conversion of nitrite to N2O is about 27 ‰ lower than for conversion of nitrate to N2O (e.g. Casciotti et al. 2007). All measurements need to be corrected for this bias, or re-analysed after NO2- removal (e.g. following the sulfamic acid protocol of Granger Sigman 2009).

More than 90% of the samples showed NO2- concentrations less than the detection limit (0.05 micro mol/L) and thus showing the NO2-/NO3- ratios less than 0.2%. The NO2- concentrations in the samples that could have NO2-/NO3- ratios more than 1% were also less than the detection limit. Because the NO3- concentrations also were low for the samples (as low as 1 micro mol/L), the possible maximum NO2-/NO3- ratios became 1–5%. As a result, we presented that "all samples showed NO2-/NO3- less than 5%".

Your suggestion to re-analyse after NO2 removal will not be successful because the NO2- was already less than the detection limit (0.05 micro mol/L) in the samples. It is also impossible to correct for NO2- contribution from the isotopic compositions as you suggested, because it is impossible to estimate the actual isotopic compositions of NO2- in the samples.

The NO3–exhausted samples showing nitrate concentrations < 5 micro mol/L were found only in summer (June or August) when the water flow rates were low. As a result, the values of delta15N and delta18O in these samples had little influence on quantifying the flow-weighted annual average isotopic compositions in each river in this study. As a result, we used the results without corrections, as was also done in most of the stable isotope studies of nitrate in freshwater/seawater in the past. We would like to add an explanation of this in the revised MS.

> 9/11-13: It is unclear how the monthly stream flow measurements are used together with the less frequent nitrate concentration and isotope measurements. Do you ignore the flow measurements from the months when no sampling has occurred?

Yes, we used the same flow rate, nitrate concentration, and isotopic compositions for the interval until the next observation. See equations (3) to (5).

> Do you interpolate the concentrations to match each flow measurement with a corresponding nitrate concentration or isotope delta value?

No. We use the same flow rate, nitrate concentration, and isotopic compositions for the interval until next observation. See equations (3) to (5).

> It seems that you ignore major parts of the flow (based on the counting index of 4).

Yes. This is the reason we used the correcting factor alpha in equation (10) when we estimated annual total flux.

> How much stream flow is "missed" due to this temporal under sampling?

Because we used 4 flow data out of 12 (annual), we missed 8. Again, this is the reason we used the correcting factor alpha in equation (10) when we estimated annual total flux.

> 10/13: Please replace "isotopic compositions" with "excess" - see above. "Composition" is not an extensive quantity.

We would like to make the suggested revision.

> 11/13-16: What are uncertainties / variability in the atmospheric end member values? Which values and which uncertainty did you actually use when you calculated the delta values of residual nitrate (see also p. 12, l. 26-31)?

We estimated the uncertainty derived from the difference in the locality as 1 per mil. This was based on the standard deviation between the annual average Delta17O values determined in four different monitoring stations located in the same mid-latitudes, in the past (La Jolla, Princeton, Rishiri, and Sado). Besides, we estimated the uncertainty derived from the seasonal difference in the Delta17O values of atmospheric nitrate as 1.8 per mil, based on the standard deviation of six-month moving averages

of atmospheric nitrate determined at the Sado monitoring station in this study (the six months corresponded to the minimum residence time of water in the watershed). Adding an additional 0.2 per mil as a margin, we adopted 3 per mil as the possible error for Delta17Oatm in the streams. We would like to add an explanation about this in the revised MS.

> Which error do you estimate the systematic neglect of dry deposition has caused?

As already presented in previous papers (Tsunogai et al., 2010; 2014), we regarded the isotopic compositions of nitrate in the wet deposition as those of total deposition because dry deposition occupied less than 20% of the total.

> 14/4: What do you mean by "almost correlated"? Please give a quantitative measure of what "almost correlated" constitutes.

We would like to add to the description of statistical treatments in the revised MS.

> 15/28-31: The last sentence does not make sense at the end of a paragraph. It should appear at the beginning of a new paragraph. You should then explain how you verified that the "remineralised" portion was actually responsible for the positive correlation with delta15N and population density.

What past studies (e.g. Ohte et al., 2010; Mayer et al., 2002) found was a positive correlation between delta15N of total nitrate (= atmospheric + remineralized) and population density. As presented in the Figure 7(c), in which positive correlation ($r^2$ = 0.64) between delta15N values of remineralized nitrate (y-axis) and population density (x-axis) can be found, we verified that the "remineralized" portion was responsible for the positive correlation between delta15N and population density. We would like to add the $r^2$ value (0.64) in the revised MS.

> 16/14: It is unclear why a uniform Catm (atmospheric nitrate contribution to the total nitrate concentration) is indicative for low denitrification. This could be offset of atmospheric inputs of equal magnitude.

Catm is the absolute concentration of atmospheric nitrate in each sample (L20/P4), and may be different from the "atmospheric nitrate contribution to the total nitrate concentration".

In any case, if the denitrification in riverbed sediments were active in streams having high population densities, Catm should be smaller in the streams, while the observed Catm were almost uniform irrespective of the population densities. Of course, the initial Catm could vary between streams. To explain the observed uniform Catm, however, unrealistic assumptions are needed, such as that the initial Catm were higher in accordance with the increase in population densities. To explain both the low and uniform d18O values of remineralized nitrate and the low and uniform Catm, we concluded that nitrate removal through denitrification in riverbed sediments should be minor in the streams. We would like to add an explanation about this in the revised MS.

> A more sensitive approach would be to look for a correlation (or absence there of) between delta15N and delta18O of "remineralised" nitrate.

We trust our approach was adequately sensitive.

> 17/23: delta15N measurements should be used to verify this hypothesis, especially incombination with delta18O (scatter plots).

We also used delta15Nre to verify that assimilation/denitrification was active or not, as suggested by you in our previous paper studying bottled drinking water (Nakagawa et al., 2013). In this study, we did the same but eventually gave it up, because the result was not so clear (low correlation). This was probably because the major factor controlling delta15Nre was not assimilation/denitrification but addition of 15N-enriched nitrate, and that it was active or not in each catchment, especially for those with high population densities, as presented in the previous Section 3.4.

We think the present evidence is sufficient to prove that assimilation/denitrification was responsible for the observed 18O-enrichment in some of the streams. To reduce the

number of figures and length of the text as requested by the other reviewer, we would like to avoid using delta15Nre to verify assimilation/denitrification.

> 22/17: Which value was used for alpha?

The alpha, estimated from equation (9), was 1.9.

> 23/15: Which value was used for gamma?

The gamma, estimated from equation (9), was 1.1.

> 23/25: Please speculate on the fate of nitrogen. Are they lost due to gaseous emissions (denitrification, anammox?) Sedimentary deposition? Eutrophication (secular nitrate concentration increases)?

Of course gaseous emissions should be substantial; otherwise, we should assume an unrealistic non-steady state in the lake. However, I don't think the other reviewer (#2) will allow us to describe such speculations in our paper without data.

> 25/19-25/25: This is not a conclusion, this is the premise/rational of the present paper. These lines should be deleted.

We would like to delete the lines suggested.

> The section Conclusions is incomplete. Seasonal and inter annual changes in stream nitrate concentrations and isotopic composition are missing. Also, the substantive section on the lake nitrogen budget is not reflected by a corresponding conclusion. I would expect a statement on the apparent nitrogen sink and possible identification including any relevant past or future work.

We would like to make the revision suggested.

We would like to thank you for the helpful comments and suggestions. We trust that the answers are satisfactory responses to your comments and questions.

Sincerely, Urumu

Cc: Drs. Takanori Miyauchi, Takuya Ohyama, Daisuke D. Komatsu, Fumiko Nakagawa, Yusuke Obata, Keiichi Sato, and Tsuyoshi Ohizumi

---

## Author Comment (AC2) · 21 Apr 2016

Reply to your comment (Referee #2).

Thank you very much for your valuable comments on our manuscript. We would like to reply by responding to each of your comments and questions.

> While the topic is highly relevant and timely, and the data presented contain valuable information, the paper suffers severely from several weaknesses. Firstly, it is way too long, starting with the Abstract and ending with the Conclusions.

We are very sorry but we do not understand what you want to say here. Compared with 20 other papers published lately in Biogeosciences, this paper could be classified as a shorter one. The style used, "starting with the Abstract and ending with the Conclusions", is the usual style for papers in Biogeosciences.

> The reader gets lost in the many detailed descriptions of results, while the description of methods is partly incomplete. For example, the method how the triple oxygen stable isotope analysis has been done is not described, even not briefly.

As presented in L9-11/P8, "The analytical procedures are the same as those detailed in previous research (Nakagawa et al. 2013; Tsunogai et al. 2014)" so we removed the details. However, we would like to add some brief descriptions on the methods in the revised MS, in response to your request.

> Also statistical and data evaluation methods are not described.

The measurement uncertainties in the isotopic composition of nitrate (delta15N, delta18O, and Delta17O) in each sample were presented in Section 3.1 (L24–25/P8). The uncertainties in the isotopic composition of the atmospheric nitrate end member (delta15Natm, delta18Oatm, and Delta17Oatm) were presented and discussed in detail in Section 3.1 (from L17/P11 to L2/P13) and 3.3 (L14–18/P14).

The uncertainties in both the absolute concentration of atmospheric nitrate (Catm) and the isotopic composition of the remineralized nitrate end member (delta15Nre, and delta18Ore) were simply calculated based on the propagation law of the errors, mostly derived from the errors in the values of delta15Natm, delta18Oatm, and Delta17Oatm in the equations (2), (6) and (7). We would like to emphasize this in the revised MS.

> But most importantly, the conclusions with respect to the effect of different land uses on the fate of atmospheric nitrate (which was the main motivation of the study) were based on many assumptions and uncertain values, especially by excluding more than two sources of nitrate (from atmosphere and from nitrification only).

All the nitrate sources other than atmospheric, such as manure, sewage, and fertilizer, are classified to remineralized nitrate under the definition in this paper, because the oxygen atoms were derived from either terrestrial O2 or H2O through mass-dependent reactions, such as nitrification. We would like to clarify this in Section 1 to avoid misleading readers.

Based on the data of delta18Ore values around -2‰ (vs VSMOW), we concluded that either nitrification in soil or sewage effluent was the major source of the remineralized portion of the nitrate in the streams, as presented in Section 3.4. This is not an assumption because it is based on isotopic composition data.

> Furthermore, there were no statistical data provided that proved an unambiguous relationship between land use in the different stream catchments and the signature of nitrate in the stream water. This weakens the key message of the paper on the effect of land use and population density on the fate of atmospheric nitrate, and needs to be rectified before the paper becomes acceptable for publication.

All the statistical data on land use and population density in the catchments had been provided in the cited reference (Ohte et al., Water Resour. Res., 2010). We do not think the quality of the statistical data were too low to weaken the key message of this paper.

> Technical corrections can be found in the annotated pdf.

Thank you for the helpful suggestions. We would like to correct accordingly.

> Abstract, Results and Discussion, and Conclusions should be shortened significantly, focusing on the main outcome of the paper.

Compared with other papers published lately in Biogeosciences, this paper (9550 words text with 403 words abstract) could be classified as a shorter one. Besides, you and the other reviewer requested us to add much more information to this paper during revision. While we would like to try our best to shorten the revised MS as you

requested, "significant" reduction might be unrealistic.

> It would be good if Lake Biwa were mentioned in the Abstract.

We would like to mention Lake Biwa in the revised MS.

> The number of references should be reduced to about 50 (from almost 80). The number of figures should be reduced to about 6.

Compared with the other papers published lately in Biogeosciences, 80 references and 8 figures were near average. Please tell us the basis for your estimation that 50 references and 6 figures are enough for this paper.

We would like to reduce the number of references in the revised MS as you requested, although 50 might be difficult. We would like to reduce the number of figures in the revised MS as well, although 6 might be difficult.

> Title: I suggest deleting "Accurate and precise" from the title, as it suggests a very high accuracy and precision of the data presented in the paper, which is not the case (the fraction of unprocessed atmospheric nitrate in relation to total nitrate in the stream waters of about 5% had a relative error of 10%, and average 15N and 18O values were assigned with an absolute error of +/- 10‰ which is really large).

Traditional quantification of atmospheric nitrate using the delta18O values of nitrate has been done in many past studies. Compared with the traditional method using delta18O values of nitrate only, our quantification of atmospheric nitrate using Delta17O was more accurate and more precise. Ohte et al. (2010) studied the same watershed using delta18O values of nitrate, for instance, but could not quantify the concentrations of atmospheric nitrate in the streams. Estimating the fraction of unprocessed atmospheric nitrate under a relative error of 10% is much better than the past studies. We would like to use the words "accurate and precise" to differentiate our results from those from past studies.

> Secondly, the reader might wonder why there is a differentiation between accurate

and precise, which occurs also several times in the paper, but which is never explained, also not the way how to achieve both high accuracy and precision.

These were explained in Section 1. We would like to emphasize this in the revised MS.

> Data evaluation and regression methods as well as statistics are not described.

The uncertainties in both absolute concentration of atmospheric nitrate (Catm) and the isotopic composition of the remineralized nitrate end member (delta15Nre, and delta18Ore) were simply calculated based on the propagation law of the errors, mostly derived from the errors in the values of delta15Natm, delta18Oatm, and Delta17Oatm in the equations (2), (6) and (7). We would like to emphasize this in the revised MS.

> The English should be checked by a native speaker.

The English of the manuscript was thoroughly edited by editage English editing service (http:// www.editage.jp/) prior to initial submission. We will have them edit the English again prior to submitting the revised manuscript.

> p. 2, l. 23-24: "important to primary production and thus eutrophication": primary production does not in itself lead to eutrophication, but only a mismatch between primary production and heterotrophic consumption, usually induced by excess nutrient load. I suggest rewording to "important to primary production, and an excess of nitrate can lead to eutrophication downstream".

We would like to make revision suggested.

> p. 2, l. 28-29: I would separate assimilation by plants and microbes and denitrification by microbes in two separate processes, as they are of completely different nature.

We would like to make the revision suggested.

> p. 3, l. 4-10: Here you cite 25 (!) references for one statement, overshooting by far. Please reduce to the 5-6 most important papers.

Most of the references cited here are cited again in later discussions. Our intention was to emphasize that using 15N/14N and 18O/16O ratios of nitrate had been conventional to determine the sources and behaviours of nitrate in stream water, especially for those who are not familiar with stable isotope traces of nitrate. In any case, we would like to reduce the number of citations, as suggested.

> p. 3, l. 10-12: As you use the bold statement ": : :can be quantified through a simple isotope mass balance approach", you should give ranges reported in the literature for the two isotope ratios for the different sources to allow the reader to assess the feasibility of the simple isotope mass balance approach.

Because the ranges were highly variable depending on the literature, it is difficult to specify a range here. Besides, we do not think such a simple approach to isotope mass balance is feasible for 15N/14N and 18O/16O ratios of nitrate as presented in the subsequent sentences. For this reason, we do not think such assessment of feasibility is essential.

> p. 3, l. 22-23: "the mixing ratios of unprocessed NO3-atm within total nitrate are minimum or uniform for whole or specific stream water samples": Meaning of this sentenceis unclear. Please reword.

We would like to reword the sentence, as suggested.

> p. 4, l. 6: "By using the Delta17O signature: : :": This term should be introduced and explained, not only by an equation, but also in words.

We would like to add the requested words.

> p. 4, l.12-13: "In addition, Delta17O is stable during the mass-dependent isotope fractionation processes within surface ecosystems.": Yes, but only if there is no oxygen exchange with the surrounding water, otherwise the Delta17O information gets lost. That is the reason why only UNPROCESSED atmospheric nitrate can be traced, not the further processing of atmospheric nitrate itself.

The oxygen exchange reaction between nitrate and water is unrealistic in the surface ecosystems (e.g. Böhlke et al., RCM, 2003; Kaneko and Kaneko & Poulson, GCA, 2013). Much higher T, as well as both lower pH and higher nitrate concentration, are needed for the progress of the oxygen exchange reaction between nitrate and water. The word 'unprocessed' is used to differentiate atmospheric nitrate from nitrate deposited as atmospheric N originally, but later assimilated by plants or microbes and then been remineralized again within surface ecosystems.

> p. 4, l. 14-15: "Therefore, although the atmospheric 15N or 18O signature can be overprinted by biogeochemical processes subsequent to deposition, Delta17O can be used as a robust tracer: : :": Again, also Delta17O can be "overprinted" by oxygen exchange, not only 15N or 18O of nitrate.

Again, the oxygen exchange reaction between nitrate and water is unrealistic (e.g. Böhlke et al., RCM, 2003; Kaneko and Kaneko & Poulson, GCA, 2013) at least in the watershed. If such oxygen exchange reaction between nitrate and water would be active, the delta18O values of remineralized nitrate should be much higher than observed.

> p. 5, l. 14: "NO3-atm is stable": I disagree. Nitrate from atmospheric deposition can and will be processed after deposition. Therefore, it cannot be considered as stable.

NO3-atm is stable DURING NITRIFICATION. What we discussed here was the variation in both NO3-atm and NO3-re during DURING NITRIFICATION. We would like to emphasize this in the revised MS to avoid misleading readers.

> p. 5, l. 18-20: "Moreover, we exclude the contribution of NO3-atm in the determined 15N and 18O values to estimate the corrected 15N and 18O values for accurate evaluation of the source and behavior of NO3-re.": Totally unclear what that means. Please explain more clearly.

Because nitrates were a mixture of NO3-atm and NO3-re, the raw delta15N and

delta18O values of nitrate (= NO3-atm + NO3-re) were somewhat different from those of NO3-re. By using Delta17O values of nitrate, we could estimate delta15N and delta18O values of NO3-re by excluding the contribution of NO3-atm in the raw 15N and 18O values. We would like to add an explanation about this in the revised MS.

> p. 5, l. 21-22: "influences of flow stagnation into the lake on nitrate": What influence is meant here? On nitrate concentration? On isotope ratios? On total amount?

We meant the influences on the concentrations of both NO3-atm and NO3-re, in streams. We have added an explanation about this in the revised MS.

> p. 6, l. 6-14: This paragraph should be moved to the end of the introduction as part of the motivation for the study.

This paragraph presented past study done in the watershed, not the motivation for this study.

> p. 7, l. 7: "To calculate the annual influx/efflux of nitrate via each stream: : : we used the sampling number n": Unclear how the annual influx/efflux of nitrate was calculated using the sampling number. Please provide a more detailed description of the calculation.

The manner of calculation is presented later in Section 2.4. We have revised this part to avoid misleading readers.

> How were peak flow events after strong precipitation events or after snow melt(if there was) taken into account? Frequently, the solute composition of stream water is significantly altered during peak flow events, and the total annual discharge is often dominated by peak flow events.

As presented in L23/P6 (section 2.2), our estimates on concentrations of nitrate in the streams were based only on those during the base flow periods. The total annual discharge of water determined by Kunimatsu (1995) and cited as Qin in equation (9) in this study; however, includes those during peak flow events in the estimation. Because

nitrate concentrations of streams during base flow periods almost represents that of subsurface runoff in humid temperate climate, as also presented in section 3.1 (L16-20/P12). Subsurface runoff was a major part of the stream flow, including that during peak flow events (e.g., Dincer et al., WRR, 6, 110-124, 1970; Sklash & Farvolden, J. Hydrol., 43, 45-65, 1979; Mcdonnel et al., WRR, 26, 445-458, 1990; McNamara et al., J. Hydrol., 206, 39-57, 1998; Kobayashi, J. Hydrol., 76, 155-162, 1985). Our estimates on annual nitrate discharge (DeltaNin, DeltaNout) included those during the peak flow events within the errors presented. This conclusion is also supported by other independent estimates on the annual nitrate discharge (DeltaNin, DeltaNout) in the lake (Kunimatsu et al., 1995; Tezuka, 1992; Yamada et al., 1996), based on more frequent measurements of nitrate in the streams.

> p. 8, l. 4-6: The principle of the method should be briefly described, despite the references.

We would like to add this in the revised MS.

> p. 8, l. 15: There is no mention of the method by which the 17O signatures of nitrate were determined. This need to be done here or above.

We would like to briefly present this in the revised MS.

> p. 8, l. 24: How do you define error here and elsewhere in the manuscript? Standard error of the mean? Standard deviation? Or else?

We used the standard error of the mean. We would like to clarify this in the revised MS.

> p. 9, l. 3-4: "showed NO2-/NO3- ratios of less than 5%; thus, the results were used with no corrections.": How does that translate in the worst case to uncertainty of the nitrate isotope values?

Because NO2- concentrations were too low to determine the stable isotopic compositions, it is impossible to estimate the worst case uncertainty of the nitrate isotope

values, in principle.

More than 90 % of the samples showed NO2- concentrations less than the detection limit (0.05 micro mol/L) and thus showing the NO2-/NO3- ratios less than 0.2 %. The NO2- concentrations in the samples that could have NO2-/NO3- ratios more than 1 % were also less than the detection limit. Because the NO3- concentrations also were low for the samples (as low as 1 micro mol/L), the possible maximum NO2-/NO3- ratios became 1-5%. As a result, we presented that "all samples showed NO2-/NO3- less than 5%".

The NO3–exhausted samples showing nitrate concentrations < 5 micro mol/L were found only in summer (June or August) when water flow rates were low. As a result, the isotopic compositions in these samples had little influence on quantifying the flow-weighted annual average isotopic compositions in each river in this study. As a result, we used the results without correction, as was also done in most of the stable isotope studies of nitrate in streams in the past. We have added an explanation about this in the revised MS.

> p. 9, l. 8: "flow-weighted": There is no mention of flow measurements further up in the Materials and Methods section. This needs to be done, and the uncertainty of interpolating nitrate concentrations between four sampling dates only for a whole year needs to be addressed.

Most of the flow rate data used in the calculation was cited from a reference (Shiga Prefecture, 2015) as presented in L17-20/P9, so we did not mention the measurements in the Materials and Methods section of this paper.

As presented in Figure S2, our annual average nitrate concentration almost correlated with those determined in past. Besides, our annual average nitrate concentration in major inflow rivers also correlated with those determined by Shiga prefecture based on continuous monitoring at least every month (not presented in this paper). Based on the dispersion of the correlation as presented in Figure S2, we can estimate that

the uncertainty in the annual average concentration in each river is around 10 micro mol/L. Because the annual average nitrate concentrations almost correlated with those determined in other studies; however, the uncertainty in the total flux (DeltaNin) estimated from equation (13) must be minimum. Rather, uncertainty in alpha in (9) must be much larger for the total flux (DeltaNin) so that we did not take into account the uncertainty derived from interpolating nitrate concentrations between four sampling dates for a whole year.

> p. 9, l. 19: "For small streams with no data for the flow rate, we used a small and stable flow rate of 0.1 m3/s for fn.": For how many of the 33 streams was that the case?

These estimates were applied for 13 streams. We would like to clarify this in the revised MS.

> p. 9, l. 21f.: The calculation of the 15N and 18O values of remineralized nitrate with a two end-member mixing model with atmospheric nitrate as second end member falls short of taking into account also other sources of nitrate, e.g. fertilizer or sewage water.

We did take into account the other sources of nitrate as the source of remineralized nitrate at this point. All the nitrate sources other than atmospheric, such as manure, sewage, and fertilizer, are classified to remineralized nitrate under the definition in this paper, because the oxygen atoms are derived from either terrestrial O2 or H2O through mass-dependent reactions, such as nitrification. We would like to clarify this in Section 1 to avoid misleading readers.

> p. 10, l. 17: What is a "clear normal correlation"? Please specify.

We would like to add the $r^2$ value in the revised MS.

> p. 11, l. 6-7: "The present results imply seasonal and regional changes in the delta18O/Delta17O ratios of tropospheric ozone and in the OH radical.": Are there any references that back up this assumption?

None of which we are aware.

> p. 11, l. 8: "On the basis of both the temporal variation in the depositional flux of NO3-atm: : :": No temporal/seasonal variation of the depositional nitrate flux has been described further up, and in Fig. 3c there is no clear seasonal pattern of the depositional nitrate flux visible, in contrast to the Delta17O values.

The temporal variation in the depositional flux of NO3-atm was presented in Fig. 3(c). We would like to clarify this in the revised MS.

The seasonal patterns are indicated in the isotopes. We did not claim that we could find clear seasonal patterns in the depositional nitrate flux.

> p. 11, l. 18-19: "additional corrections could be needed": Were they required? And if yes, how exactly were these corrections done?

They were presented in subsequent sentences, until L25/P12.

> p. 11, l. 19-20 and 24: What do you mean with "NOx oxidation channel"? Pathway?

Yes, we mean a NOx oxidation pathway.

> p. 12, l. 3: "correct for difference in arrival frequency ": What do you mean with "difference in arrival frequency? Please rephrase in an understandable way.

We would like to rephrase this part.

> And has it been corrected for in the present work?

This was presented in subsequent sentences, until L25/P12.

> p. 12, l. 9-12: This statement is too vague and weak. It needs to be backed up with literature, or it should be abandoned.

Why? Four of the four annual observations in mid latitudes reported to the present (La Jolla, Princeton, Rishiri, and Sado) coincided within 2‰ mutual differences.

> p. 12, l. 10: "by allowing an appropriate range of errors presented later": This "range of errors should be specified here at its first mention.

We would like to add this in the revised MS.

> p. 12, l. 14-18: The residence time of atmospheric nitrate could vary significantly between your different catchments with different land uses. How do you know whether the residence time was similar in all of your catchments to that of forested catchments reported elsewhere?

They were already discussed in the references presented.

> p. 12, l. 21: "we used the obtained Delta17Oavg": At this stage it is not clear how the Delta17Oavg was obtained.

They were presented on P11. We would like to clarify this in the revised MS.

> p. 12, l. 22-23: ": : :by allowing the error range of 3.0‰ considering the whole factor change of Delta17Oatm from Delta17Oavg.": What does that mean? Please describe in an understandable way. Why exactly 3.0‰ and not 2‰ 1‰ or any other value?

We estimated the uncertainty derived from the difference in the locality as 1 per mil, based on the standard deviation between the annual average Delta17O values determined in four different monitoring stations located in the same mid-latitudes in the past (La Jolla, Princeton, Rishiri, and Sado). Besides, we estimated the uncertainty derived from the seasonal difference in the Delta17O values of atmospheric nitrate as 1.8 per mil, based on the standard deviation of 6-month moving averages of atmospheric nitrate determined at Sado monitoring station in this study (the six months corresponded to the minimum residence time of water in the watershed). Adding a further 0.2 per mil for a margin, we adopted 3 per mil as the possible error for Delta17Oatm in the streams. We would like to add an explanation about this in the revised MS.

> p. 12, l. 28-30: "As a result, while using the 15Navg and 18Oavg values as 15Natm and 18Oatm, we assumed much larger error range on the values; i.e. 10‰ for both 15N and 18O." Unclear, how this error was determined. Please describe in more detail.

The possible errors of both delta15Natm and delta18Oatm were originally set as twice

the enrichment factor during assimilation of nitrate (ca. +5 per mil; Granger et al., 2010). As a result, nitrate concentration must be reduced to $1/e^2$ the original when delta values increased +10 per mil from the original. If the atmospheric nitrate concentration were reduced to $1/e^2$ of the original, it is very difficult to detect. Of course this estimation is less reliable, so that we verified its appropriateness later in Section 3.3. We have added an explanation about this in the revised MS.

> p. 13, l. 10-12: "The spatially continuous variation in the values: : : imply that the values may represent land use changes in each catchment area.": The annual average values of 15N and 18O vary by 10‰ at the most. Given the uncertainty range of these values of +/- 10‰ (as stated on p. 12, l. 30), how do you want to discern any significant differences here, not to mention to derive any statements about land-use effects on the processing of atmospheric nitrate in the different catchments?

The possible errors (10‰ were that of atmospheric nitrate, not remineralized nitrate. Because atmospheric nitrate occupied only about 5% of total nitrate on the average, the variation range was reduced to 0.5‰ much smaller than the regional variations.

> p. 14, l. 24: "determined recently": By whom? No reference provided.

The references were provided in the subsequent sentences, from L24/P14 to L28/P14. We would like to rewrite this part to avoid misleading readers.

> p. 15, l. 2-4: "We concluded that the 18O value of NO3-re produced through nitrification in the temperate watershed having 18O(H2O) values of -7.8 $\pm$ 1.0‰ was -2.9 $\pm$ 1.2‰ and that we should use such a low 18O value: : :": Did the soil and/or stream water have this delta18O(H2O) values of -7.8 $\pm$ 1.0‰ If yes, please make this clear in this sentence. If not, then the basis for this conclusion is not clear.

The delta18O value was the average of the delta18O(H2O) value of the streams studied in this paper. We would like to clarify this in the revised MS.

> p. 15, l. 8: "Although the Delta17O values of nitrate were stable during the biogeochemical processing": Again, if nitrate is biogeochemically processed, then also the Delta17O gets lost.

Not just "during the biogeochemical processing" but "during the biogeochemical processing such as partial removal through assimilation or denitrification (L8-9/P15)". We would like to clarify this in the revised MS.

> p. 15, l. 13-16: "We concluded that the range of isotopic fractionations : : : was generally small": The basis of this conclusion remains unclear. Please explain in more detail.

If such isotopic fractionations were significant for the portion of atmospheric nitrate in total nitrate, the data should be plotted on the 18O-enriched side in Figure 6, especially for those enriched in atmospheric nitrate (i.e. those showing high Delta17O values). We would like to emphasize this in the revised MS.

> p. 15, l. 16-19: "This result also supports our assumption in section 3.1 such that the actual 15N and 18O values of NO3-atm in each stream water sample : : : correlate with the 15Navg and 18Oavg estimated at Sado-seki monitoring station within an error of +/-10‰Ź": This refers to the previous sentence, which does not report a result but a conclusion, the basis of which remained unclear. That is, the statement made in section 3.1 has been based on very weak grounds.

"The statement made in section 3.1 has been based on very weak grounds" was the reason we verified the appropriateness here in Section 3.3.

> p. 15, l. 30-31: ": : :responsible for the positive correlation between the 15N values of total nitrate and population density.": Was this correlation significant? I could not find any statistical information.

We would like to add the r^2 value (0.64) in the revised MS.

> p. 17, l. 1-3: ": : :the slight deviations in the reported 15N and 18O values from our results can be explained by the following factors: : :": Could also different sources of

ammonium for nitrification could have played a role (soil, sewage water, fertilizer)?

If your comment meant the contribution of 15N-poor ammonium for nitrification on the "sewage water" nitrate reported in this study, it could be possible. However, please note that what we insisted here was the observed differences in the values of delta15N and delta18O were insignificant. The explanation for the observed differences presented here (contribution of NO3-atm and progress of denitrification) was just one the most possible explanation. The detailed discussions on the differences should be done elsewhere, adding the Delta17O data of sewage water nitrate.

> p. 18, l. 17f.: It is unclear whether this snow signal of atmospheric nitrate could be captured by the sampling design of only four samplings per year.

In the watershed studied, monitoring both the nitrate concentration and the flow rate has been done for the major streams, periodically at least every month. This is the reason we chose March, June, August and November for the samplings. One of the reasons we chose March was to capture the snow signal season (usually from February to April), when the flow rates were at maximum every year.

> p. 23, l. 3: "The estimated annual average Delta17O value of inflows, +1.3‰ : : :": Unclear, where this value comes from. Please explain.

The annual average Delta17O value of inflows was estimated using equation (15). We would like to clarify this in the revised MS.

> p. 23, l. 4: ": : :average mixing ratio of NO3-atm within total nitrate of 5.1+/-0.5%...": This value shows only up here and in the abstract, but it is unclear how and when it was calculated.

The annual average mixing ratio of NO3-atm within total nitrate was estimated from the annual average Delta17O value of inflows (+1.3‰ using equation (2). We would like to clarify this in the revised MS.

> p. 23, l. 6-7: ": : :the remainder of the nitrate was of remineralized origin (NO3-

re) likely produced through nitrification within the catchments: : :": Again, what about direct input of nitrate via fertilizer and/or sewage water without remineralization?

As already presented, all the nitrate sources other than atmospheric, such as manure, sewage and fertilizer, are classified to remineralized nitrate under the definition in this paper, because the oxygen atoms are derived from either terrestrial O2 or H2O through mass-dependent reactions, such as nitrification. We would like to clarify this in Section 1 to avoid misleading readers.

> p. 23, l. 25: "Lake Biwa also acts as a net sink for fixed N": The question is what happens with the processed nitrate? Very likely most of it is denitrified and lost to the atmosphere as N2O and/or N2. Thus, the statement that Lake Biwa acts as a net sink for fixed N is questionable.

Why ?? The fixed-N (= nitrate + ammonium + organic-N, etc.) removal to the atmosphere as N2 through denitrification is no doubt a sink for fixed-N, because N2 is not fixed-N anymore. Because the total fixed-N outflux via streams was less than total fixed-N influx via streams in Lake Biwa as presented in L10-27/P23, Lake Biwa did act as a net sink for fixed-N.

> Table 1: This table should also include the dominating land use in the respective catchment.

We would like to add this in the revised MS.

We would like to thank you for the helpful comments and suggestions. We trust that the answers are satisfactory responses to your comments and questions.

Sincerely, Urumu

Cc: Drs. Takanori Miyauchi, Takuya Ohyama, Daisuke D. Komatsu, Fumiko Nakagawa, Yusuke Obata, Keiichi Sato, and Tsuyoshi Ohizumi

---

## Author Response (AR1)

May 19, 2016

1

Dr. Roland Bol Editor of Biogeosciences

Title: Accurate and precise quantification of atmospheric nitrate in streams draining land of various uses by using triple oxygen isotopes as tracers Authors: U. Tsunogai, T. Miyauchi, T. Ohyama, D.D. Komatsu, F. Nakagawa, Y. Obata, K. Sato, and T. Ohizumi MS No.: bg-2015-627

Dear Prof. Bol:

Thank you very much for your decision on our manuscript (Publish subject to minor revisions). We have revised the manuscript accordingly. All the revisions from the previous version sent to you on May 11 are as follows:

1) The English of the manuscript was thoroughly edited by Editage English editing service (http:// www.editage.jp/) again.

2) We used red instead of pink in Figures 6 and 7.

3) We maximized the axes in Fig 8; used -1 to +5 instead of -2 to +8 for x-axis and -5 to +15 instead of -10 to +30 for y-axis.

4) We made minor improvements on the figures.

5) We added present address of Mr. Y. Obata.

Please find the revised manuscript uploaded, in which the newly added or changed sentences from BGD were presented in blue/green.

We would like to thank you and referees for the helpful comments and suggestions. We trust that the revision is satisfactory response to your comments. Thank you for your consideration.

Sincerely yours, Urumu Tsunogai, PhD

Professor Graduate School of Environmental Studies, Nagoya University Furo-cho, Chikusa-ku, Nagoya,

**464-8601, JAPAN Phone: +81-11-789-3498 E-mail: urumu@nagoya-u.jp**

Encl.

c.c. Drs. U. Tsunogai, T. Miyauchi, T. Ohyama, D.D. Komatsu, F. Nakagawa, Y. Obata, K. Sato, and T. Ohizumi

**Accurate and precise quantification of atmospheric nitrate in streams draining land of various uses by using triple oxygen isotopes as tracers**

4

5 Urumu Tsunogai1, Takanori Miyauchi1, Takuya Ohyama1, Daisuke D. Komatsu1,\*,
 6 Fumiko Nakagawa1, Yusuke Obata2,\*\*, Keiichi Sato3, and Tsuyoshi Ohizumi3,\*\*\*

- 7
- 8 [1]{Graduate School of Environmental Studies, Nagoya University, Furo-cho, Chikusa-ku,
  9 Nagoya 464-8601, Japan}
- 10 [2]{Faculty of Bioresources, Mie University, 1577 Kurimamachiya-cho, Tsu 514-8507,
  11 Japan}
- 12 [3]{Asia Center for Air Pollution Research, 1182 Sowa Nishi-ku, Niigata 950-2144, Japan}
- 13 [\*] {now at School of Marine Science and Technology, Tokai University, 3-20-1 Orito,
- 14 Shimizu, Shizuoka 424-8610, Japan}
- 15 [\*\*] {now at Graduate School of Environmental Studies, Nagoya University, Furo-cho,
- 16 Chikusa-ku, Nagoya 464-8601, Japan}
- 17 [\*\*\*] {now at Niigata Prefectural Institute of Public Health and Environmental Sciences, 314-
- 18 1, Sowa, Niigata, Niigata 950-2144, Japan}
- 19 Correspondence to: U. Tsunogai (urumu@nagoya-u.jp)
- 20

**21 Abstract**

Land use in a catchment area has significant impacts on nitrate eluted from the catchment, including atmospheric nitrate deposited onto the catchment area and remineralized nitrate produced within the catchment area. Although the stable isotopic compositions of nitrate eluted from a catchment can be a useful tracer to quantify the land use influences on the sources and behaviour of the nitrate, it is best to determine these for the remineralized portion of the nitrate separately from the unprocessed atmospheric nitrate to obtain a more accurate and precise quantification of the land use influences. In this study, we determined the spatial

distribution and seasonal variation of stable isotopic compositions of nitrate for more than 30 1 2 streams within the same watershed, the Lake Biwa watershed in Japan, in order to use 17O excess ( $\Delta^{17}$ O) of nitrate as an additional tracer to quantify the mole fraction of atmospheric 3 nitrate accurately and precisely. The stable isotopic compositions, including  $\Delta^{17}$ O of nitrate, 4 in precipitation (wet deposition; n = 196) sampled at the Sado-seki monitoring station were 5 also determined for three years. The deposited nitrate showed large 17O excesses similar to 6 those already reported for mid-latitudes:  $\Delta^{17}$ O values ranged from +18.6‰ to +32.4‰ with a 7 three-year average of +26.3‰. On the other hand, nitrate in each inflow stream showed small 8 annual average  $\Delta^{17}$ O values ranging from +0.5% to +3.1%, which corresponds to mole 9 fractions of unprocessed atmospheric nitrate to total nitrate from  $(1.8\pm0.3)\%$  to  $(11.8\pm1.8)\%$ , 10 respectively, with an average for all inflow streams of  $(5.1\pm0.5)$ %. Although the annual 11 average  $\Delta^{17}$ O values tended to be smaller in accordance with the increase in annual average 12 stream nitrate concentration from 12.7 to 106.2  $\mu$ mol L-1, the absolute concentrations of 13 unprocessed atmospheric nitrate were almost stable at  $(2.3\pm1.1)$  µmol L-1 irrespective of the 14 changes in population density and land use in each catchment area. We conclude that changes 15 in population density and land use between each catchment area had little impact on the 16 17 concentration of atmospheric nitrate and that the total nitrate concentration originated 18 primarily from additional contributions of remineralized nitrate. By using the average stable 19 isotopic compositions of atmospheric nitrate, we excluded the contribution of atmospheric nitrate from the determined  $\delta^{15}N$  and  $\delta^{18}O$  values of total nitrate and estimated the  $\delta^{15}N$  and 20 21  $\delta^{18}$ O values of the remineralized portion of nitrate in each stream to clarify the sources. We found that the remineralized portion of the nitrate in the streams could be explained by mixing 22 between a natural source having values of  $(+4.4\pm1.8)$ % and  $(-2.3\pm0.9)$ % for  $\delta^{15}$ N and  $\delta^{18}$ O, 23 respectively, and an anthropogenic source having values of  $(+9.2\pm1.3)$ % and  $(-2.2\pm1.1)$ % 24 for  $\delta^{15}N$  and  $\delta^{18}O$ , respectively. In addition, both the uniform absolute concentration of 25 atmospheric nitrate and the low and uniform  $\delta^{18}$ O values of the remineralized portion of 26 nitrate in the streams imply that in-stream removal of nitrate through assimilation or 27 28 denitrification had small impact on the concentrations and stable isotopic compositions of nitrate in the streams, except for a few streams in summer, having catchments of 29 30 urban/suburban land uses.

**1 **1 Introduction**

Nitrate (NO3-) in stream water can be an important source of information for understanding 2 the biogeochemical cycles within the catchment area of the stream (Likens et al., 1970; Durka 3 4 et al., 1994; Swank et al., 2001). In addition, the nitrate concentration in stream water is important to primary production, and an excess of nitrate can lead to eutrophication in 5 6 downstream areas, including receiving lakes, estuaries, and oceans (McIsaac et al., 2001; Paerl, 2009). However, nitrate concentrations in stream water are determined through a 7 8 complicated interplay of several processes within the catchment area including (1) the 9 addition of atmospheric nitrate ( $NO_3^{-}(atm)$ ) through deposition, (2) the production of remineralized nitrate ( $NO_3$  (re)) through microbial nitrification, (3) the removal of nitrate 10 through assimilation by plants and microbes, and (4) the removal of nitrate through 11 denitrification by microbes. In addition to natural processes, anthropogenic processes can 12 have a significant impact on the sources and dynamics of nitrate within each catchment area, 13 particularly those with urban or agricultural catchment zones. Therefore, interpretation of the 14 processes regulating nitrate concentration in stream water is not always straightforward. 15

The 15N/14N and 18O/16O ratios of nitrate have been widely applied worldwide in the 16 determination of the sources and behaviours of nitrate in stream water (Durka et al., 1994; 17 18 Campbell et al., 2002; Silva et al., 2002; Barnes and Raymond, 2010; Nestler et al., 2011; 19 Lohse et al., 2013). By combining the two isotopic ratios, the relative mole fractions among 20 various nitrate sources such as atmospheric (unprocessed), fertiliser, manure, and sewage plants can be quantified through a simple isotope mass balance approach. Partial removal of 21 nitrate through either assimilation or denitrification, however, results in residual nitrate being 22 enriched with 15N and 18O (Böttcher et al., 1990; Granger et al., 2010), which complicates the 23 24 interpretation of the ratios beyond that of the simple isotope mass balance approach. In 25 addition, trace contributions of unprocessed  $NO_3$  (atm) can have a significant impact on the 18O/16O ratios of the total nitrate in stream water (Durka et al., 1994; Kendall, 1998; Mayer et 26 al., 2001; Michalski et al., 2004; Tsunogai et al., 2010). Therefore, 18O/16O ratios are used as 27 tracers based on assumptions such as (1) the  ${}^{18}O/{}^{16}O$  ratios of nitrate in stream water simply 28 reflect the mole fraction of unprocessed  $NO_3^{-}(atm)$  within total nitrate (Durka et al., 1994; 29 Williard et al., 2001; Ohte et al., 2004; Campbell et al., 2006; Barnes et al., 2008; Burns et al., 30 2009; Ohte et al., 2010; Tobari et al., 2010; Thibodeau et al., 2013; Zeng and Wu, 2015), (2) 31 32 the mole fractions of unprocessed  $NO_3^{-}(atm)$  within total nitrate are minimum for specific

samples (such as soil solution samples) studied (Hales et al., 2007), and (3) the mole fractions 1 2 of unprocessed NO3 (atm) within total nitrate are uniform in the entire samples studied (Wankel et al., 2006; Johannsen et al., 2008). To verify the reliability of these assumptions 3 and to utilise the 18O/16O ratios for quantification of the mole fractions among various nitrate 4 sources based on the isotope mass balance approach, the mole fraction of NO3-(atm) within 5 the total nitrate in stream water must be better understood based on more accurate and more 6 precise quantification rather than on traditional quantification using the  ${}^{15}N/{}^{14}N$  and  ${}^{18}O/{}^{16}O$ 7 8 ratios of nitrate.

To overcome the limitation in using the  ${}^{15}N/{}^{14}N$  and  ${}^{18}O/{}^{16}O$  ratios, the  ${}^{17}O/{}^{16}O$  ratios of 9 10 nitrate have been used as an additional tracer of  $NO_3$  (atm) in stream water in recent studies (Michalski et al., 2004; Tsunogai et al., 2010; Costa et al., 2011; Dejwakh et al., 2012; Riha et 11 al., 2014; Tsunogai et al., 2014; Rose et al., 2015). Because the oxygen atoms of NO3 (re) are 12 derived from either terrestrial O2 or H2O through usual chemical reactions such as 13 nitrification,  $NO_3$  (re) shows mass-dependent relative variations between  ${}^{17}O/{}^{16}O$  and  ${}^{18}O/{}^{16}O$ 14 ratios. On the other hand, only unprocessed  $NO_3^{-}(atm)$  displays an anomalous enrichment in 15 16 17O from the mass-dependent relative variations, reflecting oxygen atom transfers from ozone anomalously enriched in 17O during the conversion of NOx to NO3 (atm) (Michalski et al., 17 18 2003; Morin et al., 2008). By using the  $\Delta^{17}$ O signature (the magnitude of  $^{17}$ O excess) defined by the following equation (Miller, 2002; Kaiser et al., 2007), we can distinguish unprocessed 19 NO3 (atm) ( $\Delta^{17}$ O > 0) from NO3 (re) ( $\Delta^{17}$ O = 0): 20

21
$$\Delta^{17}O = \frac{1 + \delta^{17}O}{\left(1 + \delta^{18}O\right)^{\beta}} - 1,$$
 (1)

where the constant  $\beta$  is 0.5279 (Miller, 2002; Kaiser et al., 2007),  $\delta^{18}O = R_{sample}/R_{standard} - 1$ , and *R* is the 18O/16O ratio of the sample (or the 17O/16O ratio in the case of  $\delta^{17}O$  or the 15N/14N ratio in the case of  $\delta^{15}N$ ) and each standard reference material. Please note that all the nitrate other than the unprocessed NO3 (atm) is classified into NO3 (re), including the nitrate produced through natural/anthropogenic processes in the biosphere/hydrosphere/geosphere and that stored in soil, fertiliser, manure, sewage, etc.

In addition,  $\Delta^{17}$ O is stable during the mass-dependent isotope fractionation processes within surface ecosystems. Therefore, although the atmospheric  $\delta^{15}$ N or  $\delta^{18}$ O signature can be overprinted by biogeochemical processes subsequent to deposition,  $\Delta^{17}$ O can be used as a robust tracer of unprocessed NO3-(atm) to reflect the accurate mole fraction of unprocessed NO3-(atm) within total NO3- regardless of biogeochemical partial removal processes
 subsequent to deposition by using the following equation:

$$\qquad \frac{C_{atm}}{C_{total}} = \frac{\Delta^{17}O}{\Delta^{17}O_{atm}},$$
(2)

4 where  $C_{atm}$  and  $C_{total}$  denote the concentrations of NO3-(atm) and NO3- in each water sample, 5 respectively, and  $\Delta^{17}O_{atm}$  and  $\Delta^{17}O$  denote the  $\Delta^{17}O$  values of NO3-(atm) and nitrate (total) in 6 each water sample, respectively. This is the primary merit of using the 17O/16O ratio as an 7 additional tracer of NO3-(atm).

8 Moreover, additional measurements of the  $\Delta^{17}$ O values of nitrate together with  $\delta^{15}$ N and  $\delta^{18}$ O 9 enable us to exclude the contribution of NO3-(atm) in the determined  $\delta^{15}$ N and  $\delta^{18}$ O values 10 and to estimate the corrected  $\delta^{15}$ N and  $\delta^{18}$ O values ( $\delta^{15}$ Nre and  $\delta^{18}$ Ore, respectively) for 11 accurate evaluation of the source and behaviour of NO3-(re) (Tsunogai et al., 2010; Tsunogai 12 et al., 2011; Dejwakh et al., 2012; Liu et al., 2013; Tsunogai et al., 2014; Riha et al., 2014), 13 including NO3-(re) produced through anthropogenic processes. The details of the calculation 14 are presented in Section 2.5.

Previous studies have successfully applied the  $\Delta^{17}$ O tracer to nitrate eluted from arid/semi-15 16 arid watersheds (Michalski et al., 2004; Dejwakh et al., 2012; Riha et al., 2014), forested 17 watersheds (Tsunogai et al., 2010; Tsunogai et al., 2014; Rose et al., 2015), and a large river basin (Liu et al., 2013) to determine mole fractions of unprocessed NO3 (atm) in total nitrate 18 19 more accurately and precisely than ever before, in addition to the fate of the  $NO_3^{-}(atm)$  that had been deposited into each watershed. However, relative changes in the source and fate of 20  $NO_3$  (atm) in accordance with the changes in land use of catchments have not been studied 21 thus far by using the  $\Delta^{17}$ O tracer of nitrate. 22

23 In this study, we measured the concentrations and the stable isotopic compositions of nitrate including  $\Delta^{17}$ O values for more than 30 streams flowing into a lake in Japan with catchments 24 of widely varying land uses within the same watershed, which includes urban, suburban, 25 agricultural (mostly rice paddies), and forested catchments. By using the  $\Delta^{17}$ O tracer, we 26 quantified both spatial and temporal variations in the concentrations of both  $NO_3^{-}(atm)$  and 27 28  $NO_3$  (re) in streams across the land use settings accurately and precisely to gain insight into the processes controlling the sources, transport, and fate of  $NO_3^{-}(atm)$  and  $NO_3^{-}(re)$  (Fig. 1). 29 Although  $NO_3$  (re) increases during nitrification within each catchment area,  $NO_3$  (atm) is 30

stable during nitrification, so we were able to evaluate the progress of nitrification within each 1 2 catchment area by using the changes in the concentrations of both  $NO_3^{-}(atm)$  and  $NO_3^{-}(re)$ . In addition to those from the streams, we determined the concentrations and the stable isotopic 3 compositions of nitrate including  $\Delta^{17}$ O values in precipitation (wet deposition) for comparison 4 to obtain accurate and precise mole fractions of both NO3-(atm) and NO3-(re) within nitrate 5 (total) in each stream. Moreover, by using both the estimated mole fractions of  $NO_3^{-}(atm)$  in 6 nitrate (total) and the  $\delta^{15}N$  and  $\delta^{18}O$  values of NO3 (atm), we estimated the  $\delta^{15}N_{re}$  and  $\delta^{18}O_{re}$ 7 8 values for accurate evaluation of the source and behaviour of NO3 (re) in streams. Furthermore, we determined the concentrations and the stable isotopic compositions of nitrate 9 including  $\Delta^{17}$ O values in an outflow river of the same lake to evaluate the influences of flow 10 stagnation in the lake on the concentrations of both NO3-(atm) and NO3-(re) by using the 11 differences between inflows and outflows (Fig. 1). The results presented herein increase our 12 13 understanding of the fate of  $NO_3^{-}(atm)$  deposited onto land, particularly the fate of that deposited on urban/suburban and forested catchments (Fig. 1). 14

15

**16 2 Experimental Section**

**17 2.1 Steam water samples**

Lake Biwa, located in the central part of the Japanese Islands, is the largest freshwater lake in Japan (Fig. 2). It has a surface area of 670.4 km2, a total catchment area of 3174 km2, and annual precipitation of around 2000 mm. More than 120 streams flow into the lake, but the Seta River (No. 33 in Fig. 2(b)) at the southern end of the lake, also known as the Yodo River, is the only natural outflow. The average residence time of water in the lake is 5.5 years.

23 Similar to many lakes throughout the world, Lake Biwa has experienced eutrophication in the 24 past. Urbanisation near the lake, beginning in the 1960s, particularly on the southern and 25 eastern shores, likely caused an increase in nutrient loading. Blooms of Uroglena americana and cyanobacteria have occurred since 1977 and 1983, respectively (Hsieh et al., 2011). To 26 clarify the pathways and sources of nitrate that was fed into the lake, the stable isotopic 27 compositions ( $\delta^{15}$ N and  $\delta^{18}$ O) of dissolved nitrate were determined in the major streams 28 flowing into the lake (Ohte et al., 2010). Based on the  $\delta^{15}$ N values of nitrate showing positive 29 30 correlation with the population densities of each catchment area, it was concluded that sewage effluent was the dominant source contributing to the increase in the  $\delta^{15}$ N values of nitrate. 31

In this study, stream water samples were collected near the mouths of 33 inflow streams and 1 1 outflow river (Seta River) of Lake Biwa (Table 1; Fig. 2(b)) during base flow periods four 2 times in 2013, on March 15, June 17, August 5, and October 21, except for stream Nos. 3 and 3 28 in June, which became dry arroyos at that time. The catchments of the studied inflow 4 5 streams occupied 70% of the entire Lake Biwa basin area. The streams were selected to cover those in which the concentrations and stable isotope compositions of nitrate,  $\delta^{15}N$  and  $\delta^{18}O$ , 6 7 had already been determined in 2004–2006 (Ohte et al., 2010). The categories of locations 8 classified by Ohte et al. (2010) were also used in this study to classify the location of each 9 stream (Table 1). Either a bucket or dipper was used to collect samples as far from the bank as 10 possible. Each sample was transferred into a dark polyethylene bottle that was pre-rinsed at 11 least twice with the sample itself and subsequently stored in a refrigerator. Then, the samples 12 were filtered through a pre-combusted Whatman GF/F filter with a 0.7 µm pore size within a 13 few hours after collection, and the filtrate was stored in a different dark polyethylene bottle at 4°C until analysis. 14

15 In this study, we defined the sampling number n, where n = 1, 2, 3, and 4, which represents 16 the sampling in March, June, August, and October, respectively. In addition, we defined one 17 more hypothetical sampling number (n = 5) set just one year later than the n = 1 date. Please 18 note that there are no data for sampling n = 5. Furthermore, we rated the intervals between n = 19 1 and n = 2, n = 2 and n = 3, n = 3 and n = 4, and n = 4 and n = 5 as spring, summer, autumn, 20 and winter, respectively, for the streams in this study.

**21 2.2 Wet deposition samples**

The Sado-seki National Acid Rain Monitoring Station (38°14'59"N, 138°24'00"E) was 22 established on Sado Island (Fig. 2(a)), at 110 m above sea level, as a monitoring observatory 23 24 of the Acid Deposition Monitoring Network in East Asia (EANET) representing the central 25 Japan area (EANET2 2014). Samples of wet deposition were taken at the station by using 26 standard methods for evaluating acid deposition in Japan for the three Japanese financial years (FYs) from April 2009 to March 2012. An automatic wet deposition sampler (US-420, 27 Ogasawara) was used in the collection. All of the deposition samples were stored in 1 L 28 polyethylene bottles under refrigeration until daily recovery. After measuring the volume (i.e. 29 30 precipitation rate), conductivity, and pH, the recovered samples were filtered through a 0.2 µm pore-size membrane filter (Dismic-25CS, ADVANTEC) and stored in a refrigerator until 31 32 analysis.

1 The annual wet deposition rate of nitrate was 19.3 mmol  $m^{-2}y^{-1}$  for FY2009, from April 2009 2 to March 2010; 28.0 mmol  $m^{-2}y^{-1}$  for FY2010, from April 2010 to March 2011; 27.0 mmol 3  $m^{-2}y^{-1}$  for FY2011, from April 2011 to March 2012; and 24.5 mmol  $m^{-2}y^{-1}$  on average from 4 FY2009 to 2011 (EANET, 2014). The annual wet deposition rate of NH4+ was 17.1 mmol 5  $m^{-2}y^{-1}$  on average from FY2009 to 2011 (EANET, 2014).

**6 2.3 Analysis**

The concentrations of nitrate  $(NO_3^-)$  and nitrite  $(NO_2^-)$  in each filtrate sample were measured 7 8 by ion chromatography (Prominence HIC-SP, Shimadzu, Japan) within a few days (stream 9 water samples) and within two weeks (wet deposition samples) after each sampling. The error (standard error of the mean) in the determined concentrations of nitrate was  $\pm 3\%$ . The  $\delta^{18}$ O 10 values of H2O in the samples were analysed using the cavity ring-down spectroscopy method 11 by employing an L2120-i instrument (Picarro Inc., Santa Clara, CA, USA) equipped with an 12 13 A0211 vaporizer and auto sampler; the error (standard error of the mean) in this method was 14 ±0.1‰. Both Vienna Standard Mean Ocean Water (VSMOW) and Standard Light Antarctic Precipitation (SLAP) were used to calibrate the values to the international scale. 15

To determine the stable isotopic compositions, nitrate in each filtrate sample was chemically 16 converted to N2O by using a method originally developed to determine the 15N/14N and 17 18O/16O ratios of seawater and freshwater nitrate (McIlvin and Altabet, 2005) and that was 18 19 later modified (Tsunogai et al., 2008; Konno et al., 2010; Yamazaki et al., 2011). In brief, the procedures were as follows. Approximately 10 mL of each sample solution was pipetted into 20 21 a vial with a septum cap. Then, 0.5 g of spongy cadmium was added, followed by 150 µL of a 1 M NaHCO3 solution. The sample was then shaken for 18–24 h at a rate of 2 cycles/s. Then, 22 the sample solution was decanted into a different vial with a septum cap. After purging the 23 solution using high purity helium, 0.4 mL of the azide/acetic acid buffer was added. After 45 24 min, the solution was made basic by adding 0.2 mL of 6 M NaOH. 25

26 Then, the stable isotopic compositions ( $\delta^{15}$ N,  $\delta^{18}$ O, and  $\Delta^{17}$ O) of N2O in each vial were 27 determined by using a continuous-flow isotope ratio mass spectrometry (CF-IRMS) system at 28 Nagoya University. The analytical procedures using the CF-IRMS system were the same as 29 those detailed in previous research (Komatsu et al., 2008; Hirota et al., 2010). The obtained 30 values of  $\delta^{15}$ N,  $\delta^{18}$ O, and  $\Delta^{17}$ O for N2O derived from the nitrate in each sample were 31 compared with those derived from our local laboratory nitrate standards that had been calibrated using the internationally distributed isotope reference materials to calibrate the values of the sample nitrate to an international scale and to correct for both the isotope fractionation during the chemical conversion to  $N_2O$  and the progress of oxygen isotope exchange between the nitrate-derived reaction intermediate and water (ca. 20%). In this study, we adopted the internal standard method (Nakagawa et al., 2013; Tsunogai et al., 2014) for the calibrations of sample nitrate. All values in this paper are expressed relative to air (for nitrogen) and VSMOW (for oxygen).

To determine whether samples were deteriorated or contaminated during storage and whether 8 the conversion rate from nitrate to N2O was sufficient, concentrations of nitrate in the samples 9 were determined each time we analysed isotopic compositions using CF-IRMS based on the 10  $N_2O^+$  or  $O_2^+$  outputs. We adopted the  $\delta^{15}N$ ,  $\delta^{18}O$ , or  $\Delta^{17}O$  values only when concentrations 11 measured by CF-IRMS correlated with those measured by ion chromatography just after the 12 13 sampling within a difference of 10%. About 10% of the whole isotope analyses showed 14 conversion efficiencies lower than this criterion. Nitrate in these samples was converted to 15 N2O again and re-analysed for the stable isotopic compositions. None of the samples showed significant nitrate deterioration or nitrate contamination during storage. 16

We repeated the analyses of the  $\delta^{15}$ N,  $\delta^{18}$ O, and  $\Delta^{17}$ O values of nitrate for each sample at least 17 three times to attain high precision. Most of the samples had a nitrate concentration of more 18 than 5.0  $\mu$ mol L-1, which corresponded to a nitrate quantity greater than 50 nmol in a 10 mL 19 sample. This amount was sufficient for determining the  $\delta^{15}$ N,  $\delta^{18}$ O, and  $\Delta^{17}$ O values with high 20 precision. For cases of nitrate concentration less than 5.0  $\mu$ mol L-1, the sample volume was 21 increased to 30 mL and the number of analyses was also increased. Thus, all isotopic data 22 presented in this study have an error (standard error of the mean) better than  $\pm 0.2\%$  for  $\delta^{15}N$ , 23  $\pm 0.3\%$  for  $\delta^{18}$ O, and  $\pm 0.1\%$  for  $\Delta^{17}$ O. 24

25 Nitrite  $(NO_2^{-})$  in the samples interferes with the final N2O produced from nitrate  $(NO_3^{-})$ because the chemical method also converts NO2- to N2O (McIlvin and Altabet, 2005). 26 Therefore, it is sometimes necessary to correct for the contribution of  $NO_2^{-}$ -derived N2O to 27 determine the stable isotopic compositions of the sample nitrate accurately. However, in this 28 study, more than 90% of the samples analysed for stable isotopic compositions had NO2- 29 concentrations lower than the detection limit (0.05  $\mu$ mol L-1). Even for the samples having 30  $NO_2^-$  concentrations higher than the detection limit, the  $NO_2^-/NO_3^-$  ratio was less than 1%. 31 Thus, in this study, the results were used with no correction. 32

**2.4 Calculating average concentration and isotopic compositions in each stream**

To clarify the chemical and isotopic characteristics of each stream, we determined both the flow-weighted annual average concentration ( $\overline{C}_{total}$ ) and the flow-weighted annual average  $\delta^{15}N$ ,  $\delta^{18}O$ , and  $\Delta^{17}O$  values ( $\overline{\delta}$ ) of nitrate for each stream assuming the same flow rate, the same nitrate concentration, and the same isotopic compositions for the interval until the next observation by using Eqs. (3), (4), and (5):

8
$$q = \sum_{n=1}^{4} \left( f_n \cdot \Delta t_n \right), \tag{3}$$

9
$$\overline{C}_{\text{total}} = \frac{\sum_{n=1}^{4} (C_n \cdot f_n \cdot \Delta t_n)}{q},$$
 (4)

$$\qquad \overline{\delta} = \frac{\sum_{n=1}^{4} \left( \delta_n \cdot C_n \cdot f_n \cdot \Delta t_n \right)}{\sum_{n=1}^{4} \left( C_n \cdot f_n \cdot \Delta t_n \right)}, \tag{5}$$

11 where  $C_n$  and  $\delta_n$  denote the concentration ( $C_{total}$  in Eq. (2)) and isotopic values ( $\delta^{15}N$ ,  $\delta^{18}O$ , or 12  $\Delta^{17}O$ ) of nitrate in each stream during each observation n, respectively;  $f_n$  denotes the flow 13 rate of each stream during each observation n; and  $\Delta t_n$  denotes the time interval between the 14 observation n and the next observation n+1. When possible, we used the flow rate of each 15 stream that was determined monthly by the Shiga Prefecture (Shiga\_prefecture, 2015) for  $f_n$ . 16 For small streams with no data for flow rate (n = 13), we used a small and stable flow rate of 17 0.1 m3/s for  $f_n$ .

**18 2.5 Calculating $\delta^{15}$ N and $\delta^{18}$ O of remineralized nitrate**

To exclude the contribution of  $NO_3^{-}(atm)$  from the  $\delta^{15}N$  and  $\delta^{18}O$  values of nitrate (total) and to clarify the sources and behaviour of  $NO_3^{-}(re)$  by using both  $\delta^{15}N$  and  $\delta^{18}O$  as tracers, we estimated the end-member  $\delta^{15}N$  and  $\delta^{18}O$  values of the remineralized nitrate portion,  $\delta^{15}N_{re}$ and  $\delta^{18}O_{re}$ , by excluding the contribution of  $NO_3^{-}(atm)$  in nitrate (total) (Tsunogai et al., 2010; Tsunogai et al., 2011; Dejwakh et al., 2012; Liu et al., 2013; Tsunogai et al., 2014; Riha et al., 2014) by using Eqs. (6) and (7):

$$\qquad \delta^{15} N_{re} = \frac{C_{total} \cdot \delta^{15} N - C_{atm} \cdot \delta^{15} N_{atm}}{C_{total} - C_{atm}}, \tag{6}$$

$$\qquad \delta^{18} O_{re} = \frac{C_{total} \cdot \delta^{18} O - C_{atm} \cdot \delta^{18} O_{atm}}{C_{total} - C_{atm}}, \tag{7}$$

where  $C_{atm}$  and  $C_{total}$  denote the concentrations of NO3-(atm) and nitrate (total) in each water 3 sample, respectively, and  $\delta^{15}N_{atm}$ ,  $\delta^{18}O_{atm}$ , and  $\Delta^{17}O_{atm}$  denote the  $\delta^{15}N$ ,  $\delta^{18}O$ , and  $\Delta^{17}O$  values 4 of NO3 (atm) in each sample, respectively. The actual values of  $\delta^{15}N_{atm}$ ,  $\delta^{18}O_{atm}$ , and  $\Delta^{17}O_{atm}$ 5 used in this study were determined from the  $\delta^{15}$ N,  $\delta^{18}$ O, and  $\Delta^{17}$ O values of nitrate in the wet 6 7 deposition samples and are reported in Section 3.1 along with the ranges of errors. Please note that the errors in the estimated values of  $\delta^{15}N_{re}$  and  $\delta^{18}O_{re}$  become larger in accordance with 8 an increase in the  $C_{atm}/C_{total}$  ratio due to the propagation law of errors, even if the errors in the 9 values of  $\delta^{15}N_{atm}$  and  $\delta^{18}O_{atm}$  are the same. 10

**11 **2.6** Possible variations in $\Delta^{17}$ O during the progress of partial removal and 12 mixing**

13 Because we used the power law shown in Eq. (1) for the definition of  $\Delta^{17}$ O, the  $\Delta^{17}$ O values 14 are different from those based on the linear definition (Michalski et al., 2002). Please note that 15 our  $\Delta^{17}$ O values would be (0.1±0.1)‰ higher for the stream water nitrate and (0.9±0.3)‰ 16 higher for the atmospheric nitrate if we had used the linear definition for calculation.

Compared with  $\Delta^{17}$ O values based on the linear definition,  $\Delta^{17}$ O values based on the power 17 law definition are more stable during mass-dependent isotope fractionation processes, so we 18 rated the  $\Delta^{17}$ O values of nitrate as always stable irrespective of any biogeochemical partial 19 20 removal processes subsequent to deposition, such as assimilation or denitrification. On the other hand,  $\Delta^{17}$ O values based on the power law definition are not conserved during mixing 21 processes between fractions having different  $\Delta^{17}$ O values, so the Catm/Ctotal ratios estimated 22 using Eq. (2) are somewhat deviated from the actual  $C_{atm}/C_{total}$  ratios in the samples. However, 23 in this study, the extent of the deviations of the Catm/Ctotal ratios of the stream nitrate was less 24 than 0.15%, so we disregard this effect in the discussion. 25

**1 **3 Results and Discussion**

**3.1 Atmospheric nitrate**

2

3

4

5

6

7

The  $\delta^{15}$ N,  $\delta^{18}$ O, and  $\Delta^{17}$ O of atmospheric nitrate NO3-(atm) are shown in Figs. 3(a–c), respectively, as a function of sampling day (local time, UT +\_9:00), and the daily depositional flux of NO3-(atm) when each of the wet deposition samples was taken is also shown in Fig. 3(c). The daily depositional flux of NO3-(atm) was calculated from the nitrate concentration and the daily precipitation (Sup. Table S2).

The atmospheric nitrate at the Sado-seki monitoring station showed large 17O excesses with 8  $\Delta^{17}$ O values from +18.6‰ to +32.4‰. Moreover, a clear normal correlation between  $\Delta^{17}$ O 9 and  $\delta^{18}$ O was shown ( $r^2 = 0.878$ ) (Fig. 4). A similar trend was reported for atmospheric nitrate 10 aerosols collected for a one-vear period in La Jolla, California (32.7°N, 117.2°W) (Michalski 11 et al., 2003), and similar results also have been obtained in other areas of the world (Kaiser et 12 al., 2007; Morin et al., 2009). Michalski et al. (2003) interpreted that the linear correlation 13 corresponds to the mixing line between tropospheric ozone and tropospheric H2O, and thus 14 tropospheric OH radicals, with  $\Delta^{17}O = 0\%$  and  $\delta^{18}O = -5\%$ . However, the NO3-(atm) data 15 obtained at the Sado-seki monitoring station showed a somewhat different trend in the  $\Delta^{17}O-$ 16  $\delta^{18}$ O plot between summer, from May to October, and winter, from November to April (Fig. 17 4). Although the line fitted to the summer data showed a slope of  $2.21\pm0.22$  and an intercept 18 of (+19.7±5.1)‰ in the  $\Delta^{17}O-\delta^{18}O$  plot, that of the winter data showed a statistically 19 significant larger slope of  $2.89\pm0.38$  and a smaller intercept of  $(+3.0\pm9.2)$ %; all errors were 20 in the 2  $\sigma$  range. Although the winter data included an intercept of -5% reported by 21 Michalski et al. (2003) as the end-member  $\delta^{18}$ O value of the tropospheric OH radical within 22 the possible error range, the intercept of summer data deviated strongly from the value. 23 24 Because the monitoring station is located in the Asian monsoon area, the major air mass that arrived at the station was different seasonally: Pacific air originated from south-east was 25 26 dominant in summer, whereas continental air originated from the north-west was dominant in winter. The present results imply seasonal and regional changes in the  $\delta^{18}O/\Delta^{17}O$  ratios of 27 tropospheric ozone and the OH radical. 28

29 On the basis of both the temporal variation in the daily depositional flux of NO3-(atm), shown 30 in Fig. 3(c), and the  $\Delta^{17}$ O value, we estimated the monthly average  $\Delta^{17}$ O value of NO3-(atm) 1  $(\Delta^{17}O(m))$  deposited at the Sado-seki monitoring station for each month (m) from April 2009 2 to March 2012 by using

3
$$\Delta^{17}O(m) = \frac{\sum_{k} \left( C_{k} \cdot V_{k} \cdot \Delta^{17}O_{k} \right)}{\sum_{k} \left( C_{k} \cdot V_{k} \right)},$$
(8)

4 where  $C_k$  denotes the concentration of nitrate in each wet deposition sample and  $V_k$  denotes 5 the total water volume of each wet deposition sample. Then, we estimated the annual and the 6 three\_year average  $\Delta^{17}O$  values of NO3-(atm) ( $\Delta^{17}O_{avg}$ ) as +25.5‰ for FY2009, +27.2‰ for 7 FY2010, +25.7‰ for FY2011, and +26.3‰ for the three years by using

8
$$\Delta^{17}O_{avg} = \frac{\sum_{m} \left( D(m) \cdot \Delta^{17}O(m) \right)}{\sum_{m} D(m)},$$
(9)

9 where D(m) denotes the monthly wet deposition rate of nitrate at the Sado-seki monitoring 10 station determined by EANET (EANET, 2014). Because no wet deposition sample for 11 measuring stable isotopes was taken in May 2009 or March 2012, we used the  $\Delta^{17}O(m)$ 12 values of May 2010 and March 2011, respectively, for these values. Substituting  $\Delta^{17}O$  with 13  $\delta^{15}N$  ( $\delta^{18}O$ ) in Eqs. (8) and (9), we estimated  $\delta^{15}N_{avg}$  ( $\delta^{18}O_{avg}$ ) as -4.4% (+78.5‰) for 14 FY2009, -3.8% (+81.8‰) for FY2010, -4.4% (+78.6‰) for FY2011, and -4.2% (+79.8‰) 15 for the three years.

To apply the  $\Delta^{17}O_{avg}$  values for the three years obtained at the Sado-seki monitoring station as 16  $\Delta^{17}$ O of NO3 (atm) deposited on the studied watershed (i.e.  $\Delta^{17}$ Oatm in Eq. (2)), additional 17 corrections could be needed because the  $\Delta^{17}$ O value of NO3 (atm) is a function of the NOx 18 oxidation channels in the atmosphere, which shift depending on the intensity of sunlight, 19 20 temperature, and oxidant levels (e.g. Michalski et al., 2003; Morin et al., 2008; Kunasek et al., 2008; Alexander et al., 2009; Morin et al., 2012; Savarino et al., 2013). The latitudinal 21 22 difference between the Sado-seki monitoring station (38°15'N, 138°24'E; Fig. 2) and the 23 watershed studied (35°15'N, 136°5'E; Fig. 2) could change the intensity of sunlight and thus the NOx oxidation channel. Moreover, Tsunogai et al. (2010) reported that nitrate in polluted 24 air masses derived directly from megacities in winter showed slightly larger  $\Delta^{17}$ O values than 25 26 nitrate in background air masses in the same seasons owing likely to the relative increase in the reaction via NO3 radicals within the entire  $NO_3$  (atm) production channel to produce 27 NO3 (atm) in the polluted air mass. The annual average  $\Delta^{17}$ O value determined in this study 28

1 was lowest in FY2009 when the deposition rate of nitrate was the smallest, at 19.3 mmol 2  $m^{-2}y^{-1}$  (EANET, 2014), whereas the annual average  $\Delta^{17}O$  value was highest in FY2010 when 3 the deposition rate was the largest, at 28.0 mmol  $m^{-2}y^{-1}$ , within the three years of observation. 4 These results also imply that the difference in the deposition rate of nitrate must be also 5 corrected to apply the  $\Delta^{17}O_{avg}$  value to  $\Delta^{17}O_{atm}$ .

Nevertheless, both the annual average and the seasonal variation range of  $\Delta^{17}O$  correlated 6 strongly with those determined at the Rishiri monitoring station (45°07'11"N, 141°12'33"E; 7 Fig. 2(a)) in FY2008, at +26.2‰ (Tsunogai et al. 2010), where the wet deposition rate of 8 9  $NO_3$  (atm) was an average of 40% smaller than that at the Sado-seki monitoring station from 2000 to 2013 (EANET, 2014). Moreover, the values also coincided with those reported for 10 mid-latitudes, such as at La Jolla, at 33°N (Michalski et al. 2003) and at Princeton, at 40°N 11 (Kaiser et al. 2007). We concluded that by allowing 1‰ of error, the standard deviation of the 12  $\Delta^{17}O_{avg}$  values determined at the four different monitoring stations located within the same 13 mid-latitude range in the past, the obtained  $\Delta^{17}O_{avg}$  value of NO3-(atm) can be considered 14 representative for middle latitudes worldwide, including the Lake Biwa watershed basin. 15

In addition, the actual  $\Delta^{17}O_{atm}$  values of NO3 (atm) in each stream water sample can differ 16 from the  $\Delta^{17}O_{avg}$  owing to the seasonal variation in the  $\Delta^{17}O$  values of NO3 (atm). In 17 correcting for the seasonal variation, however, it is not adequate to use the  $\Delta^{17}O$  values 18 determined for the seasons of sampling, as  $\Delta^{17}O_{atm}$  in Eq. (2), because the residence time of 19 water is longer than a few months for most of the catchments in Japan with a humid temperate 20 21 climate (Takimoto et al., 1994; Kabeya et al., 2007). That is, the nitrate in base flow stream water had been stored previously in subsurface runoff and groundwater, for which seasonal 22  $\Delta^{17}$ O changes have not been found thus far (Tsunogai et al., 2010; Nakagawa et al., 2013). 23 We concluded that by allowing an additional 1.8% of error, the standard deviation of the six-24 25 month moving average of atmospheric nitrate determined at the Sado monitoring station in this study, the obtained  $\Delta^{17}O_{avg}$  value of NO3 (atm) represented those eluted from the Lake 26 27 Biwa watershed basin.

In summary, we used the three-year average  $\Delta^{17}$ O value of NO3-(atm) obtained at the Sadoseki monitoring station in this study ( $\Delta^{17}O_{avg} = +26.3\%$ ) as the  $\Delta^{17}O_{atm}$  in Eq. (2) to estimate Catm in the streams of the Lake Biwa watershed basin by allowing an error range of 3‰, considering the factor changes of  $\Delta^{17}O_{atm}$  from  $\Delta^{17}O_{avg}$  described above. About 65% of all of 1 the  $\Delta^{17}$ O data of NO3 (atm) obtained at the Sado-seki monitoring station were included in this 2 range of (+26.3±3.0)‰.

In the case of the  $\delta^{15}N$  and  $\delta^{18}O$  of NO3 (atm) in each stream water sample (i.e.  $\delta^{15}N_{atm}$  and 3  $\delta^{18}O_{atm}$  in Eqs. (6) and (7)), the values differed further from  $\delta^{15}N_{avg}$  and  $\delta^{18}O_{avg}$  owing to 4 isotopic fractionation during partial removal subsequent to deposition. As a result, while using 5 the three-year average values of  $\delta^{15}N$  ( $\delta^{15}N_{avg} = -4.2\%$ ) and  $\delta^{18}O$  ( $\delta^{18}O_{avg} = +79.8\%$ ) as 6  $\delta^{15}N_{atm}$  and  $\delta^{18}O_{atm}$ , we assumed much a larger error range in the values, i.e.  $\pm 10\%$  for both 7  $\delta^{15}N$  and  $\delta^{18}O$ , twice the enrichment factor during assimilation of nitrate. Because the 8 concentration of atmospheric nitrate would be reduced to  $e^{-2}$  of the original value if  $\delta^{15}N$  and 9  $\delta^{18}$ O values increased +10% from their original values through assimilation, it might be 10 11 difficult to detect atmospheric nitrate within total nitrate. Of course, this estimation is less 12 reliable, and we further discuss the appropriateness of these error ranges in Section 3.3. Because of the small Catm/Ctotal ratios of stream water of generally less than 7% (Section 3.2), 13 the error propagated to  $\delta^{15}N_{re}$  and  $\delta^{18}O_{re}$  was generally small, less than 1‰ and 2‰, 14 respectively, for most of the data presented in this study (Sup. Table S5). 15

**16 **3.2 Stream nitrate overview**

The concentrations (Ctotal) and the  $\delta^{15}$ N,  $\delta^{18}$ O, and  $\Delta^{17}$ O values of nitrate in the stream water 17 samples determined for each observation (n = 1, 2, 3, and 4) are presented in Figs. 5(a)\_(d). 18 The annual average concentration ( $\overline{C}_{total}$ ) and the annual average  $\delta^{15}N$ ,  $\delta^{18}O$ , and  $\Delta^{17}O$  values ( 19  $\overline{\delta^{15}N}, \overline{\delta^{18}O}$ , and  $\overline{\Delta^{17}O}$ , respectively) in each stream estimated by using Eqs. (3), (4), and (5) are 20 shown in the figure as black bars. In this figure, each stream was plotted on the x-axis in the 21 22 order of location, beginning from stream No. 31, which lies southwest of all of the streams (Fig. 2), and proceeding in a clockwise direction. The errors are comparable to the sizes of the 23 symbols in Figs. 5 (a)\_(d). The spatially continuous variation of the values of  $\overline{\delta^{15}N}$ ,  $\overline{\delta^{18}O}$ , and 24  $\overline{\Delta^{17}O}$  imply that the values may represent land use changes in each catchment area. 25

Although the  $\Delta^{17}$ O values presented significant spatial and temporal variation from +0.0‰ to +6.8‰, the range of the  $\overline{\Delta^{17}}$ O values from +0.5‰ to +3.1‰ was typical for nitrate in natural stream water (Michalski et al., 2004; Tsunogai et al., 2010; Liu et al., 2013; Tsunogai et al., 2014; Rose et al., 2015). These results correspond to mole fractions of unprocessed NO3-(atm) to total nitrate from (1.8±0.3)% to (11.8±1.3)%, obtained by using Eq. (2). By using the concentration ( $C_{total}$ ) and the  $\Delta^{17}$ O values of nitrate, the NO3-(atm) concentration ( $C_{atm}$ ) was calculated using Eq. (2) and is plotted in Fig. 5(e). In addition, the annual average concentration of NO3-(atm) ( $\overline{C}_{atm}$ ) in stream nitrate was calculated and is also presented in Fig. 5(e) as black bars. Please note that the errors in  $C_{atm}$  and  $\overline{C}_{atm}$  are not presented in Figure 5(e) (See Sup. Table S4 for the respective ranges of errors).

To verify possible secular changes (i.e. long-term non-periodic variation), the estimated  $\overline{C}_{total}$ 6  $\overline{\delta^{15}N}$ , and  $\overline{\delta^{18}O}$  for each stream were compared with those determined by Ohte et al. (2010), in 7 which annual average concentration and annual average  $\delta^{15}$ N and  $\delta^{18}$ O values of nitrate (total) 8 were determined for the same streams in 2004 to 2006. Although both concentrations and 9  $\delta^{15}$ N and  $\delta^{18}$ O values in the streams showed significant spatial and temporal variations during 10 2013, as presented in Fig. 5, the annual average values almost correlated with the values 11 determined in 2004 to 2006 (Sup. Fig. S1). The average differences from the values 12 determined in the streams in 2004 to 2006 were +5.3  $\mu$ mol L-1 for  $\overline{C}_{total}$ , +0.6% for  $\overline{\delta}^{15}N$ , and 13 +1.6% for  $\overline{\delta^{18}O}$ , whereas the standard deviation ranges of the differences were 14.9  $\mu$ mol L-1 14 for  $\overline{C}_{total}$ , 1.6‰ for  $\overline{\delta^{15}N}$ , and 2.1‰ for  $\overline{\delta^{18}O}$ . That is, the differences from the values 15 determined in 2004 to 2006 were smaller than their standard deviation ranges, so the 16 differences were not significant. We concluded that secular changes were minimal for nitrate 17 in the streams, at least for the most recent 10-year period of observations. 18

19 **3.3**

**.3 Relationship between $\Delta^{17}$ O and $\delta^{18}$ O**

One of the features in the spatial variation shown in Fig. 5 is the positive correlation between  $\Delta^{17}O$  and  $\delta^{18}O$ . As is clearly represented by the relationship between  $\overline{\Delta^{17}O}$  and  $\overline{\delta^{18}O}$  (Fig. 6), these values showed linear correlation with an  $r^2$  value of 0.88. Because NO3-(atm) is characterised by highly elevated values of both  $\Delta^{17}O$  and  $\delta^{18}O$  (Figs. 3 and 4), changes in the mole fraction of unprocessed NO3-(atm) within the total nitrate pool must be responsible for the positive correlation between  $\overline{\Delta^{17}O}$  and  $\overline{\delta^{18}O}$  for nitrate in the streams.

The slope value of the least–squares-fitted line between  $\overline{\Delta^{17}O}$  and  $\overline{\delta^{18}O}$  (Fig. 6) also supports this hypothesis. By extrapolating the least–squares-fitted line to the region of NO3-(atm) having a  $\Delta^{17}O$  value of +26.3‰, we obtained  $\delta^{18}O = (+86\pm7)\%$ , which also corresponds with the average  $\delta^{18}O$  value of NO3-(atm) of +79.8‰ obtained in Section 3.1. We concluded that the  $\overline{\delta^{18}O}$  values also primarily reflect the mole fraction of NO3-(atm) within nitrate.

Without  $\Delta^{17}$ O data and without some assumptions, it was difficult to decide the major factor 1 controlling the  $\delta^{18}$ O values of the stream nitrate. However, by adding  $\Delta^{17}$ O data, as presented 2 above, it became apparent that the changes in the mole fraction of unprocessed  $NO_3^{-}(atm)$ 3 within the total nitrate pool were primarily responsible for the  $\delta^{18}$ O variation between the 4 streams. These results further support our hypothesis, presented in Section 1, that 5 interpretations on the stable isotopic compositions of nitrate ( $\delta^{15}N$  and  $\delta^{18}O$ ) made without 6  $\Delta^{17}$ O values can often be misleading. When using stable isotopic compositions, particularly 7 the  $\delta^{18}$ O value, of nitrate in freshwater environments to trace its sources and fate, the 8 determination of  $\Delta^{17}$ O values is essential. 9

By extrapolating the linear correlation between  $\overline{\Delta^{17}O}$  and  $\overline{\delta^{18}O}$  to  $\overline{\Delta^{17}O} = 0\%$ , we obtained the 10  $\delta^{18}$ O value of (-2.9±1.2)‰ as the average  $\delta^{18}$ O value of the remineralized portion of nitrate 11 (NO3-(re)) in the streams. Although the  $\delta^{18}$ O value was substantially 18O-depleted compared 12 with that produced through microbial nitrification in soil during in vitro incubation 13 experiments in past studies (Mayer et al., 2001; Burns and Kendall, 2002; Spoelstra et al., 14 2007), it correlated strongly with the  $\delta^{18}$ O values of NO3 (re) determined recently by using 15 the linear relationship between  $\Delta^{17}$ O and  $\delta^{18}$ O of nitrate eluted from forested watersheds, such 16 as  $NO_3^{-}$  (re) in the groundwater of cool-temperate forested watersheds at  $(-4.2\pm2.4)$ %, where 17 the  $\delta^{18}O(H_2O)$  was around -13% (Tsunogai et al. 2010), and NO3 (re) in stream water in a 18 cool-temperate forested watershed at  $(-3.6\pm0.7)$ %, where the  $\delta^{18}O(H_2O)$  was around -11% 19 (Tsunogai et al. 2014). Moreover, the  $\delta^{18}$ O value of NO3 (re) obtained in this study, 20  $(-2.9\pm1.2)$ %, is close to the possible lowermost  $\delta^{18}$ O value of NO3 (re) produced through 21 microbial nitrification under  $H_2O$  of  $(-7.8\pm1.0)$ % (the average and the standard deviation of 22 the  $\delta^{18}$ O values of H2O of the streams; Sup. Table S3) (Buchwald et al., 2012). Furthermore, 23 the  $\delta^{18}$ O value of NO3-re correlates strongly with that obtained through in vitro incubation 24 25 experiments in recent studies that simulated temperate forest soils (Fang et al., 2012). We concluded that the  $\delta^{18}$ O value of NO3 (re) produced through nitrification in the temperate 26 watershed having a  $\delta^{18}O(H_2O)$  value of  $(-7.8\pm1.0)$ % was  $(-2.9\pm1.2)$ % and that we should 27 use such a low  $\delta^{18}$ O value for the NO3 (re) produced through nitrification in the watershed. 28 Understanding the relationship between  $\Delta^{17}$ O and  $\delta^{18}$ O of nitrate shown in Fig. 6 is highly 29 useful for determining the  $\delta^{18}$ O value of NO3 (re) in each watershed (Tsunogai et al. 2010). 30

31 Although the  $\Delta^{17}$ O values of nitrate were stable during partial biogeochemical processinga 32 such as partial removal through assimilation or denitrification, the  $\delta^{18}$ O values of nitrate could

1 vary through the isotopic fractionation processes within each catchment area. Nevertheless, the  $\delta^{18}$ O values of nitrate in the streams plotted on the mixing line between the NO3 (atm) that 2 had been deposited in the watershed and NO3-(re) having  $\delta^{18}$ O and  $\Delta^{17}$ O values close to those 3 produced through nitrification in the catchments. Thus, we concluded that the range of 4 isotopic fractionations owing to partial removal through assimilation or denitrification 5 6 subsequent to deposition of  $NO_3^{-}(atm)$  or production of  $NO_3^{-}(re)$  within each catchment area 7 was generally small for the major portion of nitrate eluted from the watershed. If such isotopic fractionations were significant for the portion of  $NO_3$  (atm) in total nitrate, the data should 8 plot on the 18O-enriched side of Fig. 6, especially for those data enriched in  $NO_3^{-}(atm)$  (i.e. 9 those showing high  $\Delta^{17}$ O values). This result also supports our assumption in Section 3.1 that 10 the actual  $\delta^{15}N$  and  $\delta^{18}O$  values of NO3 (atm) in each stream water sample ( $\delta^{15}N_{atm}$  and 11  $\delta^{18}O_{atm}$  in Eqs. (6) and (7)) correlate with the  $\delta^{15}N_{avg}$  and  $\delta^{18}O_{avg}$  estimated at theSado-seki 12 monitoring station within an error of  $\pm 10\%$ . 13

**14 **3.4** $\delta^{15}$ N values of remineralized nitrate in streams**

To trace the source of the 18O-depleted  $NO_3^{-}(re)$  eluted from the watershed into the lake, the 15 annual average  $\delta^{15}$ N and  $\delta^{18}$ O values of the remineralized portion of nitrate ( $\overline{\delta^{15}}$ Nre and  $\overline{\delta^{18}}$ Ore) 16 in each inflow stream were estimated using Eqs. (6) and (7) and are plotted as a function of 17 18 population density in Fig. 7(c) and Fig. 7(d), respectively. The original  $\delta^{18}$ O is also in Fig. 7(d). Because of the large  $\delta^{18}$ O differences of about 80% between the nitrate in streams and 19 the NO3-(atm), the  $\delta^{18}O_{re}$  values were a few % lower than each original  $\delta^{18}O$  value in total 20 nitrate (Fig. 7(d)). On the contrary, because of the small  $\delta^{15}$ N differences of less than 15‰ 21 22 between the total nitrate in streams and  $NO_3$  (atm), as well as the small  $C_{atm}/C_{total}$  ratios in the streams, most of the  $\delta^{15}N_{re}$  values showed small deviations of less than 1‰ from each 23 corresponding original  $\delta^{15}$ N value in the total nitrate in most of the streams, so the original 24  $\overline{\delta^{15}N}$  values are not presented in Fig. 7(c). Although the annual average  $\overline{\delta^{18}O}_{re}$  values were 25 low and almost uniform, from -4.0% to -0.1%, as implied in the linear correlation between 26  $\overline{\Delta^{17}O}$  and  $\overline{\delta^{18}O}$  in Fig. 6,  $\overline{\delta^{15}N}_{re}$  showed larger variation from +1.7% to +10.9%. 27

Moreover,  $\overline{\delta^{15}N_{re}}$  showed positive linear correlation with the population density in logarithmic scale (r2=0.64, p<0.001) (Fig. 7(c)). A similar trend was reported for the  $\delta^{15}N$  values of total nitrate (= NO3-(atm) + NO3-(re)) in past studies in this watershed (Ohte et al., 2010) and others (Mayer et al., 2002). We further verified that the remineralized portion of nitrate

1 (NO3-(re)) was responsible for the positive correlation between the  $\delta^{15}$ N values of total nitrate 2 and population density that has been found often in various streams in the world.

Both the concentrations and the isotopic compositions shown in Fig. 7 clearly demonstrate 3 that most portions of the nitrate eluted from the catchments with lower population densities of 4 less than 100 km-2, showing  $\delta^{15}$ N values of (+4.4±1.8)‰ and  $\delta^{18}$ O values of about 5  $(-2.3\pm0.9)$ %, were produced through nitrification in naturally occurring soil organic matter 6 7 (Kendall et al., 1995; Ohte et al., 2010). In the latter half of this section, we discuss the source of the 15N-enriched  $NO_3^{-}$  (re) eluted from the catchments with higher population densities of 8 more than 1000 km-2, showing  $\delta^{15}$ N values of (+9.2±1.3)‰ or more and  $\delta^{18}$ O values of about 9 10  $(-2.2\pm1.1)$ %.

Denitrification in riverbed sediments adjacent to riparian zones or groundwater bodies 11 (McMahon and Böhlke, 1996) can increase the  $\delta^{15}$ N value of stream nitrate. However, if such 12 post-production alternation were responsible for the  $^{15}N$  enrichment of  $NO_3^{-}(re)$  and thus the 13 total nitrate, the values of  $\delta^{18}O_{re}$  in addition to those of  $\delta^{15}N_{re}$  would be increased (Granger et 14 al., 2008). Moreover, the absolute concentration of NO3 (atm) (Catm) would decrease in 15 accordance with the progress of denitrification, but the observed Catm was almost uniform 16 irrespective of population density. Of course, the initial  $C_{atm}$  could vary between the streams. 17 However, to explain the observed uniform Catm, unrealistic assumptions are needed for Catm, 18 such as the initial Catm being higher in accordance with higher population density. The low 19 and uniform  $\delta^{18}O_{re}$  values (Fig. 7(d)) as well as the uniform  $C_{atm}$  irrespective of population 20 density (Fig. 7(b)) imply that denitrification in riverbed sediments was minor for the nitrate in 21 the streams. Rather, the  $NO_3^{-}$  (re) must be enriched in 15N from its initial production through 22 nitrification within the catchments with high population densities. In addition, the small 23 differences in  $\delta^{18}$ O values of NO3 (re) between those values irrespective of the population 24 densities in the catchment area (Fig. 7(d)) imply that the essential parameters for determining 25 the  $\delta^{18}$ O values of nitrate during nitrification, such as the  $\delta^{18}$ O values of H2O and the pH of 26 soils (Buchwald et al., 2012; Fang et al., 2012), should be similar among them. 27

Based on the  $\delta^{15}N$  values of total nitrate eluted from catchments with high population densities, as well as the positive correlation between the  $\delta^{15}N$  values of total nitrate and the population densities, Ohte et al. (2010) proposed sewage effluent as the dominant source contributing to the increase in the  $\delta^{15}N$  values of total nitrate eluted from such catchments. The  $\delta^{15}N$  and  $\delta^{18}O$  values of NO3-(re) newly estimated in this study, (+9.2±1.3)‰ or more

and  $(-2.2\pm1.1)$ %, respectively, also imply that the dominant source contributing to the 1 increase in the  $\delta^{15}$ N values of total nitrate had been produced through nitrification in which 2 the source N of the nitrate had already been enriched in 15N. Although the  $\delta^{15}$ N and  $\delta^{18}$ O 3 values of total nitrate in sewage effluent determined in past studies (Aravena et al., 1993; 4 Widory et al., 2005; Wankel et al., 2006; Xue et al., 2009) were a few % higher than the  $\delta^{15}N$ 5 and  $\delta^{18}$ O values of NO3 (re) eluted from the high population density catchments,  $\delta^{15}$ Nre = 6  $(+9.2\pm1.3)$ % and  $\delta^{18}O_{re} = (-2.2\pm1.1)$ %, the slight deviations in the reported  $\delta^{15}N$  and  $\delta^{18}O_{re}$ 7 8 values from our results can be explained by several factors, such as (1) a slight contribution of 9  $NO_3$  (atm) and (2) the progress of denitrification subsequent to production. We concluded that sewage effluent was the most probable pollution source of nitrate to explain the observed 10 concentrations and isotopic compositions of nitrate eluted from the catchments with high 11 population densities, particularly for those of more than  $1000 \text{ km}^{-2}$ . 12

**13 **3.5 Seasonal variation**

Although the annual average values of  $\Delta^{17}$ O and  $\delta^{18}$ O in each stream,  $\overline{\Delta^{17}}$ O and  $\overline{\delta^{18}}$ O. 14 respectively, showed linear correlation, as presented in Fig. 6, the same results were not 15 always attained for those in each season. Particularly for those values obtained during June 16 and August (i.e. summer), some of the streams showed significant deviations in  $\delta^{18}$ O of more 17 than a few ‰ from the hypothetical mixing line between NO3 (atm) ( $\Delta^{17}O = +26.3$  ‰ and 18  $\delta^{18}O = +79.8$  ‰) and NO3 (re) ( $\Delta^{17}O = 0$  ‰ and  $\delta^{18}O = -2.9$  ‰) (Fig. 8). Even though the 19 values of  $\Delta^{17}$ O and  $\delta^{18}$ O of NO3 (atm) showed seasonal variation, as presented in Figs. 3 and 20 4, the large deviations from the mixing line could not be explained based on the seasonal 21 changes in NO3 (atm). Rather, we must assume some seasonal changes in the biogeochemical 22 nitrogen cycles within each catchment area to explain the relationship, because unlike the 23  $\Delta^{17}$ O values, the  $\delta^{18}$ O values of nitrate could vary during biogeochemical processing within 24 each catchment area. As a result, we can evaluate the seasonal changes in the biogeochemical 25 processing within each catchment area by using the seasonal changes in the relationship 26 between  $\Delta^{17}$ O and  $\delta^{18}$ O shown in Fig. 8. 27

The increases in the number of data deviated from the hypothetical mixing line, especially those plotted on the 18O-enriched region (i.e. vertically upward direction in the figures), by more than a few ‰ from the hypothetical mixing line in June and August imply that partial nitrate removal through assimilation or denitrification was active within each catchment area in these months. The spatial differences in the 18O enrichment also support this hypothesis. As
presented in Fig. 8 by the orange squares, 18O enrichment was common in samples obtained
from the southern streams having high population densities in their catchment areas. We can
assume elevated loading of both nutrients and organic matter of anthropogenic origin in these
catchments, both of which naturally enhance assimilation and denitrification.

6 On the contrary, most samples obtained during March and October were distributed on the hypothetical mixing line between  $NO_3^{-}(atm)$  and  $NO_3^{-}(re)$ , as presented in Fig. 8. We 7 8 concluded that the range of isotopic fractionation subsequent to production, such as partial 9 removal through assimilation or denitrification, in winter was generally small for the major portion of nitrate eluted from the watershed and fed into the lake. Therefore, the annual 10 average values (i.e.  $\overline{\delta^{18}O}$  and  $\overline{\Delta^{17}O}$ ) of the streams distributed on the hypothetical mixing line, 11 12 as shown in Fig. 6, because the nitrate influx in winter occupied a major portion of the annual 13 nitrate influx. Active removal of nitrate from the streams through denitrification/assimilation 14 in summer was also responsible for the small relative importance of nitrate influx into the lake in summer. In conclusion, the relationship between  $\Delta^{17}O$  and  $\delta^{18}O$  of nitrate eluted from a 15 catchment area is a useful indicator for evaluating the biogeochemical processing within the 16 17 catchment area, including the seasonal change.

**18 **3.6** Spatial and temporal $\Delta^{17}$ O variation**

19 By using the  $\delta^{18}$ O values of nitrate as a tracer, Ohte et al. (2010) found that the mole fractions 20 of unprocessed NO3-(atm) within the total nitrate pool were high in the northern streams of 21 the watershed in winter, from November to late April. Our present results shown in Fig. 5 22 further verify the past results by adding more robust evidence through the use of the  $\Delta^{17}$ O 23 tracer for NO3-(atm).

Based on the high accumulation rate of snow in the catchment zones of the northern streams, 24 Ohte et al. (2010) concluded that high loading of unprocessed  $NO_3^{-}(atm)$  via snow in the 25 26 catchment zones increased the stored unprocessed  $NO_3^{-}(atm)$  in the snowpack, which was 27 subsequently released into the streams during the melting seasons. This process enhanced the 28 mole fraction of unprocessed  $NO_3^{-}(atm)$  within the total nitrate pool during the melting 29 season, as was also reported for streams worldwide (Kendall et al., 1995; Ohte et al., 2004; 30 Piatek et al., 2005; Ohte et al., 2010; Pellerin et al., 2012; Tsunogai et al., 2014). However, the contribution of nitrate from anthropogenic sources could be smaller in this area because of 31

1 lower population densities in the catchments (Table 1). Because a major portion of the 2 possible anthropogenic nitrate in the catchments must be occupied by  $NO_3^-(re)$  (Ohte et al., 3 2010), a lower  $NO_3^-(re)$  supply from anthropogenic sources in each catchment area could 4 elevate the mole fraction of unprocessed  $NO_3^-(atm)$  within the total nitrate pool, even if the 5 absolute concentration of  $NO_3^-(atm)$  (Catm) was uniform in the streams.

To determine the  $C_{atm}$  variability among the streams, the  $C_{atm}$  values estimated in this study were plotted as a function of population density, as shown in Fig. 7(b). The  $C_{atm}$  was almost uniform at  $(2.3\pm1.1)_{\mu}$ mol L-1 irrespective of changes in the population density of the catchment areas. However, a clear  $C_{total}$  enrichment trend was noted in accordance with increasing population density of the catchments (Fig. 7(a)). Similar  $C_{total}$  enrichment trends have been reported in previous studies (Ohte et al., 2010).

The northern streams such as Nos. 3, 4, and 5 were enriched in Catm, showing Catm annual 12 average values of (5.3±1.1), (2.9±0.7), and (4.3±0.9)  $\mu$ mol L-1, respectively, and  $\Delta^{17}$ O values 13 of +3.1%, +1.9%, and +2.9%, respectively. These results support the previous observation of 14 the streams determined by using the  $\delta^{18}$ O tracer. Similar Catm enrichment of about 3 µmol L-1 15 or more, however, was also found in streams in other areas, such as Nos. 14 ( $C_{atm} = (3.3 \pm 1.0)$ ) 16  $\mu$ mol L-1), 25 ((3.2±0.8)  $\mu$ mol L-1), and 21 ((4.2±0.9)  $\mu$ mol L-1), but these streams showed 17 lower  $\Delta^{17}$ O values of +0.9‰, +1.5‰, and +2.0‰, respectively, and thus low mole fractions 18 19 of unprocessed  $NO_3$  (atm) within total nitrate. We concluded that the difference in the 20 addition of anthropogenic nitrate composed of NO3 (re) in the catchments was primarily 21 responsible for the difference in the mole fraction of unprocessed NO3 (atm) within the total 22 nitrate pool, as well as the Ctotal variation in accordance with the population densities of the 23 catchment areas, as illustrated in Fig. 1. That is, a small contribution of anthropogenic nitrate in the catchments of the northern rivers was primarily responsible for the low Ctotal and thus 24 the high mole fraction of unprocessed  $NO_3^{-}(atm)$  within the total nitrate pool, or the  $C_{atm}/C_{total}$ 25 ratio, in the northern streams of the watershed. 26

Although the difference in the accumulation rate of snow between each catchment zone was not the major factor controlling the  $C_{atm}/C_{total}$  ratios, the concentrated release of NO3 (atm) stored in the snowpack during the melting seasons should be one of the important factors determining the  $C_{atm}$  variation among the streams. Most of the  $C_{atm}$ -enriched streams, such as Nos. 3, 4, 5, 14, and 25, originated from a forested catchment at a high elevation of more than 800 m above sea level; thus, we can anticipate heavy snowpack in winter in each headwater

1 region. Moreover, the maximum Catm values in these streams were found in March, which is 2 the season of snowmelt (Fig. 5). On the contrary, most of the Catm-depleted streams, such as Nos. 29 ((0.9±0.3)  $\mu$ mol L-1), 19 ((0.8±0.3)  $\mu$ mol L-1), 23 ((0.5±0.2)  $\mu$ mol L-1), and 30 3  $((0.6\pm0.2) \mu mol L^{-1})_{a}$  originated from low elevations having urban and suburban catchment 4 5 areas (Table 1). As a result, the concentrated release of stored  $NO_3^{-}(atm)$  in the snowpack to 6 the forest floor in the catchment zone during the melting seasons is strongly responsible for 7 the Catm enrichment of some of the streams, particularly that in the streams during the month 8 of March, as presented in Fig. 1.

9 The only exception is stream No. 21, located in the southernmost part of the watershed, which 10 showed a high annual average  $C_{atm}$  of (4.2±0.9) µmol L-1. This small stream originates from a 11 low elevation of about 200 m and has a small catchment area of 4 km2. In addition, although 12 the other  $C_{atm}$ -enriched streams showed the maximum  $C_{atm}$  in March,  $C_{atm}$  in stream No. 21 13 was highest in August, showing an extraordinarily high value of more than 10 µmol L-1. It is 14 unlikely that NO3-(atm) stored in the snowpack in winter was the major source of NO3-(atm) 15 in this stream.

16 The catchment zone of stream No. 21 had the highest population density of the catchments of the streams studied (Table 1). About one-third of the catchment includes residential areas. 17 18 Artificial drainage systems in urban or residential areas and agricultural lands in humid 19 temperate regions are usually designed to drain rainwater efficiently into streams (Takimoto 20 et al., 1994). As a result, a significant portion of the  $NO_3$  (atm) deposited into the catchment area was deposited onto paved surfaces and was then drained directly into the stream via 21 22 storm sewers without penetrating the ground. Thus, no interaction with soils occurred, as presented in Fig. 1. Because biogeochemical interactions within soils are the major sink for 23  $NO_3$  (atm) and thus for 17O excess of nitrate (Nakagawa et al., 2013; Tsunogai et al., 2014), 24 25 the development of such sewage systems in urban/suburban areas is largely responsible for 26 the high  $C_{atm}$  in stream No. 21. Similar bypassing effects of NO3 (atm) from soil contact by paved surfaces have been suggested in urban/suburban watersheds by using  $\delta^{18}O$  values of 27 nitrate as a tracer (Burns et al., 2009; Kaushal et al., 2011). We further verified that the 28 sewage systems in urban/suburban catchments changed the fate of the  $NO_3^{-}(atm)$  deposited 29 30 onto land to some extent.

The observed uniform  $C_{atm}$  irrespective of population density and headwater elevation shown in Fig. 7 implies that the influences of snowpacks and paved surfaces were still minor in determining the  $C_{atm}$  values in the streams. Rather, the observed stable  $C_{atm}$  implies that most of the NO3 (atm) in the streams had been stored in groundwater/subsurface runoff in the watershed, which had similar Catm concentrations and then gushed to the surface at the respective headwater zones with various elevations and land uses as presented in Fig. 1.

5 When using the  $\delta^{18}$ O tracer, it was difficult to determine the precise absolute concentration of 6 NO3-(atm) (Catm) in each stream watera as presented in this studya and to determine whether 7 the absolute concentration of NO3-(atm) was stable among the streams. However, by using 8 the  $\Delta^{17}$ O values, we can determine the precise Catm in each stream for each season and thus 9 clarify the fate of NO3-(atm).

**10 **3.7** Differences in outflows from inflows**

The concentrations and  $\delta^{15}$ N,  $\delta^{18}$ O, and  $\Delta^{17}$ O values of nitrate in the outflow river (Seta River; 11 No. 33) are also presented in Fig. 5. In a manner similar to the inflow streams (i.e. by using 12 Eqs. (3) to (5)), we estimated the annual average concentration of total nitrate in the outflow 13 river ( $\overline{C}_{total}$ ) to be 13.3 µmol L-1, the annual average  $\delta^{15}N$  value ( $\overline{\delta^{15}N}$ ) to be +13.1‰, the 14 annual average  $\delta^{18}$ O value ( $\overline{\delta^{18}O}$ ) to be +1.5‰, and the annual average  $\Delta^{17}$ O value ( $\overline{\Delta^{17}O}$ ) to 15 16 be +0.9‰, as presented in Fig. 5. Moreover, in a manner similar to that used for the inflow streams (i.e. by using Eqs. (2), (6), and (7)), we estimated the annual average concentration of 17 NO3-(atm) in the outflow river ( $\overline{C}_{atm}$ ) to be (0.4±0.1) µmol L-1 (Fig. 5(e)), the annual average 18  $\delta^{15}N_{re}$  value ( $\overline{\delta^{15}N_{re}}$ ) to be (+13.7±0.6)‰, and the annual average  $\delta^{18}O_{re}$  value ( $\overline{\delta^{18}O_{re}}$ ) to be 19  $(-1.2\pm0.9)$ % (Sup. Table S5). Similar to those for inflows, the  $\Delta^{17}$ O values were typical for 20 21 nitrate in natural stream waters.

The striking features of the outflow in comparison with the inflows were the depletions in the 22 outflow of both Ctotal and Catm as well as the enrichment in 15N (Fig. 5). Because the 23 denitrification/assimilation processes remove both nitrate and  $NO_3$  (atm) and preferentially 24 consume 14N during the removal, the process of denitrification/assimilation in the lake water 25 column can be strongly responsible for the removal of both nitrate and NO3 (atm) and for the 26 15N enrichment of nitrate in the outflow compared with the inflow. If this were the case in 27 Lake Biwa, the total nitrate efflux would have to be smaller than the total nitrate influx. To 28 29 verify this hypothesis quantitatively and to evaluate the influences of the stagnant flow in the 30 lake on nitrate, we estimated the total influx through all of the inflow streams for nitrate and NO3 (atm),  $\Delta N_{in}$ , and  $\Delta A_{in}$ , respectively, and the total efflux for nitrate and NO3 (atm),  $\Delta N_{out}$ , 31

1 and  $\Delta A_{out}$ , respectively, as well as the flow-weighted average  $\delta^{15}N$ ,  $\delta^{18}O$ , and  $\Delta^{17}O$  values of

- 2 all inflows and outflows to discuss their changes in the lake.
- 3 The  $\Delta N_{in}$  and  $\Delta A_{in}$  in each interval between the observation n and the next observation n + 1
- 4 (i.e. each season) and the flow-weighted average  $\delta^{15}N$ ,  $\delta^{18}O$ , and  $\Delta^{17}O$  values of the inflows 5 ( $\delta(n)$ ) during each interval between the observation n and the next observation n + 1 were 6 determined by using the following equations, assuming the same flow rate, the same nitrate
- 7 concentration, and the same isotopic compositions for the interval until the next observation:

$$\qquad \alpha = \frac{Q_{in}}{\sum_{i} q_{i}},\tag{10}$$

9
$$\Delta N_{in}(n) = \sum_{i} C_{i} \cdot f_{i} \cdot \Delta t_{i} \cdot \alpha, \qquad (11)$$

$$\qquad \delta(\mathbf{n}) = \frac{\sum_{i} \delta_{i} \cdot \mathbf{C}_{i} \cdot \mathbf{f}_{i} \cdot \Delta \mathbf{t}_{i}}{\sum_{i} \mathbf{C}_{i} \cdot \mathbf{f}_{i} \cdot \Delta \mathbf{t}_{i}}, \tag{12}$$

11
$$\Delta A_{in}(n) = \Delta N_{in}(n) \cdot \frac{\Delta^{17}O_{in}(n)}{\Delta^{17}O_{atm}},$$
(13)

12
$$\Delta N_{in} = \sum_{n=1}^{4} \Delta N_{in}(n), \qquad (14)$$

13
$$\Delta A_{in} = \sum_{n=1}^{4} \Delta A_{in}(n), \qquad (15)$$

14
$$\delta = \frac{\sum_{n=1}^{2} \delta(n) \cdot \Delta N_{in}(n)}{\sum_{n=1}^{4} \Delta N_{in}(n)},$$
(16)

where  $Q_{in}$  denotes the annual gross influx of water into the lake;  $C_i$  and  $\delta_i$  denote the concentration and isotopic values ( $\delta^{15}N$ ,  $\delta^{18}O$ , or  $\Delta^{17}O$ ) of nitrate in each stream i during each observation n, respectively;  $f_i$  denotes the flow rate of each stream i during each observation n; and  $\Delta t_n$  denotes the time interval between the observation n and the next observation n + 1.

For Qin, we used the annual influx of water estimated by Kunimatsu et al. (1995), in which influx via streams and via groundwater were included. To include the influx of nitrate via groundwater and the other minor streams not measured in this study in the calculations, we

- 1 used the correction factor  $\alpha$  in Eq. (10), whereby we assumed that both the average 2 concentration and average isotopic compositions of the inflows determined in this study 3 represented those of all inflows into the lake, while assuming an error range of 20% in  $\alpha$ . 4 Under the Qin, the correction factor  $\alpha$  used in this study became 1.9±0.4.
- By using the aforementioned equations, we estimated the total influx of nitrate to the lake ( $\Delta N_{in}$ ) for each interval, together with the average  $\delta^{15}N$ ,  $\delta^{18}O$ , and  $\Delta^{17}O$  values of nitrate during each interval, as presented in Table 2. Moreover, by using the values of  $\Delta N_{in}$  during each interval, as well as their  $\delta^{15}N$ ,  $\delta^{18}O$ , and  $\Delta^{17}O$  values, we estimated the total influx of  $NO_3^-(atm)$  to the lake ( $\Delta A_{in}$ ) and the average  $\delta^{15}N_{re}$  and  $\delta^{18}O_{re}$  values for each interval, as presented in Table 2, by using Eqs. (13), (6), and (7). Furthermore, we estimated the annual total influx and the various annual average influx values, as shown in Table 2.
- The annual average  $\Delta^{17}$ O value of inflows estimated by using Eq. (16) was +1.3‰, which corresponds to an average mole fraction of NO3-(atm) within total nitrate of (5.1±0.5)% by Eq. (2). We concluded that about 5% of the total nitrate in the inflows originated directly from the atmosphere; therefore, the remainder of the nitrate had a remineralized origin (NO3-(re)) and was likely produced through nitrification within the catchments, as discussed in Section 3.4. In addition, we estimated the annual total influx of nitrate to the lake ( $\Delta N_{in}$ ) to be (199±40) Mmol and that of NO3-(atm) ( $\Delta A_{in}$ ) to be (10.1±2.0) Mmol.
- 19 Moreover, we estimated the total efflux of nitrate and  $NO_3^{-}(atm)$  from the lake via the 20 outflows ( $\Delta N_{out}$  and  $\Delta A_{out}$ ) for each interval by using Eqs. (10) to (16) in which  $\Delta N_{in}$  was replaced with  $\Delta N_{out}$ , and  $\Delta A_{in}$  was replaced with  $\Delta A_{out}$ . Additionally,  $Q_{in}$  was replaced with 21 22 Qout, which is the annual gross efflux of water. To include the minor effluxes of nitrate to 23  $\Delta N_{out}$ , such as those via canals, we used the correction factor  $\gamma$  instead of  $\alpha$  in Eqs. (10) and 24 (11), whereby we assumed that both the concentration and isotopic compositions of the natural outflow determined for each season in this study represented all outflows. For Qout, we 25 26 used the annual efflux of water from Lake Biwa estimated by Kunimatsu et al. (1995), which included the efflux via a natural river (Seta River, No. 33) and that via canals. Under the Qout, 27 28 the correction factor  $\gamma$  used in this study became 1.1±0.2.
- 29 Compared with the annual  $\Delta N_{in}$  and the annual  $\Delta A_{in_s}$  both the annual  $\Delta N_{out}$  and the annual 30  $\Delta A_{out}$  were significantly smaller by about 66% and 78%, respectively. Hence, Lake Biwa acts 31 as a net sink for both nitrate and  $NO_3^-$  (atm), as previously implied from the 15N enrichment in 32 outflows. Considering that nitrate constituted about 70% of the total fixed N pool in the

inflows and about 40% of the total fixed N pool in the outflows (Shiga prefecture, 2015), 1 2 Lake Biwa also acts as a net sink for fixed N. Similar results were obtained in previous 3 studies that discussed the fixed N input/output of the lake (Tezuka, 1985, 1992; Kunimatsu, 1995; Yamada et al., 1996). As implied by the significant 15N enrichment in the remineralized 4 portion of nitrate  $(\delta^{15}N_{re})$  in the outflow, (+13.7±0.6)‰, compared to the inflow, 5 (+5.6±0.3)‰, partial removal of nitrate through either assimilation or denitrification is 6 strongly responsible for the (8.1±1.1)‰ increase in  $\delta^{15}N_{re}$  as well as the net removal of both 7 nitrate and  $NO_3^{-}(atm)$  from the lake. 8

On the contrary, the  $\delta^{18}$ O differences in the remineralized portion of nitrate ( $\delta^{18}$ Ore) between 9 the inflows and outflows were significantly smaller than those of  $\delta^{15}N_{re}$ , at an annual average 10 of only (1.6±1.8)‰ (Table 2). If the nitrate in the outflows is the residual nitrate of 11 assimilation/denitrification in the lake,  $\delta^{18}O_{re}$  should also increase (Granger et al., 2004; 12 Granger et al., 2008). The much smaller  $\delta^{18}O_{re}$  difference implies that nitrate supplied directly 13 14 from inflows occupied a small portion of the nitrate in the outflows and that most of nitrate with high  $\delta^{15}$ N values in the outflows was produced through nitrification in the lake water 15 column in which the fixed N was enriched in 15N. Isotopic fractionations during fixed N 16 cycling in the lake, such as during denitrification or assimilation, and the subsequent removal 17 of 15N-depleted organic N during sedimentation (Fig. 1) are likely responsible for the 15N 18 enrichment of the total fixed N. That is, most of nitrate fed into the lake via the inflows was 19 20 removed at least once from the lake water column and was involved in the total fixed N cycling in the lake, in which the 15N-enriched nitrate in the outflow was produced (Fig. 1). 21 The stagnation of flow in the lake (around 5 years) encouraged primary production and thus 22 23 the net removal of total fixed N through either denitrification or sedimentation, which resulted in 15N enrichment of the total fixed N pool compared with that in the inflows. Further studies 24 on N cycling in the lake are needed to verify these results. 25

26

**27 4 Concluding Remarks**

In this study, we applied the  $\Delta^{17}$ O tracer of nitrate to determine accurate and precise mole fractions of unprocessed NO3-(atm) within the total nitrate value for more than 30 streams in the Lake Biwa watershed basin. Although the nitrate concentration varied from 12.7 to 106.2  $\mu$ mol L-1 among the inflow streams and the mole fraction of NO3-(atm) within the total nitrate also varied from 1.8% to 11.8%, the absolute concentration of NO3-(atm) (Catm) in

each stream water was almost stable at  $(2.3\pm1.1)$  µmol L-1 irrespective of the changes in 1 population density and land use among the catchment areas. We concluded that changes in 2 population density and land use among the catchment areas had little impact on Catm and that 3 the total nitrate concentration was determined primarily by the extent of the additional 4 5  $NO_3$  (re) contribution, which was mostly from anthropogenic sources. When relying on only the  $\delta^{15}$ N and  $\delta^{18}$ O tracers of nitrate, it was difficult to determine the precise Catm in the stream 6 water and whether  $C_{atm}$  was uniform among the streams. By using the  $\Delta^{17}O$  values, we were 7 8 able to estimate accurate and precise Catm in each stream for each season; thus, we could 9 clarify the fate of the  $NO_3^{-}(atm)$  deposited into the catchments.

Moreover, additional measurements of the  $\Delta^{17}O$  values of nitrate together with  $\delta^{15}N$  and  $\delta^{18}O$ 10 enabled us to exclude the contribution of NO3 (atm) from the determined  $\delta^{15}N$  and  $\delta^{18}O$ 11 values and to use the corrected  $\delta^{15}$ N and  $\delta^{18}$ O values,  $\delta^{15}$ Nre and  $\delta^{18}$ Ore, to evaluate the source 12 and behaviour of NO3 (re) in each stream. Based on the correction, we successfully estimated 13 the  $\delta^{15}$ N and  $\delta^{18}$ O values of NO3 (re) in the streams to be (+4.4±1.8)‰ and (-2.3±0.9)‰, 14 respectively, for  $NO_3^{-}$  produced through nitrification in naturally occurring soil organic 15 matter and  $(+9.2\pm1.3)$ % and  $(-2.2\pm1.1)$ %, respectively, for NO3- supplied from 16 anthropogenic sources, most of which were sewage effluent. In addition, the low and uniform 17 annual average  $\delta^{18}O_{re}$  values of NO3 (re) in the streams implied that denitrification in the 18 riverbed sediments was minor in the streams. 19

Furthermore, we clarified the seasonal changes in the range of isotopic fractionation through partial nitrate removal via assimilation or denitrification by using the relationship between  $\Delta^{17}O$  and  $\delta^{18}O$  of nitrate in the streams. The changes were small in winter in all of the catchment areas but large in summer in some catchments. Therefore, the relationship between  $\Delta^{17}O$  and  $\delta^{18}O$  of nitrate eluted from a catchment area is a powerful indicator for evaluating the biogeochemical nitrogen cycles within a catchment area, including the seasonal changes.

Based on the annual influx and efflux of both nitrate and  $NO_3^-(atm)$  in Lake Biwa newly estimated in this study, we found that Lake Biwa is a net sink for both nitrate and  $NO_3^-(atm)$ . Additionally, we found significant 15N\_enrichment ((+8.1±1.1)‰) in the remineralized portion of nitrate in the outflow compared with those in the inflows, whereas the 18O\_enrichment was only (+1.6±1.8)‰. We concluded that most of the nitrate fed into the lake via the inflows was removed at least once from the lake water column and was involved in the total fixed N cycling in the lake, by which the 15N-enriched nitrate in the outflow was produced (Fig. 1). The stagnation of flow in the lake encouraged primary production and thus net removal of
 total fixed N through either denitrification or sedimentation, which resulted in 15N enrichment
 of the total fixed N pool compared with that of the inflows.

**4 Acknowledgements**

5 We are grateful to Kosuke Ikeya, Hiroki Sakuma, Sho Minami, Kenta Ando, Shuichi Hara, 6 Toshiyuki Matsushita, Takahiro Mihara, Teresa Fukuda, Yoshiumi Matsumoto, Rei Nakane, Lin Cheng, Yuuko Nakano, and other present and past members of the Biogeochemistry 7 8 Group, Nagoya University, for their valuable support throughout this study. We thank Drs. 9 Shin-ichi Nakano, Tadatoshi Koitatabashi, Yukiko Goda, and other staff of the Center for 10 Ecological Research, Kyoto University, for their valuable support during the field study in the Lake Biwa watershed basin. We also thank the members of the Machine Shop of Nagova 11 12 University Technical Center for their valuable support in developing the sampling and 13 analytical devices used in this study. We thank anonymous reviewers for valuable remarks on 14 an earlier version of this manuscript. This work was supported by a Grant-in-Aid for Scientific Research from the Ministry of Education, Culture, Sports, Science, and Technology 15 16 of Japan under grant numbers 24651002, 26241006, and 15H02804.

17

**18 **References**

[revised manuscript text omitted]

- 6

| No.    | Name    | Loc.# | Basin
Area * | Population
Density * | Land
Use \$ | No | . Name   | Loc.# | Basin
Area * | Population
Density * | Land
Use \$ |
|--------|---------|-------|----------------------------|------------------------------------|---------------------------|----|----------|-------|----------------------------|------------------------------------|---------------------------|
|        |         |       | $(km^2)$                   | $({\rm km}^{-2})$                  |                           |    |          |       | $(km^2)$                   | $({\rm km}^{-2})$                  |                           |
| Inflow |         |       |                            |                                    |                           |    |          |       |                            |                                    |                           |
| 31     | Tenjin  | West  | 10                         | 539                                | Forest                    | 14 | Seri     | East  | 74                         | 462                                | Forest                    |
| 30     | Mano    | West  | 23                         | 1048                               | Forest                    | 15 | Inukami  | East  | 102                        | 109                                | Forest                    |
| 29     | Wani    | West  | 17                         | 186                                | Forest                    | 16 | Ajiki    | East  | 15                         | 1002                               | Agr                       |
| 28     | U       | West  | 7                          | 66                                 | Forest                    | 17 | Uso      | East  | 84                         | 411                                | Agr                       |
| 1      | Kamo    | West  | 47                         | 89                                 | Forest                    | 18 | Bunroku  | East  | 14                         | 595                                | Agr                       |
| 2      | Ado     | West  | 306                        | 27                                 | Forest                    | 19 | Nomazu   | East  | 7                          | 758                                | Agr                       |
| 3      | Ishida  | North | 60                         | 84                                 | Forest                    | 20 | Echi     | East  | 211                        | 110                                | Forest                    |
| 4      | Momose  | North | 13                         | 65                                 | Forest                    | 27 | Hino     | South | 226                        | 338                                | Forest                    |
| 5      | Chinai  | North | 51                         | 44                                 | Forest                    | 26 | Yanomune | South | 42                         | 859                                | Agr                       |
| 6      | Ohura   | North | 39                         | 98                                 | Forest                    | 25 | Yasu     | South | 391                        | 324                                | Forest                    |
| 7      | Oh      | North | 20                         | 55                                 | Forest                    | 24 | Yamaga   | South | 6                          | 2540                               | Agr                       |
| 8      | Yogo    | North | 7                          | 141                                | Forest                    | 23 | Sakai    | South | 2                          | 979                                | Agr                       |
| 9      | Chonoki | North | 10                         | 412                                | Agr                       | 22 | Hayama   | South | 34                         | 2048                               | Agr                       |
| 10     | Та      | North | 36                         | 301                                | Agr                       | 34 | Kusatsu  | South | 48                         | 370                                | Agr                       |
| 11     | Ane     | North | 372                        | 61                                 | Forest                    | 21 | Nagaso   | South | 4                          | 3174                               | Res                       |
| 12     | Yone    | North | 15                         | 2047                               | Agr                       | 32 | Fujinoki | South | 4                          | 1805                               | Forest                    |
| 13     | Amano   | North | 111                        | 226                                | Forest                    |    |          |       |                            |                                    |                           |
|        |         |       |                            |                                    |                           | Ou | Outflow  |       |                            |                                    |                           |
|        |         |       |                            |                                    |                           | 33 | Seta     | South | 3848                       | 323                                | Forest                    |

**Category of location classified by Ohte et al. (2010).**

2

\* Data source: Ohte et al. (2010).

§ Dominant land use in the respective catchment of agricultural land (Agr), forest, or residential (Res). See Table S1 for the specific contents.

- 1 Table 2 Estimated gross influx/efflux of total nitrate ( $\Delta N$ ) and atmospheric nitrate ( $\Delta A$ ) via
- 2 inflows/outflows during each observation interval, together with the average  $\delta^{15}$ N,  $\delta^{18}$ O, and
- $\Delta^{17}$ O values of total nitrate and remineralized portions of nitrate ( $\delta^{15}$ Nre and  $\delta^{18}$ Ore) in the

| 4 in | flows/outflows | during | each | interval |
|------|----------------|--------|------|----------|
|------|----------------|--------|------|----------|

|                                       | Spring         | Summer         | Autumn         | Winter          | Annual          |
|---------------------------------------|----------------|----------------|----------------|-----------------|-----------------|
|                                       | (n=1 to 2)     | (n=2 to 3)     | (n=3 to 4)     | (n=4 to 5)      | (n=1 to 5)      |
| Duration (days)                       | 94             | 49             | 77             | 145             | 365             |
| Inflow                                |                |                |                |                 |                 |
| $\Delta N_{in}$ (10 6 mol) | $69 \pm 14$    | $3 \pm 1$      | $13 \pm 3$     | $114 \pm 23$    | $199\pm40$      |
| $\Delta A_{in}$ (10 6 mol) | $6.4 \pm 1.3$  | 0.1            | $0.8 \pm 0.2$  | $2.8 \pm 0.6$   | $10.1 \pm 2.0$  |
| $10^3 \delta^{15} N$                  | +4.0           | +6.8           | +5.6           | +5.6            | +5.1            |
| $10^3 \delta^{18}$ O                  | +6.1           | -0.8           | +3.3           | -1.5            | +1.4            |
| $10^3 \Delta^{17} O$                  | +2.5           | +0.8           | +1.7           | +0.6            | +1.3            |
| $10^3 \delta^{15} N_{re}$             | $+4.8\pm0.7$   | $+7.1\pm0.2$   | $+6.3\pm0.5$   | $+5.9\pm0.2$    | $+5.6\pm0.3$    |
| $10^3  \delta^{18} O_{re}$            | $-1.5 \pm 1.8$ | $-3.2 \pm 0.5$ | $-2.0 \pm 1.2$ | $-3.5\pm0.4$    | $-2.8\pm0.9$    |
| Outflow                               |                |                |                |                 |                 |
| $\Delta N_{out} (10^6 \text{ mol})$   | $24 \pm 5$     | $6 \pm 1$      | $5 \pm 1$      | $32 \pm 6$      | $67 \pm 13$     |
| $\Delta A_{out} (10^6 \text{ mol})$   | $1.4 \pm 0.1$  | 0.1            | 0.2            | $0.4 \pm 0.1$   | $2.2 \pm 0.4$   |
| $10^3 \delta^{15} N$                  | +7.3           | +11.4          | +10.4          | +18.0           | +13.1           |
| $10^3 \delta^{18}$ O                  | +3.4           | +4.8           | +3.0           | -0.7            | +1.5            |
| $10^3 \Delta^{17} O$                  | +1.6           | +0.4           | +1.4           | +0.4            | +0.9            |
| $10^3 \delta^{15} N_{re}$             | $+8.1\pm0.6$   | $+11.7\pm0.4$  | $+11.2\pm0.9$  | $+18.3 \pm 0.4$ | $+13.7 \pm 0.6$ |
| $10^3  \delta^{18} O_{re}$            | $-1.5 \pm 1.4$ | $+3.6\pm0.6$   | $-1.2 \pm 1.2$ | $-1.9\pm0.5$    | $-1.2\pm0.9$    |

Figure 1. Schematic diagram showing the biological processing of atmospheric nitrate  $(NO_3^{-}(atm))$  and remineralized nitrate  $(NO_3^{-}(re))$  in the watershed with catchments of various land uses and in the lake water column.